# Active machine learning model for the dynamic simulation and growth mechanisms of carbon on metal surface

Di Zhang [1,2] ✉, Peiyun Yi[1,3], Xinmin Lai[1,3], Linfa Peng[1,3] ✉ & Hao Li [2] ✉

Substrate-catalyzed growth offers a highly promising approach for the controlled synthesis of carbon nanostructures. However, the growth mechanisms on dynamic catalytic surfaces and the development of more general design strategies remain ongoing challenges. Here we show how an active machine-learning model effectively reveals the microscopic processes involved in substrate-catalyzed growth. Utilizing a synergistic approach of molecular dynamics and time-stamped force-biased Monte Carlo methods, augmented by the Gaussian Approximation Potential, we perform fully dynamic simulations of graphene growth on Cu(111). Our findings accurately replicate essential subprocesses–from the preferred diffusion of carbon monomer/dimer, chain or ring formations to edge-passivated Cu-aided graphene growth and bond breaks by ion impacts. Extending our simulations to carbon deposition on metal surfaces like Cu(111), Cr(110), Ti(001), and oxygen-contaminated Cu(111), our results align closely with experimental observations, providing a practical and efficient approach for designing metallic or alloy substrates to achieve desired carbon nanostructures and explore further reaction possibilities.

Controllable synthesis of carbon nanomaterials, such as single-crystalline, large-area graphene, specifically chiral carbon nanotubes, or carbon films with certain $sp^2$ or $sp^3$ content, is a crucial challenge for realizing their potential applications in future electronics or energy devices[1–3]. The substrate-catalyzed deposition has been considered to be one of the most promising ways to achieve the controllable growth of two or three-dimensional covalently bonded network of carbon atoms[4,5], because the surface composition and facet identity of the underlying substrate greatly affect their crystallinity[6,7], orientation[8], edge geometry[9], and compressive strain[10]. Apart from the well-explored metallic catalysts like Cu and Ni, there have been reports on the efficient production of wafer-scale single-crystalline graphene on less common substrates such as high-index Cu facets[7], Cu-Ni alloys[11], $Cu_2O$ surface[12], hydrogen-terminated Ge[13], and hexagonal boron nitride[14]. While the growth mechanisms on common surfaces have been extensively studied, there is limited knowledge regarding the dynamic and atomic-level factors that govern the quality of graphene on high-index or composite surfaces, including nucleation and growth kinetics. This research gap significantly hinders the development of theory-guided design approaches for novel catalyzed metal substrates in the growth of carbon nanostructures.

Searching for metallic or alloy catalysts experimentally poses a considerable challenge, primarily due to the extensive range of potential substrates and the sensitivity of the carbon nanomaterial growth process to various experimental parameters. Thus, there is plenty of room for theoretical simulations, where many atomic details are readily available. As a standard approach to study a variety of chemical processes, ab initio density functional theory (DFT)

[1]State Key Laboratory of Mechanical System and Vibration, Shanghai Jiao Tong University, 200240 Shanghai, People's Republic of China. [2]Advanced Institute for Materials Research (WPI-AIMR), Tohoku University, Sendai 980-8577, Japan. [3]Shanghai Key Laboratory of Digital Manufacture for Thin-walled Structures, Shanghai Jiao Tong University, 200240 Shanghai, People's Republic of China. ✉e-mail: zhangdi2015@sjtu.edu.cn; penglinfa@sjtu.edu.cn; li.hao.b8@tohoku.ac.jp

calculations can compute the step-by-step energy levels of a sequence of states from carbon feedstock, carbon monomer[15], dimers[16], or small clusters[17] to final graphene or carbon films on metal[18,19] or alloy surfaces[11], atomic steps[20–22], dislocations[4], grain boundaries[9,23], and subsurface[24]. Theoretical simulations, such as kinetic Monte Carlo (KMC) and ab initio molecular dynamics (AIMD), have significantly enhanced the understanding of graphene growth on certain metal surfaces. However, it is still challenging to go beyond these atomic details to obtain the fully dynamic and overall picture of carbon growth on an arbitrary metal or alloy substrate. For instance, a DFT-based KMC model[25] was employed to study the diffusion-limited growth of carbon dimers on Cu(111) and Cu(100). Nevertheless, KMC simulations have limitations in capturing the complete dynamics of carbon growth on a dynamically evolving substrate due to the difficulty in constructing event tables for the off-lattice simulations[26]. On the other hand, AIMD suffers from limited time and length scales despite advancements in computational power. Conventional methods such as MD/MC simulations with empirical interatomic potentials face difficulties in accurately calculating interactions between graphitic layers[27], experimentally observed $sp^2/sp^3$ fractions[3], and the interactions between carbon species and metal surfaces[28] due to the limitations of their fixed functional form. Therefore, there is an ongoing and urgent demand for a robust design model capable of accurately depicting the growth mechanisms of carbon on metallic surfaces.

Machine-learning potentials (MLPs) based on artificial neural networks[29–31] or kernel-based methods[32–34] have been suggested to be a powerful method to address the limited accuracy and transferability of classic force fields and maintain a DFT-level accuracy. This is well demonstrated in the studies on understanding the growth mechanisms of high $sp^3$ content in tetrahedral amorphous carbon[3,35], the pressure-induced phase transition in silicon[36], and complex aqueous systems[37]. Despite these significant achievements in data-driven MD simulations, the construction of an accurate MLP remains a difficult task because it can take years to sample in broad regions of phase space, especially for deposition simulations, in which high-energy incidents can lead to locally strongly disordered structures and the potential should be highly flexible[35]. One solution to this issue is the on-the-fly learning techniques[37,38], which allow one to sample a small region of configuration space under specific thermodynamic and boundary conditions. In the production of the on-the-fly training set, the selection of the most diverse structures is crucial for achieving high efficiency and accuracy[39]. So far, several selective principles have been proposed to construct a training set with predictive uncertainties, including the Bayesian inference[38,40], spilling factor[41], Maxvol method[42], and CUR decomposition[43]. Notably, the CUR decomposition method stands out because it does not rely on a prior assumption of a Gaussian distribution for potential energy, as is often the case in Bayesian inference. Additionally, with its foundation in a configuration-averaged metric, the CUR decomposition method has consistently shown impressive accuracy and robustness, especially when applied to carbon-based and metal materials[43]. However, relying only on the configuration-averaged metric for selecting new structures during deposition simulation could omit structures that exhibit significant variations only in the local areas surrounding the deposited atom. To enhance the efficiency and effectiveness in the on-the-fly training of deposition processes, a well-defined selection protocol is required. On the other hand, the dynamics of carbon growth on a metal substrate can be governed by important rare events[44], such as surface diffusion and graphene nucleation. Therefore, how to enhance the training efficiency of MLPs by coupling enhanced sampling methods to classical dynamics needs further study.

This work presents an approach to generate MLPs with minimum human effort through a data-driven automatic learning framework suited for carbon growth on metal or alloy surfaces. To achieve this task, we make use of (1) the Gaussian Approximation Potential (GAP) machining learning model, which has been widely validated to capture the deposition process correctly when benchmarked against experiments, including the phonon dispersion relations, thermal expansion, and Raman spectra at different temperatures, and carbon depositions[3,33–35]; (2) an enhanced sampling method named time-stamped force-biased Monte Carlo (tfMC) method[27,45,46] to accelerate the relaxation process after carbon depositions, thus including the important rare events in the training database; (3) an effective strategy for the selection of representative training data based on the descriptor of smooth overlap of atomic positions (SOAP)[47]; (4) a well-developed carbon training set[34]; (5) an automated screening, fitting, and validation procedure. The resulting potential is then applied to study the deposition growth of carbon atoms on a Cu (111) surface. This approach can correctly capture the critical processes of carbon growth on Cu (111), such as the formation and migration of subsurface carbon monomer and surface dimer[5], the appearance of 1-dimensional carbon nanoarches[18], graphene nucleation involved with edge-passivated Cu atoms and carbon chains, and the precipitated growth process[4]. Our simulations of the initial nucleation on different metallic surfaces, specifically carbon deposition on Cu(111), Cr(110), Ti(001), and O-contaminated Cu(111), exhibit consistency with experimental observations and DFT calculations. This study provides a practical and efficient methodology for designing metallic or alloy substrates to achieve desired carbon nanostructures and explore further opportunities.

## Results

Figure 1 illustrates the schematic of an active-learning algorithm by which an MLP is generated on the fly during the MD/tfMC simulation of graphene growth on a Cu (111) surface. The carbon-growth-on-metal machine-learning potential is henceforth dubbed CGM-MLP. The construction of CGM-MLP begins with a well-developed database for carbon potential functions generated by Rowe et al.[34], namely GAP-20, and several equilibrium configurations of carbon clusters on Cu (111) surface (i.e., $C_1$–$C_{18}$ in Fig. 1a). By including all of these structures, the initial training dataset provides a good starting point for generating an MLP capable of accurately describing Cu-C interactions. The potential energy and forces of the structures in the initial training set are calculated using DFT to obtain the initial iteration of the MLP using the GAP methodology[32]. Subsequently, the generated MLP is employed in a hybrid MD/tfMC simulation to simulate carbon deposition on the Cu(111) surface.

During the MD/tfMC simulations, four carbon atoms are deposited in each round of training. From these simulations, a subset of structures is generated with a sampling rate of $N_f$ frame per carbon atom. A screening process is then employed, using SOAP-based similarity metrics (see Methods for the definitions) to refine the dataset further. The similarity measure is the heart of the active-learning algorithm, determining whether a new structure should be added to the training set and ensuring a balanced representation of different regions of the phase space. Notably, this work introduces an additional measure, $D^{max}$, which, though resembling $D^{ave}$ in form, plays a pivotal role in refining the on-the-fly training set. These measures specifically capture the maximum and average SOAP-based distances between atoms in newly observed and previously chosen structures, respectively.

By considering both maximum and average SOAP distances, we can effectively capture variations in local environments around the deposited atoms, which may be overlooked if only the average distance[43] is considered (see Supplementary Fig. 4). Hence if a new structure's $D^{ave}$ or $D^{max}$ exceeds the specified screening parameter ($S^{max}$ or $S^{ave}$), it will be included the new structure in the training set. To evaluate the effectiveness of the screening parameters, we selected a subset of structures from the MD/tfMC simulations using different

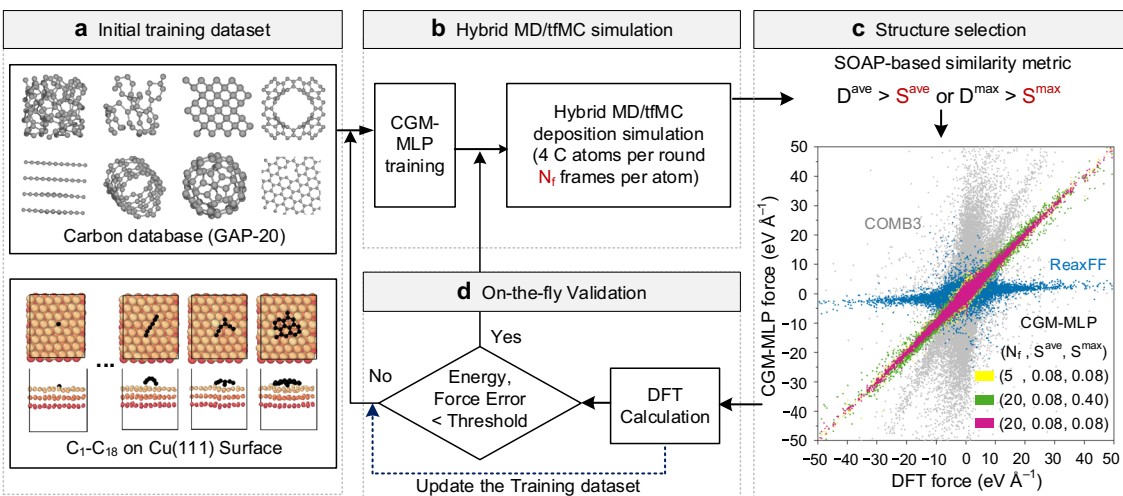

**Fig. 1 | Schematic illustrations of carbon-growth-on-metal machine-learning potential (CGM-MLP) generated by active learning on-the-fly during hybrid molecular dynamics and time-stamped force-biased Monte Carlo (MD/tfMC) simulations. a** The initial training dataset includes representative carbon structures from Gaussian Approximation Potential (GAP-20)[34] and $C_1$-$C_{18}$ carbon clusters on Cu(111) surfaces. **b** The CGM-MLP trained from this dataset is then used in a deposition simulation employing a hybrid MD/tfMC method[27]. **c** A smooth overlap of atomic positions (SOAP-based) algorithm is used to select the most representative structures from the MD/tfMC simulations. The inset figure presents the force correlation plots by using different quality control parameters, namely $N_f$ (the number of structures sampled for each deposited carbon atom), $S^{max}$, and $S^{ave}$ (i.e., the thresholds for the maximum and average SOAP distances, $D^{ave}$ and $D^{max}$). The definitions of the similarity matrix $D^{ave}$ and $D^{max}$ are available in the "Methods" section. Source data and code are provided. **d** DFT benchmarks energy and force, and if the error is below a threshold, MD/tfMC continues. Otherwise, the training dataset is updated with newly selected structures.

values of $N_f$, $S^{max}$, and $S^{ave}$. Meanwhile, approximately 500 structures were randomly selected from the held-out structures to serve as the testing sets. By increasing $N_f$ and lowering $S^{max}$, as shown in Fig. 1c, the CGM-MLP exhibits improved force accuracies and a significant improvement compared to classical empirical potentials, such as COMB3[48] and ReaxFF[49]. In Supplementary Fig. 3, we also provide an energy correlation plot and additional tests of the parameter $S^{max}$ (Supplementary Fig. 4) for various structure types (Supplementary Table 1). The results demonstrate that a predictive energy mean absolute error ($E_{MAE}$, with the detailed formula provided in Supplementary Method 3) of below 0.05 eV atom$^{-1}$ and a force MAE of less than 0.5 eV Å$^{-1}$ are attainable for all relevant structures in a Cu-C system when $S^{max}$ is set to less than 0.08 (Supplementary Method 4). Specifically, the energy and force MAE of the CGM-MLP trained with $N_f = 20$ and $S^{ave}$, $S^{max} = 0.08$ converge to approximately 0.013 eV atom$^{-1}$ and 0.43 eV Å$^{-1}$, respectively. Given the intrinsic challenges associated with complex hybridizations in carbon and the long-range interactions beyond the MLPs' cutoff, the energy and force errors we observed are believed to approach the peak accuracy achieved by MLPs for systems containing amorphous or defective carbon[34,50]. A more detailed test of long-range interactions for metal-carbon systems can be found in Supplementary Discussion. 1 and Supplementary Fig. 5. As a practical and efficient means to explore a wide range of reaction pathways during carbon growth, it is believed that the achieved energy and force MAE are appropriate for the research goals. Given the distinct effects of long-range interactions across different metal-carbon systems, strategies such as adopting varying force or energy thresholds for distinct deposition stages, or setting system-dependent total-energy training convergence thresholds, can be considered to further enhance accuracy. Yet, addressing the challenges posed by long-range interactions remains crucial for refining the precision and adaptability of simulations, especially those centered on metal-catalyzed low-density carbon structures.

During the error estimation step (Fig. 1d), we use an energy MAE of <0.05 eV atom$^{-1}$ and a force MAE of <0.5 eV Å$^{-1}$ to determine the inclusion of newly generated structures in the training set or if carbon atom deposition can proceed without further training. Additional details in the training process can be found in Supplementary Fig. 6.

The decision to end the iterations depends on the desired number of carbon atoms to be deposited on the metal surface. In this study, the number of carbon atoms to be deposited is predetermined as 100 carbon atoms. Additionally, if further growth processes or the simulation of carbon-based films are of interest, the predetermined number of carbon atoms can be increased accordingly.

By leveraging the high accuracy of the CGM-MPL and incorporating rare atomistic events in the MD/tfMC method[51], we successfully replicate essential subprocesses associated with graphene nucleation and carbon growth on metallic surfaces, as exemplified in Fig. 2.

### Carbon monomer and dimer

In the first stage, impinging gas phase carbon atoms readily adsorb on the Cu(111) surface and subsequently diffuse to the sublayer of the Cu(111) surface spontaneously (Fig. 2a, $C_7$). This process is consistent with previous static DFT calculations, which suggested that C monomers prefer to remain at an octahedral site in the subsurface layer of Cu(111). The adsorption energy of the sublayer Cu atoms increases by about 0.5 eV[15]. In all deposition simulations with different impinging energies, no carbon atom enters the bulk of the Cu substrate in the initial stage. This is consistent with previous findings that carbon exhibits very low solubility in copper[1]. During the ongoing deposition process, subsurface carbon atoms have the potential to intermittently form bonds. Once carbon dimers are formed in the subsurface, they can subsequently migrate back to the surface regions (Fig. 2a, $C_{14}$ or Fig. 2b, $C_{11}$). Notably, our simulations indicate that carbon monomers do not always migrate to the subsurface layer. Instead, if the surface copper atoms can absorb the kinetic energy of the incident carbon atoms and transform them into adsorbed copper atoms, vacancies may be created on the Cu(111) surface. In cases where the deposited carbon atoms are near the adsorbed copper atoms, they tend to remain on the surface rather than migrate to the subsurface.

### Carbon-chain formation

Owing to the continued addition of carbon, the extrusive carbon dimers start to connect to the carbon atoms stabilized by adsorbed Cu or off-surface Cu atoms. The short carbon chains appear on Cu(111) surface (Fig. 2a, $C_{43}$). When the incident energy of carbon is as low as

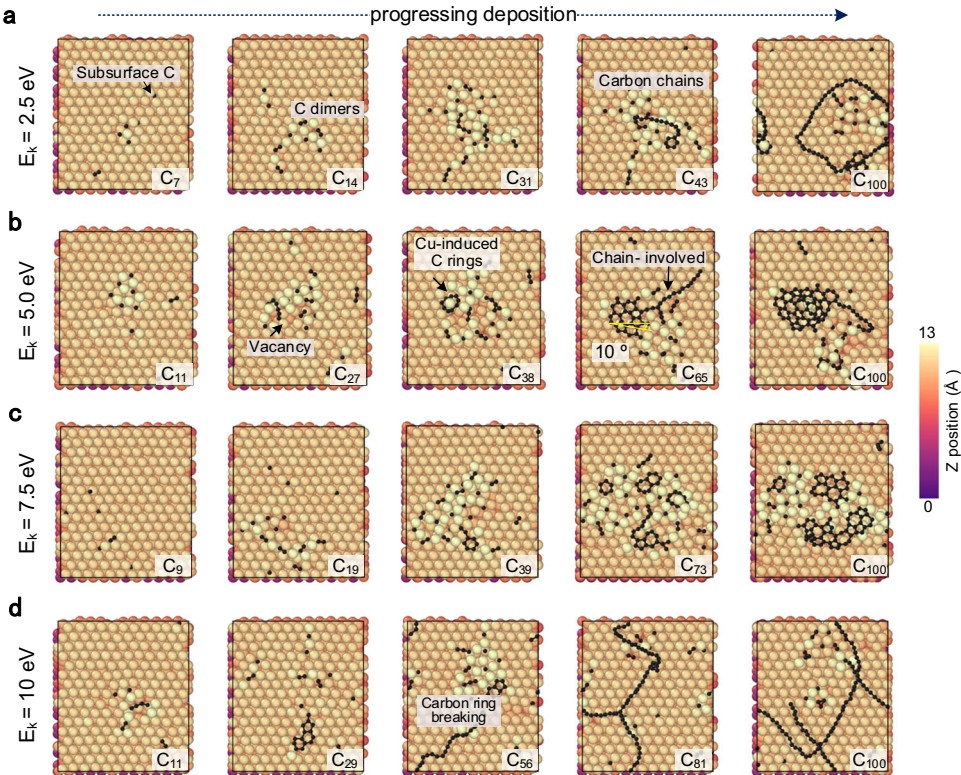

**Fig. 2 | CGM-MLP driven simulations of graphene growth on Cu(111) with different carbon incident kinetic energies ($E_k$).** **a** 2.5 eV, **b** 5.0 eV, **c** 7.5 eV, and **d** 10 eV. Carbon atoms are colored black, and copper atoms are color-coded according to their height coordination. For the labels $C_X$, $X$ denotes the number of deposited carbon atoms. Different graphitization degrees can be observed when the carbon incident energy varies during the simulations. The fully dynamical simulations have correctly reproduced many subprocesses, such as the spontaneous diffusion of carbon monomers and dimers[15], the stabilization effect of adsorbed Cu atoms for carbon rings, the energetically favorable property of carbon chains[18], and the carbon ring breaking process[18], achieving excellent agreements with previous DFT studies[19]. To ensure reproducibility, a repeated run is available in Supplementary Fig. 12.

2.5 eV, it is more probable for carbon chains to grow longer and detach from the Cu(111) surface, rather than forming carbon rings on Cu(111) (Fig. 2a, $C_{100}$). An explanation for the preferred form of carbon chains has been furnished by a previous DFT study[18], which compared the stability of carbon nanoarches and compact carbon nanoislands of equal sizes. Their calculations revealed that the formation of carbon chains was energetically favorable if there were fewer nucleation sites, such as adsorbed Cu atoms and Cu steps. The fully dynamic simulations have reproduced the formation of carbon chains (Fig. 2d) and shown that different carbon chains could share one binding site, as shown in Fig. 2d, $C_{81}$.

## Carbon ring and graphene island formation

For a moderate irradiation energy, i.e., 5 eV or 7.5 eV, hexagonal carbon rings can be observed at the initial stage of the deposition simulations, as shown in Fig. 2b, $C_{38}$ and Fig. 2c, $C_{39}$. Due to the more vigorous bombardment of carbon atoms, the number of off-surface Cu atoms may increase. The presence of adsorbed copper atoms or Cu-C bridges surrounding the initial hexagonal carbon rings has been previously observed in STM images[52]. This observation aligns well with experimental findings[53] that the initial carbon rings tend to nucleate at surface steps or impurities on Cu(111). Furthermore, it is consistent with experiments showing that well-ordered graphene structures only form above 790 °C, attributed to the motion of Cu steps induced by sublimation during growth[22]. Our research has also demonstrated, through dynamic simulations (the animation is available in Supplementary Movie 1), the significance of surface Cu steps in graphene growth. We also show that the critical incident energy of creating a graphene nucleation site on a flat Cu(111) surface may be within 5–7.5 eV.

Our simulations also reveal that the thermodynamics and kinetics analyses of surface growth of graphene on Cu(111) should take into account the involvement of adsorbed Cu atoms, such as C-Cu bridges[15], Cu atoms passivating graphene edges, and edged carbon chains (Fig. 2b, $C_{65}$). In addition, the preferred growth orientation of graphene growth was also reproduced by our simulation model. As shown in Fig. 2b, $C_{65}$, we measured the growth orientation of the final graphene island by using the same benchmark angle proposed by Ding and co-workers[19], and the angle was measured as about 10°, at which the potential energy of graphene rotating on the Cu(111) surface reached a minimum[19]. As highlighted by researchers[25,54], traditional KMC simulations on static metal surfaces may overlook crucial reaction processes, and AIMD simulations often have limited timescales to capture the complete reaction pathway involving Cu atoms. Previous COMB3 and ReaxFF potentials were unable to correctly simulate the growth of graphene on Cu(111) (see Supplementary Discussion. 3, Supplementary Fig. 9, and Supplementary Fig. 10). This is precisely why the CGM-MLP-based MD/tfMC method emerges as the most promising and efficient approach to elucidate the growth mechanism of carbon on metallic surfaces.

## Carbon ring breaking

To gain atomistic insight into the observed suppression of graphene island formation at an incident energy of 10 eV in Fig. 2d, we performed further analyses including the evolution of hexagonal carbon rings and hybridization analysis on the carbon atoms deposited on Cu(111) surfaces, as shown in Fig. 3a. At an incident energy of 2.5 eV, the deposited carbon atoms primarily form carbon chains, resulting in a higher proportion of sp-hybridized carbon compared to sp²- or sp³-carbon. With increasing incident energy, the content of sp²-

**a**

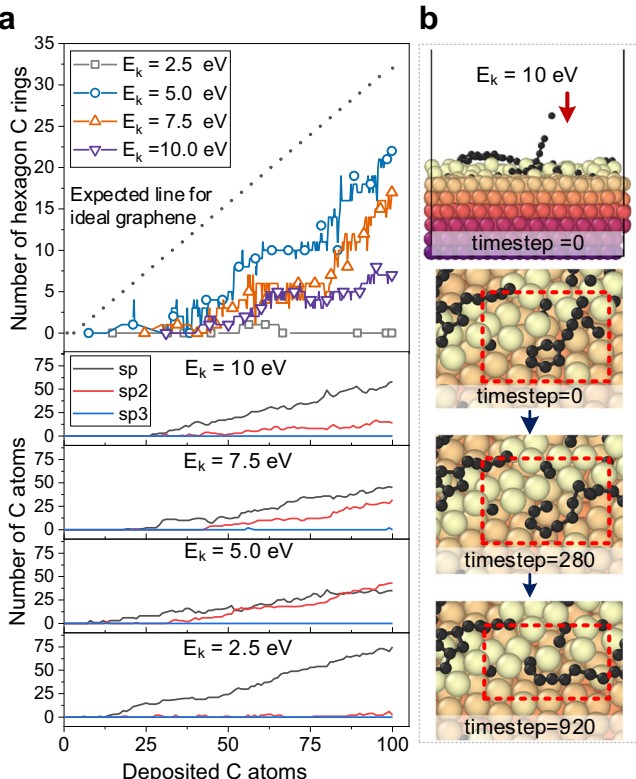

**b**

**Fig. 3 | Carbon structure analysis and observation of carbon ring breakage by high-energy bombardments. a** Evolution of hexagonal carbon rings and hybridization analysis with increasing deposited carbon atoms. Coordination numbers with a 1.8 Å cutoff radius for the nearest neighbors were used to determine the hybridization orbitals. Source data are provided as a Source Data file. $E_k$ is the incident kinetic energy of carbon atoms. The maximum number of hexagonal carbon rings and optimal $sp^2$-content are achieved at a moderate incident energy of 5 eV. Lower or higher incident energies hinder graphene formation due to insufficient adsorbed Cu atoms or carbon ring breakage, respectively. **b** Our simulations capture the transformation from carbon rings to chains, as highlighted by the dashed red boxes.

carbon initially increases and then decreases, indicating that high-energy bombardments can induce more nucleation sites for stabling carbon rings. Previous DFT studies[18] also indicated that without the adsorbed copper atoms to stabilize the hexagonal carbon rings, they would change to linear chains easily. To provide direct evidence of the process, Fig. 3b presents the transformation from carbon ring fragmentation into carbon chains when subjected to high-energy bombardments. This transformation begins with a carbon ring on a Cu(111) surface being subjected to a 10 eV carbon atom bombardment (timestep 0). Subsequent images illustrate the gradual dissociation of carbon rings and the formation of carbon chains resulting from the impact of high-energy carbon atoms. This observation aligns with the findings in Fig. 3a, suggesting that high-energy bombardment impedes the formation of carbon rings and, consequently, the nucleation of graphene. Additionally, we have included an animated demonstration in Supplementary Movie 2, showcasing the powerful capability of the MLP-based MD/tfMC method in capturing dynamic atomistic processes.

The CGM-MPL enables us to perform a fully dynamic simulation of graphene growth on the metal surface and then allows us to comprehensively understand possible growth mechanisms on a dynamical catalytic surface instead of a stagnant catalytic surface. Based on the growth animations obtained from our simulations, some new insights on how the copper atoms get involved in the growth of graphene on

Cu(111) surfaces can be provided, offering possible reaction paths for the following energetic analysis.

In Fig. 4, we proposed a growth model involving the passivation of Cu atoms to the graphene edge and employed the DFT-based, or CGM-MPL-based climbing-image nudged elastic band (CI-NEB) method to demonstrate the accuracy of the machine-learning potentials in thermodynamically or kinetically. As shown in Fig. 4a, the C monomer prefers to remain at an octahedral site in the sub-surface layer. Compared to C adatom on the surface, the DFT and CGM-MPL predict an increase in the binding energy of about 0.5 eV[15]. Then, combining two monomers, i.e., forming a carbon dimer (C2-a – C2-c), overcomes a small energy barrier of only 0.06 eV atom$^{-1}$ (0.12 eV in total) and releases 1.15 eV atom$^{-1}$. These energies obtained from the CGM-MPL are in excellent agreement with those from DFT calculations[15]. Next, in Fig. 4b, we studied the two possible reaction paths from four carbon dimers to a C8 chain or a C8 ring. There is less than 0.1 eV atom$^{-1}$ energy barrier (0.7 eV in total) during the carbon-chain reaction process, which releases an energy of 0.5 eV atom$^{-1}$ obtained from the DFT calculation. In the ring-reaction path, there is a larger energy barrier compared with the carbon-chain path, i.e., approximately 0.2 eV atom$^{-1}$ (1.5 eV in total), and this ring-forming reaction releases 0.25 eV atom$^{-1}$ obtained from DFT. From a kinetic standpoint, carbon-chain formation is more favorable than carbon-ring formation, as the energy barrier for carbon-ring formation is approximately twice as high. From a thermodynamic perspective, the carbon-chain reaction process releases an energy of 0.5 eV atom$^{-1}$ while the ring-forming reaction releases 0.25 eV atom$^{-1}$. The higher energy release in the carbon-chain reaction suggests greater stability and a lower energy state of the chain configuration. Therefore, the spontaneous formation of carbon rings on a flat Cu(111) surface is rarely observed due to the unfavorable thermodynamics and kinetics of graphitic ring formation.

To stabilize the hexagonal carbon ring, transforming surface copper atoms to adsorbed copper atoms is suggested to be necessary during the growth of graphene on the Cu(111) surface. At high deposition temperatures or under energetic carbon bombardments, the surface Cu atoms can quickly move to an adatom site, offering the initial nucleation site for hexagonal carbon rings. Furthermore, it is observed that the binding energy of initial graphene clusters increases with the addition of passivated Cu atoms (Supplementary Discussion 2 and Supplementary Fig. 7). Therefore, the growth of graphene on a flat Cu(111) is supposed to start from the simple structure shown in Fig. 4c, I. Firstly, the deposited carbon atoms migrate to the other side of adsorbed Cu atoms, which offer a nucleation site for graphene and form the metal bridge[52]. Then, the following snapshots from Fig. 4c, II to IV, show a repeatable cycle of incorporating three carbon atoms onto pristine graphene edges. It can be seen in Fig. 4c that the whole process only has a small energy barrier of less than 0.95 eV atom$^{-1}$ (2.82 eV in total), which is much less than the energy barrier without the incorporation of the Cu atoms (2.47 eV atom$^{-1}$)[20]. In Supplementary Fig. 8, to ensure a transparent and comprehensive error assessment, we also provided a comparison of energy barriers as calculated by DFT and CGM-MLP.

The differences between the potential energies along the reaction coordinates calculated by CGM-MLP (solid lines) and DFT (dashed lines) are generally below 0.2 eV atom$^{-1}$. This level of accuracy represents a significant improvement compared to classical empirical potentials. Although there are inherent limitations of CGM-MLP, such as energy errors due to cutoff radius and the complexity of carbon interactions, the potential reaction pathways can be further identified and refined by the subsequent DFT-NEB calculations. In addition, Fig. 4c demonstrates the effect of enhancing the sampling rate $N_f$ on the inclusion of more transition states in the training set, resulting in energy barriers that approach DFT calculations.

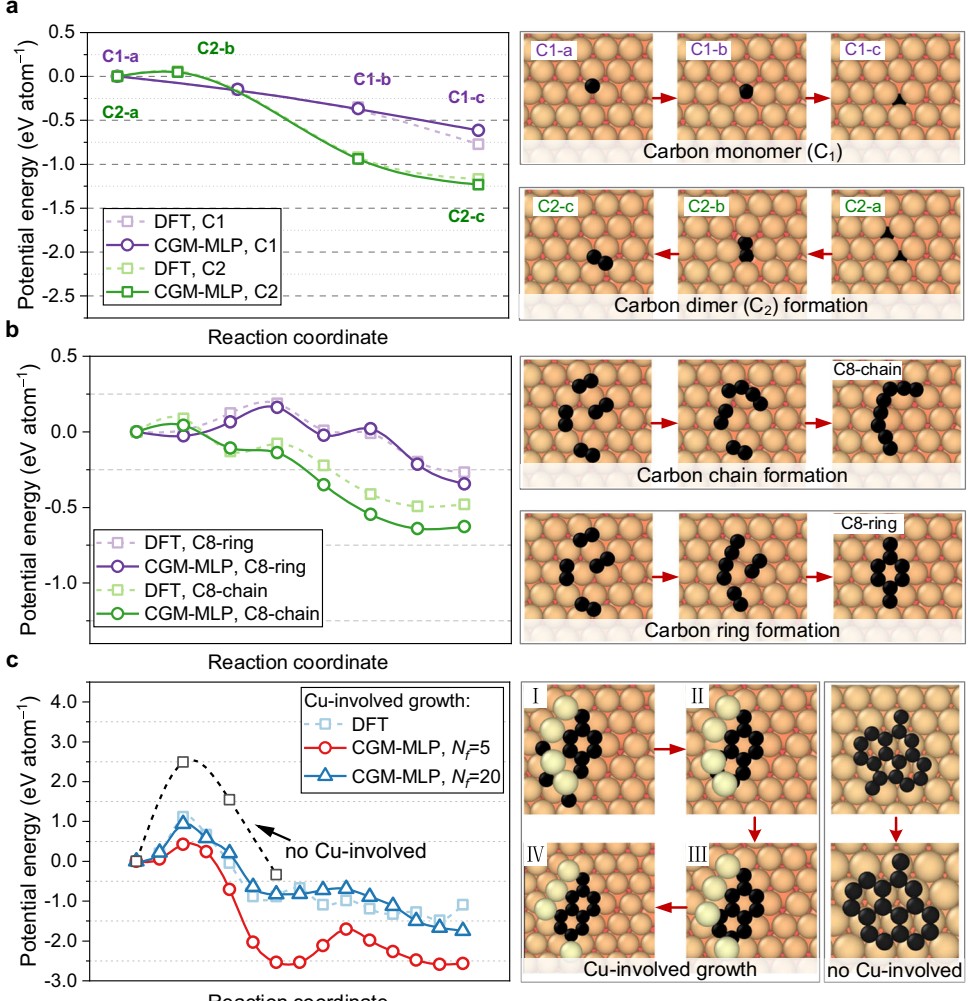

**Fig. 4 | Minimum energy paths of carbon diffusion and graphene nucleation obtained using carbon-growth-on-metal machine-learning potential (CGM-MLP) and DFT-based climbing-image nudged elastic band (CI-NEB) calculations. a** C monomer and dimer diffusion. $C_X$-a/b/c indicates the correspondence between the energy points on the left side and the structures on the right side. **b** Conversion from dimers to carbon chains or rings, and **c** Graphene growth with passivated edges by Cu atoms. Carbon atoms (black), Cu atoms (orange), and adsorbed Cu atoms (wheat). Incorporating edge-passivated Cu atoms reduces the energy barrier for graphene growth (0.95 eV atom$^{-1}$) compared to pristine edges (2.47 eV atom$^{-1}$, the dashed line)[20]. $N_f$ is the number of structures sampled for each deposited carbon atom. Source data are provided as a Source Data file.

To demonstrate the transferability of our CGM-MLP training framework, we extended our study by training additional MLPs, namely Cr-C and Ti-C MLPs, as well as considering the C-Cu system with surface oxygen contamination. Using these CGM-MLPs, we conducted simulations of carbon deposition on metallic surfaces, as depicted in Fig. 5. Our simulation results reveal that compared to carbon growth on Cu(111), fewer carbon rings are observed on the Cr(110) surfaces, and almost no carbon rings are observed on Ti(001). To validate our simulations, we employed a magnetron sputtering system to deposit approximately 30 nm of carbon film on Cu, Cr, and Ti surfaces. High-resolution transmission electron microscopy (HRTEM) images and selected area electron diffraction (SAED) patterns in Fig. 5 show that the carbon film deposited on Cu(111) exhibits the highest crystalline degree, followed by the Cr(110) surface, while the catalytic effect on the Ti(001) surface is the weakest, matching well with the simulations. It should be noted that the diffraction rings observed in the HRTEM and SAED images may arise from both the metal substrate and the carbon films. However, the metallic layers used in this study are quite thin, with an approximate thickness of 10 nm. Moreover, based on the surface morphologies, it is believed

that the catalytic effect on the formation of nanocrystalline carbon follows the order of Cu, Cr, and Ti. Generally, under the same deposition temperature and energy, the crystalline degree of carbon films mainly depends on the initial nucleation rate and the metal-carbon interface. Our simulation results demonstrate that the initial nucleation rate (Fig. 5d) of these three metals follows the order of Cu, Cr, and Ti, which is consistent with experimental observations and previous DFT calculations[6].

Using the Cu-C-O MLP-based MD/tfMC method, we performed deposition simulations of carbon on O-contaminated Cu(111). Detailed growth processes and discussions are available in Supplementary Fig. 11. After depositing 100 carbon atoms, the absence of oxygen contamination led to the formation of small graphene islands on Cu(111) (Fig. 5a). However, in the presence of oxygen, the final structure consisted of cross-linked carbon chains (Fig. 5e inset). Therefore, the observed C-Cu-O bridges and terminated oxygen atoms (Supplementary Fig. 11) significantly reduced the initial nucleation rate of graphene on Cu(111), which is in agreement with previous experimental observations[55]. More discussions can be found in Supplementary Discussion 4.

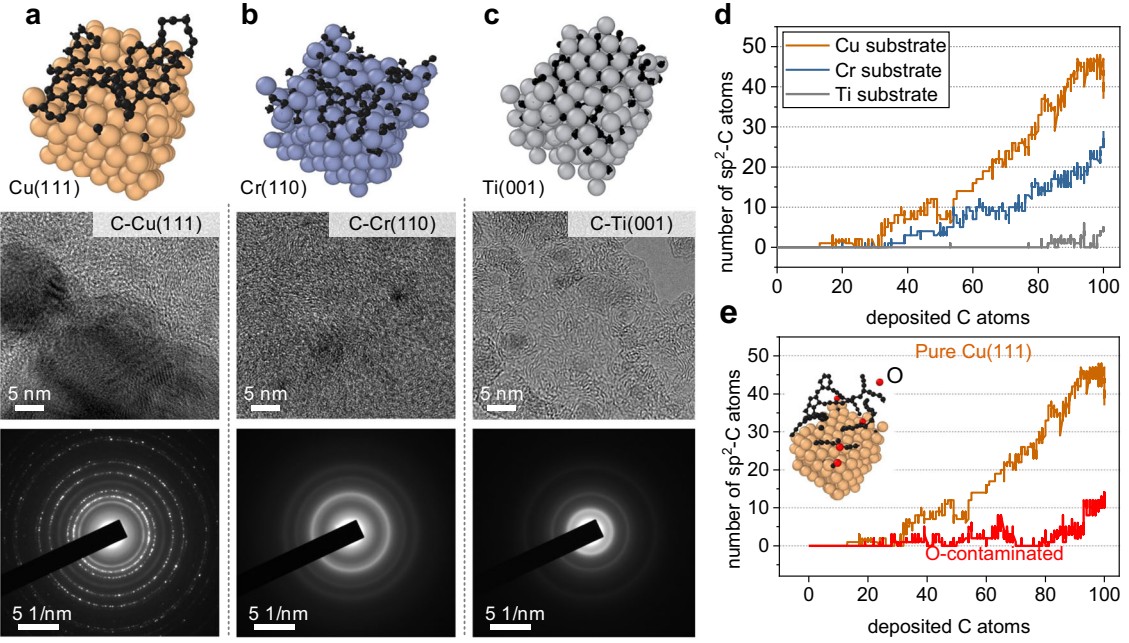

**Fig. 5 | Representative metallic surfaces for the growth of carbon nanostructures. a** pure Cu(111), **b** Cr(110), and **c** Ti(001) surface. Below each surface, High-resolution transmission electron microscopy (HRTEM) images and selected area electron diffraction (SAED) images of carbon nanostructures prepared by magnetron sputtering deposition are provided. **d** The number of sp²-C as a function of deposited carbon atoms on different metal substrates and **e** O-contaminated Cu(111). Source data are provided as a Source Data file.

## Discussion

In summary, this study represents a pioneering advancement in integrating MLPs and MD/tfMC, offering a transferable and efficient strategy for designing metallic or alloy substrates to achieve desired carbon nanostructures. The CGM-MLPs effectively combine the accuracy of first-principles methods with the efficiency of classical force fields. Moreover, the tfMC method overcomes the timescale limitations of traditional AIMD or classic MD approaches. Among various MC methods, tfMC stands out as a trajectory-quasi approach for extending the timescale in simulating off-lattice structures. In addition, the automatic training framework of CGM-MLPs incorporates a specialized query strategy for constructing an on-the-fly training set in deposition simulations, emphasizing the significance of considering the local environment surrounding the deposited atoms.

These developments have enabled the straightforward theoretical study of carbon growth mechanisms on complex metal surfaces. We demonstrate the efficacy of our model through its application to the well-established system of graphene growth on a Cu(111) surface, showcasing high accuracy compared to previous static DFT predictions and experimental observations. Our simulations reproduce the microscopic physical processes correctly and efficiently providing a wide range of possible reaction pathways. Several critical processes and physical insights into how Cu adatoms stabilize graphene nucleation and how graphene grows with its edge passivated by Cu atoms or carbon chains are revealed. DFT-based CI-NEB calculations validate the model's high accuracy in thermodynamics and kinetics, presenting low-energy-barrier processes for graphene growth on Cu(111). Additionally, to demonstrate transferability, we expand our investigation by training additional carbon-metal MLPs, specifically Cr-C and Ti-C MLPs, while also considering the C-Cu system with surface oxygen contamination. HRTEM and SEAD observations confirm the simulated catalytic effect on nucleation and the formation of nano-crystalline carbon on Cu, Cr, and Ti. The machine-learning-driven deposition models presented in this study may open up opportunities for investigating multi-element metallic or alloy substrates in the growth of diverse carbon nanostructures, such as graphene, CNTs, graphite- or diamond-like carbon films.

## Methods

### Hybrid molecular dynamics and time-stamped force-biased Monte Carlo methods

The hybrid MD/tfMC simulations were implemented in the large-scale atomic/molecular massively parallel simulator (LAMMPS)[56] with a plugin of the Quantum Mechanics and Interatomic Potentials (QUIP) software package, which can be found at https://github.com/libAtoms/QUIP. Carbon atoms were deposited along the -Z direction on the metal substrate. Periodic boundary conditions were applied to the X- and Y- directions. To avoid energy recycling through the periodic boundaries, a 2 Å thick velocity rescaling wall with a lateral displacement greater than 6 Å from the initial position of each incident atom was applied to the substrate[51] (see Supplementary Fig. 1 for more details). The deposition simulations consisted of a cycle of MD and tfMC methods. Carbon atoms with varying kinetic energies were deposited during the MD simulations, followed by a 2 ps NVT MD simulation with a timestep of 0.5 fs. The 2 ps duration was sufficient to reach equilibrium, as evidenced by the temperature and total energy of the system remaining constant over an observable time.

During tfMC simulations, in each step, all atoms in the selected group are displaced using the stochastic tfMC algorithm[46]. However, the tfMC method differs from MD algorithms by employing a force-bias probabilistic description of atomic motion (see Supplementary Method 1). Two critical parameters in tfMC are the temperature $T$ and the maximum allowed displacement $\Delta$. The parameter $\Delta$ is determined through a series of simulation tests at different deposition temperatures, following the criterion proposed by Timonova et al.[46]. This criterion ensures that a perfect crystal remains perfect after tfMC simulation and a short MD equilibration. The parameter values $\Delta = 0.18$ Å and $T = 573$ K were carefully determined and utilized in our previous work[27] and were also employed in this study.

## Active machine-learning-driven modeling of carbon growth on the metal surface

The MLP training starts from a training set including numerous carbon allotropes from an accurate and transferable machine-learning potential for carbon[34] and simple carbon clusters ($C_1$-$C_{18}$) on Cu(111) surface. The GAP fitting in this work also uses a 2-body (2b), 3-body (3b) SOAP descriptor to describe the nature of bonding interactions for carbon and copper atoms. A GAP-fitted 2b component with a cutoff of 3.7 Å is used as the 2-body part of our model. A 3b descriptor and the SOAP descriptor represent the many-body contributions to the potential energy. The 3b term is a symmetrized transformation of the Cartesian coordinates of triplets of atoms with a cutoff of 2.5 Å. In the construction of the SOAP descriptor, an expansion of the neighbor density up to $l_{max} = 4$, $n_{max} = 12$, a cutoff of 3.7 Å, $\sigma_{force} = 0.01$ eV Å$^{-1}$, $\sigma_{energy} = 0.001$ eV, and $\zeta = 4$, which are systematically tested for the optimization of the GAP model previously[34] and full details of hyperparameters can also be found in Supplementary Method 3 and Supplementary Fig. 2.

Throughout the training procedure, we introduce two SOAP-based screening parameters, $D^{max}$ and $D^{ave}$, specifically designed to optimize the training set. Their respective equations are presented below:

$$D^{ave} = \operatorname*{ave}_{i < N_1} \left( \min_{j < N_0}(|\vec{A}(a_i), \vec{A}(a_j)|) \right) \tag{1}$$

$$D^{max} = \operatorname*{max}_{i < N_1} \left( \min_{j < N_0}(|\vec{A}(a_i), \vec{A}(a_j)|) \right) \tag{2}$$

where $\mathbf{A}(a_i)$ and $\mathbf{A}(a_j)$ represent the SOAP vectors of the atom $i$ and $j$ position ($a_i$ and $a_j$), respectively. The $\| \|$ denotes the Euclidean distance between the two SOAP vectors. The source code for computing $D^{max}$ and $D^{ave}$ is available at https://github.com/sjtudizhang/CGM-MLP/Calculate_similarity.py[57]. First, the environment fingerprint of each particle is encoded using the SOAP descriptor. The Euler distance between SOAP vectors is calculated as a similarity measure between atoms from different structures. The minimum Euler distance between the particles in one structure and all particles in the other structure is determined to define the similarity measure between the two structures. Average ($D^{ave}$) and maximum ($D^{max}$) values of this minimum Euler distance are calculated for different atoms. Configurations are selected based on the condition that both $D^{ave}$ and $D^{max}$ with previously selected configurations exceed the screen parameters ($S^{ave}$ and $S^{max}$). Detailed tests of the screen parameters can be found in Supplementary Fig. 3 and Supplementary Fig. 4.

## DFT and CI-NEB calculations

All the energies and forces of the structures in the training database are calculated by the QUICKSTEP scheme in CP2K[58,59]. Electronic wave functions were described at the Gamma point using a mixed-basis scheme with Goedecker–Teter–Hutter (GTH) pseudopotentials[60] and a cutoff and the relative cutoff energy of 300 Ry and 60 Ry. Shorter range Double-ζ quality basis sets optimized for GTH Perdew–Burke–Ernzerhof (PBE)[61] pseudopotential were used. Dispersion corrections are included by the well-established Grimme's D3 method[62]. For CI-NEB calculations, we used a total of 4 or 8 replica geometries along the path with a spring constant of 0.001 to restrain the replicas. Other details can be found in Supplementary Method 2.

## Carbon film preparation and characterizations

Carbon films were prepared using a Miba-Teer UDP850 closed-field unbalanced magnetron sputtering system. Metallic seed layers and carbon films were directly deposited onto TEM Cu grids for TEM analysis. The vacuum chamber was evacuated to less than $4.0 \times 10^{-3}$ Pa. Cu, Cr, or Ti seed layers were deposited on Cu grids using a power density of approximately 2.2 W cm$^{-2}$ for 2 min. Carbon films were then deposited on the metallic seed layers using a power density of 3.6 W cm$^{-2}$ on two graphite targets for 5 min. The carbon films had a thickness of approximately 30 nm. The deposition temperature was maintained at 300 °C, and substrate bias voltages were disabled to avoid ion bombardments. HRTEM images were obtained using a JEOL 2100 F device at an acceleration voltage of 200 kV.

## Reporting summary

Further information on research design is available in the Nature Portfolio Reporting Summary linked to this article.

## Data availability

All of the final training datasets are available at https://github.com/sjtudizhang/CGM-MLP[57]. Source data and molecular dynamics trajectories generated in this study have been deposited in the figshare database[63] under accession code: https://doi.org/10.6084/m9.figshare.24591774.v2.

## Code availability

The training code for CGM-MLP is available from the website https://libatoms.github.io/GAP/. The GAP fitting Shell script, Python code for calculating the structural similarity, and C++ code for carbon ring analyses are available at https://github.com/sjtudizhang/CGM-MLP[57].

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

## Acknowledgements
This work was supported by the National Natural Science Foundation of China (No. 22309109 and No. 52225504) (D.Z. and L.F.P.). It was also supported by Research Projects of the National Key Laboratory (No. JCKYS2022603C009) (P.Y.Y., L.F.P., X.M.L.) and the State Key Laboratory of Mechanical System and Vibration (No. MSVZD202009) (P.Y.Y., L.F.P., X.M.L.). D.Z. acknowledges the support of the China National Postdoctoral Program for Innovative Talents from the China Postdoctoral Science Foundation (BX2021178) (D.Z.). The computations in this paper were run on the Siyuan-1 cluster supported by the Center for High-Performance Computing at Shanghai Jiao Tong University. H.L. acknowledges the support received from JSPS KAKENHI (Grant No. JP23K13703) (H.L.) and the Iwatani Naoji Foundation. Additionally, the Center for Computational Materials Science at the Institute for Materials Research, Tohoku University, is acknowledged for providing access to MASAMUNE-IMR (Project No. 202212-SCKXX-0204) (H.L.). The Institute for Solid State Physics (ISSP) at the University of Tokyo is acknowledged for granting access to their supercomputers.

## Author contributions
D.Z. conceived the idea and performed all the simulations and analyses. P.Y.Y., X.M.L., L.F.P., and H.L. supervised the research. All authors contributed to the manuscript before submission.

## Competing interests
The authors declare no competing interests.
