## [Peer Review File · Nature Communications]

Reviewers' Comments:

Reviewer #1:

Remarks to the Author:

In the manuscript "Active Machine Learning Model for the Dynamic Simulation and Growth Mechanisms of Carbon on Metal Surface", the authors build a Gaussian Approximation Potential for carbon on copper using an active learning method that selects training structures based on a SOAP-based distance measure. The use of Monte Carlo simulation to accelerate sampling is a nice addition to the emerging suite of active-learning based tools for building ML potentials. The authors use their potential to perform interesting simulations of carbon deposition on copper that are in line with existing DFT and experimental literature and support the thesis that adsorbed copper is key to graphene formation. I recommend the manuscript for publication but have a few concerns that I think should be addressed.

Major concerns:

1. The authors should give more information about their training protocol. Are four carbon atoms deposited in each round of training, or are four carbon atoms deposited total (line 106)? Is this deposition performed for each of the initial C1-C18 structures? How many structures are selected in each round of training, and how many total rounds of training are required to achieve convergence? This information would help readers compare the authors' method with existing active-learning methods.

2. I found several of the claims in the "Carbon Ring Breaking" section beginning on line 229 confusing:

2a. The authors' results (Fig. 4) suggest that high-energy bombardment inhibits carbon ring formation, and as a possible mechanism for this they state that carbon rings can easily change to chains in the absence of adsorbed copper atoms (line 238-240). Wouldn't we expect high-energy bombardment to create more adsorbed copper atoms, not fewer? The authors seem to suggest this earlier in the paper: "Due to the more vigorous bombardment of carbon atoms, the number of off-surface Cu atoms increases" (lines 200-201).

2b. The authors state "the formation of graphitic rings is neither thermodynamically nor kinetically favorable on a flat Cu(111) surface" (lines 272-273). However, the energy barrier they report for ring formation, 0.25 eV/atom, is only slightly higher than the barrier for chain formation, 0.2 eV/atom. The energy released does not seem dramatically lower (0.62 eV/atom vs. 0.43 eV/atom with CGM-MPL). This suggests that ring formation should be less prevalent but not entirely absent.

2c. The authors claim a large energy barrier of 2.47 eV in the absence of Cu atoms (line 287). This claim is key to the authors' thesis that Cu adsorption facilitates graphene formation. The authors should therefore include this barrier in Fig. 1c and report it in eV/atom for consistency.

2d. Fig. 5 would be clearer if per-atom potential energies were plotted instead of total potential energies to match the descriptions in the text.

Minor concerns:

1. The discussion of on-the-fly learning techniques in lines 64-68 is inaccurate. Recent work in this area, including Refs. 36 and 38 cited by the authors, makes use of predictive uncertainties when selecting training structures and shows they are correlated with true error.

2. It is not clear from the text what carbon clusters were included in the initial training set: line 100 states that C1-C18 structures were included, which seems to match Fig. 1a, but line 326 in Methods claims that only C1-C8 structures were included. Which is correct?

3. It is unclear how the "predictive error" discussed in lines 120-128 and plotted in Fig. 1c is defined. From the SI, my best guess is it is the average per-atom energy error on a held-out set of structures that were not selected by the screening protocol. The authors should state the definition

clearly in the text.

4. While I find the authors' results interesting and compelling, they should avoid overstating their results, e.g. when they claim to "perfectly reproduce many key structures that previous static DFT calculations have studied" (line 291).

5. There are some grammatical errors in the text that should be corrected. A few examples:

"The substrate-catalyzed growth" (line 12) should be "Substrate-catalyzed growth".

"Controllable synthesizing carbon nanomaterials" (line 27) should be "Controllable synthesis of carbon nanomaterials".

"While other widely used methods..." (line 52) is not a complete sentence.

"also needs further studies" (line 72) should be "also needs further study".

Reviewer #2:

Remarks to the Author:

In this contribution, Zhang et al. employ a combination of Gaussian Approximation Potential (GAP), hybrid molecular dynamics (MD), and time-stamped Monte Carlo method to conduct fully dynamic simulations of substrate-catalyzed growth of graphene on Cu(111). The simulations reveal crucial subprocesses, including diffusion paths, formation of carbon chains or rings, and carbon bond breaking. The authors deduce the low-barrier reaction paths of graphene nucleation and growth with the incorporation of edge-passivated Cu atoms. The study is conducted systematically with a robust and well-documented methodology, which integrates tailor-made MC simulations to extend the time-scale span of the simulations with efficient and accurate (active/on-the-fly) machine-learned interatomic potentials, while also comparing them to their reference ab initio DFT calculations and the literature reports. The manuscript is presented in a concise and clear style, with proficient use of English language and minimal grammar errors or typos, making it easy to follow. Proper citation and discussion of previous literature work, except for a couple of missing ones (see below), is also evident.

That being said, however I have some concerns regarding the novelty of the results presented in this study. Most of the key atomic-level physical insights into the chemical vapor deposition (CVD)-enabled graphene nucleation and growth mechanism on pure Cu substrates have already been reported in previous theoretical (mainly based on ab initio and reactive force-field simulations) and experimental studies. While the authors provide appropriate citations to previous work, such as refs 15, 18-20, 23, 25, 26, 45-48, and make connections to the main findings, it seems that a large portion (85-90%) of the current findings shown in the Results section is used to validate the new machine-learned potential (CGM-MLP) and the combined MC+ML-MD approach. In addition, I must underline that the new active machine-learned potential set (force-field) developed within this work is very vital in terms of computational methodology and demonstrating a wider application of ML potentials in materials modelling. However, the combination of MC (in different forms, e.g. kMC, SOF-kMC, etc.) and MD simulations has been previously applied to the specific process of graphene growth on a Cu(111) substrate as well as other related substrates, as discussed in a recent review by Momeni et al. (npj Computational Materials 2020). In particular, the work by Qiu et al. (Acc. Chem. Res. 2018) combines kMC simulations with ab initio static calculations and MD simulations to study the kinetics and mechanism of graphene growth on Cu(100) and Cu(111) surfaces, and highlights the importance of edge passivation by Cu atoms for proper graphene nucleation and growth. However, the work by Qiu et al. has not been cited in the manuscript.

Another crucial aspect that seems to have been overlooked in this study is the effect of surface contamination of the Cu substrate on the graphene growth mechanism, particularly the role of oxygen and hydrogen contamination. Presence of surface oxygen groups on Cu substrates due to oxidation in CVD experimental setups, even under ultra-vacuum conditions, has been detailed in previous studies, such as Science 342, 720-723 (2013). Similarly, hydrocarbon molecules, such as methane, commonly used as the carbon source in CVD processes, introduce hydrogen

contamination that should be considered for a more comprehensive analysis of nucleation and growth dynamics. The role of hydrogen contamination in the growth process has been demonstrated in the kMC+DFT study by Qiu et al. (Acc. Chem. Res. 2018).

Considering these major points, I believe the manuscript is not eligible for publication in Nature Communications in the current form. It might be worthwhile to reconsider it if the Authors can adequately (1) clarify the novelty of this current work by directly comparing the new findings with the known facts from previous reports and (2) explicitly address the surface contamination aspect in the simulations.

Based on the major points highlighted, I believe that the manuscript may not meet the eligibility criteria for publication in Nature Communications in its current form. To reconsider its suitability for publication, I recommend that the authors address the following concerns:

1. Novelty: The authors should further clarify the novelty of their work by directly comparing their findings with known facts from previous reports on graphene growth, which have been extensively addressed in recent literature through various atomistic simulation techniques such as KS-DFT, tight-binding, machine-learned and reactive force-fields as well as experiments. Relevant reviews by Bohwmik et al. (iScience, 2022), Dong et al. (J. Phys. Chem. Lett., 2021), and Momeni et al. (npj Computational Materials, 2020) should be considered to accurately position the current work in the context of existing research.

2. Surface contamination: The authors should explicitly address the aspect of surface contamination in their simulations. This is an important factor that can significantly affect the quality of the deposited graphene, and should be appropriately considered and discussed in the manuscript.

Otherwise, I would recommend the consideration of the manuscript for a more sophisticated/technical journal.

Other minor points to consider:

-The authors stated in the paper that 'little is known about dynamic and atomic-level factors that control the quality of graphene, such as nucleation and growth kinetics.' (page 2, line 37-38). Unfortunately, I cannot agree with this statement in view of the ever-growing body of literature addressing the atomistic simulations, see the recent review papers: Bohwmik et al. (iScience 2022), Dong et al. (J. Phys. Chem. Lett. 2021), and Momeni et al. (npj Computational Materials 2020).

- Transferability of machine-learned potentials: The authors suggest that their machine-learned potentials can be applied to explore other substrates and design substrate architectures for growing carbon nanostructures. However, it should be noted that active/on-the-fly machine-learned potentials are typically specific to the system for which they are trained. Therefore, further dedicated studies would be required to investigate the transferability of the current potentials to other substrates, similar to what has been done in this study using reference sets of carbon machine-learned potentials (e.g. GAP20 from Rowe et al, Journal of Chemical Physics, 2020) and considering the additional interactions with the new substrates under investigation.

-Accuracy of energy barriers and reaction energies: The authors claim an 'excellent' agreement between the energy barriers and reaction energies obtained from density functional theory (DFT) and machine-learned potentials (CGM-MPL) for the reaction pathways shown in Fig. 5b and 5c, and Fig. S2. However, a difference of 0.5-1.0 eV between DFT and machine-learned potentials is far off from the chemical accuracy. The authors should discuss these large differences and their potential implications on the interpretation of the kinetic and thermodynamic processes studied, and compare them with values reported for other types of machine-learned potentials based on the GAP model, as demonstrated in various papers by Deringer, Csanyi, and Bartok-Partay.

- A coordination/hybridization (sp/sp²/sp³) analysis on the carbon atoms should be performed to accurately quantify the different carbon allotropes present in the model systems, as done in similar previous studies. This analysis will support statements such as 'The less graphene-like structure may be mainly attributed to breaking carbon rings into 1-dimensional chains.' (l. 237)

- Regarding the use of COMB3 results, Authors claim that 'If one uses the existing empirical potentials to simulate the growth process of carbon growth on metals, the simulation results could appear inconsistent with the experiments either at the initial or final stage of nucleation (Comparative simulation results based on empirical Cu-C potentials are available in the supplementary information).' In contrast, Klaver et al (Carbon, 2015) showed that COMB3 can reproduce the experimental finding to a satisfactory degree. The authors should further comment

on this discrepancy and provide direct comparisons with experimental results (with proper citations) that contradict their COMB3 results, along with relevant discussions in the supporting information.

-The authors should discuss how they control/prevent the formation of thin carbon films beyond graphene, particularly at higher impact energies. This discussion would provide insights into potential limitations and challenges associated with their simulations.

More specific points:

- line 17: Authors state that they present 'the first fully dynamic simulations of graphene growth on Cu(111)'. Unfortunately this statement may not be accurate as similar simulations have been reported by Klaver et al. (Carbon, 2015) using COMB3 reactive interatomic potentials and probably others.
- Line 99: The reference seems to be incorrect, and it is likely that the authors are referring to Rowe et al. (ref 32) instead of ref 35. It is important to double-check the in-text citations and references for accuracy.
- Line 140: The authors should provide more information on how they determine that the training of CGM-MLP is completed. Clarifying the criteria or convergence metrics used for training completion would be helpful.
- Line 174: Figure 5 appears before Figure 3 in the text, and this order should be corrected to match the order of appearance in the text.
- Line 228: The statement 'achieving excellent agreements with previous DFT studies' requires references to support the claim. The authors should provide appropriate references to validate their findings.
- Line 231: Change 'step-to-step' to 'step-by-step' for better clarity and accuracy.
- Line 278: Change 'as' to 'with' to improve the wording and accuracy of the sentence.
- Line 303: 'Gaussian approximation potential' should be written as 'Gaussian Approximation Potential' for correct capitalization and consistency.
- Fig. 2a: The type of energy reported in the plot should be clarified, whether it is the total energy per atom or formation energy, for better understanding.
- Fig. 2b: It is not clear how the authors define and determine the force error shown in Fig. 2b. The authors should provide more details, such as whether the errors are averaged over all atoms (RMSE) or averaged over the trajectory, and how it is directly linked to the number of deposited atoms.
- Fig. 3: The labels 'I', 'II', 'III', etc. used in the caption of Fig. 3 need to be clarified and explained for better comprehension.
- Fig. 4: The inclusion of 'Rings' in the key legend of Fig. 4 may not be necessary. The authors should provide more details on how the rings were analyzed (i.e., method used), and consider including the expected number of rings for achieving ideal graphene for each cluster size as a guide-to-eye.
- Fig. 5: Adding a color code for different elements in Fig. 5 would be helpful for better interpretation of the results.
- Computational Details of the CI-NEB calculations, such as the number of images, convergence criteria, optimization algorithm, etc., are missing in the text as well as in the Supporting Information. The authors should provide these details for transparency and reproducibility.
- The study by Qiu et al. (Acc. Chem. Res., 2018, 51, 3, 728-735), which presents kinetic Monte Carlo simulations combined with first-principles molecular dynamics for studying the kinetics and mechanistic studies of undoped/N-doped graphene growth on Cu(100) and Cu(111) surfaces, and other relevant studies should be cited to provide a comprehensive review of the literature.

Reviewer #3:

Remarks to the Author:

In this study the authors investigate the catalytic growth of carbon nanostructures on Cu(111) by means of atomistic simulations based on a machine learning potential (MLP). They employ an active learning strategy to train the MLP to density functional theory calculations. Using an Monte-Carlo (MC) sampling scheme they simulate the growth process upon carbon atom deposition from the gas phase. In their simulations, the authors observe many processes of the early stages of growth like, carbon monomer diffusion and dimer formation, their nucleation and the subsequent

growth to chains and hexagonal clusters. Additionally, the authors find the Cu surface to be dynamically changing during the growth process. Based on their observations they propose a mechanism highlighting a stabilizing role of Cu adatoms, which can act as nucleation seeds for chains or hexagonal clusters depending on their abundance.

The present studies' concept and the overall simulation strategy are appealing and the active learning is in accord with state-of-the-art MLP protocols. The demonstration of the complex growth of carbon nanostructures on a dynamically changing catalyst surface is scientifically interesting and a great demonstration of the MLP capabilities. Despite of this favorable general assessment, the study reveals profound shortcomings in the training and quality assessment of the MLPs. Many standard procedures to ensure the reliability of MLPs are omitted which shows in the resulting accuracy of the few benchmark cases presented. This is contrasted by a conflictive language praising the produced MLP as "highly accurate" and celebrating the quantitative agreement, while in fact errors are as large as > 1 eV for different processes. Qualitatively, simulation results are likely physical, but quantitatively below the standard commonly reachable for MLPs. This demonstration of a rather sloppy handling of MLP training which leads inferior accuracy compared to other purpose driven MLPs, does not include standard quality control, and is accompanied by a misleading language may in fact hurt the careful standards developed by the MLP community in the last decade. In addition to this major issue, the publication appears in part incomplete since its missing many details necessary to understand/appreciate the conducted work and is accompanied by a partially imprecise and ambiguous language.

The stated issues (which I detail further below) are so severe, that they require a major rework of the MLP training and all simulations that cannot be addressed in a revision and prohibit publication. For this reason, I need to reject this manuscript for publication.

Detailed comments:

- No basic information about the training results is given. I.e. beyond the starting structures – which also don't contain details about the carbon clusters on Cu(111) – it is unclear what model structure was used in the MD/tfMC training simulations, how many configurations were added each iteration, or in total, and how many iterations were conducted. Force/energy correlation plots & learning curves are missing and no final energy, force and stress error is provided. The error metrics are not clearly defined, only a formula is provided in the SI showing a MAE metric for the forces.
- The convergence criteria are unreasonable loose and not adequate for MLPs. An energy error of 0.05 eV/atom and a force error of 0.5 eV/Å are about an order of magnitude higher than what is usually reported. The large final errors are somewhat recognizable in Fig. 2 which shows a non-discussed spread for low-energy structures and a systematic shift in the total energy of > 0.5 eV which needs to be discussed. The y-axis for the force error in Fig. 2 requires magnification to judge the accuracy of the MLP.
- From the text any information of MLP validation beyond the target simulations is omitted. It would be expected that a test set is assembled in parallel to the training set to check for overfitting.
- It appears, that the hyper-parameters and descriptors from the GAP-20 potential were just taken over without adjusting for the inclusion of Cu which may require careful re-validation.
- The screening parameter S has several issues: The testing procedure explained in the SI is unclear, e.g. how distances are computed (between what sets) and how the test and training set are split exactly. The choice of parameter S is also only weakly motivated. It appears it is expected to be transferable since structures of C and Cu allotropes are compared, but the application of S is for the Cu-C system. It needs to be stated (and treated) as an arbitrary and non-transferable parameter. The similarity measure can imply different errors for different training sets. A performance measure for this criterion would be adequate (statistics of what type of structure were added).
- Quality of the MLP is not insured re-checked in production runs (e.g. via further similarity screening).
- The authors do not clarify what a "time stamp" MC scheme is and how it is different from a standard MC simulation; the publication should be self-containing with some brief explanation and not only refer to older publications.
- It is unclear how energy/velocity rescaling walls affect the kinetic energy of impeding C atoms.

- No explanation is provided why high kinetic energy (10 eV) simulations leads to different carbon cluster growth. It is unclear from the data because in the snapshots it looks like same amount of Cu adatoms is produced.
- The soap distance formula and symbols require explanation in Fig 1.
- In SI screening parameter is inconsistently termed TS
- Fig 3 has bad quality, marker I-V are not explained (timesteps?)
- Language is generally imprecise and ambiguous. E.g. often "well known truths" or similar problematic expressions are used for experimental observations.

Dear Reviewers,

Thank you for taking the time to review our paper. Your feedback and suggestions are highly appreciated, and we are grateful for your effort in helping us improve our work. We have carefully considered your comments and suggestions, and we have made the changes to address the issues you have raised to the best of our abilities. We would like to respond to each of your comments in detail.

To Reviewer #1:

In the manuscript “Active Machine Learning Model for the Dynamic Simulation and Growth Mechanisms of Carbon on Metal Surface”, the authors build a Gaussian Approximation Potential for carbon on copper using an active learning method that selects training structures based on a SOAP-based distance measure. The use of Monte Carlo simulation to accelerate sampling is a nice addition to the emerging suite of active-learning based tools for building ML potentials. The authors use their potential to perform interesting simulations of carbon deposition on copper that are in line with existing DFT and experimental literature and support the thesis that adsorbed copper is key to graphene formation. I recommend the manuscript for publication but have a few concerns that I think should be addressed.

Comment 1: The authors should give more information about their training protocol. Are four carbon atoms deposited in each round of training, or are four carbon atoms deposited total (line 106)? Is this deposition performed for each of the initial C1-C18 structures? How many structures are selected in each round of training, and how many total rounds of training are required to achieve convergence? This information would help readers compare the authors’ method with existing active-learning methods.

✓ Thank you for the questions. In our training protocol, four carbon atoms are deposited in

each round of training. At the end of each round, DFT calculations benchmark the energies and forces, and if the error is below the selected thresholds, MD/tfMC continues with the structure from the last round and deposits four more carbon atoms. If the error exceeds the thresholds, the training dataset is updated with newly selected structures, and another training process starts. This information has been clarified in the revised manuscript to provide a better understanding of our methodology, as also shown in **Fig. R1**. The initial training dataset includes GAP-20 carbon structures¹ and all the C₁-C₁₈ structures on the Cu(111) surface. By including these structures, the initial training dataset provides a good starting point for generating a machine learning potential capable of accurately describing Cu-C interactions.

Fig. R1 Schematic illustrations of CGM-MLP generated by active learning on-the-fly during MD/tfMC simulations. (a) The initial training dataset includes representative carbon structures from GAP-20¹ and C₁-C₁₈ carbon clusters on Cu(111) surfaces. (b) The CGM-MLP trained from this dataset is then used in a deposition simulation employing a hybrid MD/tfMC method². (c) A SOAP-based algorithm used to select the most representative structures from the MD/tfMC simulations. The inset formula calculates the

similarity matrix D^{max} and D^{ave} between the existing training dataset and a new structure, where $\vec{A}(a_i)$, $\vec{A}(a_j)$ represent the SOAP vectors of atom i and j , respectively. The $\| \cdot \|$ denotes the Euclidean distance between the two SOAP vectors. (see Supplementary Information for more details). (d) DFT benchmarks the energies and forces, and if the error is below a threshold, MD/tfMC continues. Otherwise, the training dataset is updated with newly selected structures.

- ✓ **Fig. R17.** presents the energy and force errors during a training process, as well as the number of structures in the training dataset. Each iteration includes MLP fitting, MD/tfMC simulations, structure selection, DFT calculations, and error estimation. During the error estimation step, we use an energy criterion of less than 0.05 eV/atom and a force error of less than 0.5 eV/Å to determine whether newly generated structures should be added to the training set or if the deposition of carbon atoms can proceed without further training. The number of structures added to the training set in each iteration depends on the sampling frequency (N_f) and screen parameters (S^{max} and S^{ave}). For instance, with $N_f = 20$, $S^{max} = 0.08$, approximately 40~60 structures will be added to the training set in each cycle.

Fig. R2 Energy and force errors of 20 iterations during a training process. An energy criterion of less than 0.05 eV/atom and a force error of less than 0.5 eV/Å is used to determine whether newly generated structures should be added to the training set or if the deposition of carbon atoms can proceed without further training. With $N_f = 20$, S^{max} , $S^{ave} = 0.08$, approximately 40~60 structures will be added to the training set.

- ✓ The decision to end the iterations depends on the specific research objectives or the desired number of carbon atoms to be deposited on the metal surface. In this study, the number of carbon atoms to be deposited is predetermined as 100 carbon atoms (25 iterations in total) to demonstrate the capability of the CGM-MLP-based MD/tfMC model in capturing the dynamics of the nucleation and initial growth of graphene on metallic surfaces. Additionally, if further growth processes or the simulation of carbon-based films are of interest, the predetermined number of carbon atoms can be increased accordingly. It is worth noting that if the deposited carbon atoms have negligible interactions with the metal substrate, the training process can be terminated at any point because the training dataset already includes well-developed carbon structures.

- ✓ We have included the above details in the revised manuscript and Supplementary Information to provide a comprehensive understanding of our approach.

Line 121–129:

“During the MD/tfMC simulations, four carbon atoms are deposited in each round of training. From these simulations, a subset of structures is generated with a sampling rate of N_f frame per carbon atom to form the initial training dataset for CGM-MLP. A screening process is then employed, using a SOAP-based similarity metric in Fig. 1c, to refine the dataset further. The similarity measure is the heart of the active learning algorithm, determining whether a new structure should be added to the training set and ensuring a balanced representation of different regions of the phase space. In Fig. 1c, the equation defines two similarity measures, namely \mathbf{D}^{\max} and \mathbf{D}^{ave} , which quantify the maximum and average SOAP-based distance between atoms in the new and previously selected structures (see Supplementary Information for more details).”

line 153–162:

“During the error estimation step (Fig.1), we use an energy criterion of less than 0.05 eV/atom and a force error of less than 0.5 eV/Å to determine whether newly generated structures should be added to the training set or if the deposition of carbon atoms can proceed without further training. (see Fig. S5 for more details). The decision to end the iterations depends on the specific research objectives or the desired number of carbon atoms to be deposited on the metal surface. In this study, the number of carbon atoms to be deposited is predetermined as 100 carbon atoms (25 iterations in total) to demonstrate the capability of the CGM-MLP-based MD/tfMC model in capturing the dynamics of the

nucleation and initial growth of graphene on metallic surfaces. Additionally, if further growth processes or the simulation of carbon-based films are of interest, the predetermined number of carbon atoms can be increased accordingly.”

Comment 2: I found several of the claims in the “Carbon Ring Breaking” section beginning on line 229 confusing:

2a. The authors’ results (Fig. 4) suggest that high-energy bombardment inhibits carbon ring formation, and as a possible mechanism for this they state that carbon rings can easily change to chains in the absence of adsorbed copper atoms (line 238-240). Wouldn’t we expect high-energy bombardment to create more adsorbed copper atoms, not fewer? The authors seem to suggest this earlier in the paper: “Due to the more vigorous bombardment of carbon atoms, the number of off-surface Cu atoms increases” (lines 200-201).

✓ Thank you for your comments. The high-energy bombardment has two main effects. Firstly, it increases the energy of copper atoms on the Cu surface, leading to enhanced thermal motion and increased adsorption likelihood on the surface. Secondly, direct high-energy bombardment can cause significant breakage of existing carbon rings, especially the isolated ones, due to their thermal dynamic instability.

Fig. R3 presents direct evidence of carbon ring fragmentation into carbon chains when subjected to high-energy bombardments. At timestep 0, a carbon ring on a Cu(111) surface was subjected to a 10 eV carbon atom bombardment. **Fig. R3b** shows that under the impact of high-energy carbon atoms, the existing carbon rings undergo gradual dissociation, leading to the formation of carbon chains. This phenomenon explains why high-energy bombardment does not facilitate the nucleation of graphene but rather hinders the formation

of carbon rings.

Fig. R3 Carbon structure analysis and observation of carbon ring breakage by high-energy bombardments. (a) Evolution of hexagonal carbon rings and hybridization analysis with increasing deposited carbon atoms. The maximum number of hexagonal carbon rings and optimal sp²-content are achieved at a moderate incident energy of 5.0 eV. Lower or higher incident energies hinder graphene formation due to insufficient adsorbed Cu atoms or carbon ring breakage, respectively. (b) MLP-based MD/tfMC dynamic simulation capturing the transformation from carbon rings to chains. Coordination numbers with a 1.8 Å cutoff radius for the nearest neighbors were used to determine the hybridization orbitals.

To further illustrate the dual effects of high-energy bombardments, we have made

additional statements to Fig. 4 in the main text.

Line 239-257:

“Carbon Ring Breaking. To gain atomistic insight into the observed suppression of graphene island formation at an incident energy of 10 eV in Fig. 2d, we performed further analyses including the evolution of hexagonal carbon rings and hybridization analysis on the carbon atoms deposited on Cu(111) surfaces, as shown in Fig. 3a. At incident energy of 2.5 eV, the deposited carbon atoms primarily form carbon chains, resulting in a higher proportion of sp-hybridized carbon compared to sp²- or sp³-carbon. With increasing incident energy, the content of sp²-carbon initially increases and then decreases, indicating that high-energy bombardments can induce the breaking of carbon rings into carbon chains. Previous DFT studies¹⁸ also indicated that without the adsorbed copper atoms to stabilize the hexagonal carbon rings, they would change to linear chains easily. To provide direct evidence of the process, Fig. 3b presents the transformation from carbon ring fragmentation into carbon chains when subjected to high-energy bombardments. This transformation begins with a carbon ring on a Cu(111) surface being subjected to a 10 eV carbon atom bombardment (timestep 0). Subsequent images illustrate the gradual dissociation of carbon rings and the formation of carbon chains resulting from the impact of high-energy carbon atoms. This observation aligns with the findings in Fig. 3a, suggesting that high-energy bombardment impedes the formation of carbon rings and, consequently, the nucleation of graphene. Additionally, we have included an animated demonstration in the supplementary materials (i.e., Carbon_Ring_Breaking.mov), showcasing the powerful capability of the MLP-based MD/tfMC method in capturing dynamic atomistic processes.”

✓ We have also included an animated demonstration of carbon ring breaking in the Supplementary Materials. Please see the supplementary animation titled "Carbon_Ring_Breaking.mov".

2b. The authors state “the formation of graphitic rings is neither thermodynamically nor kinetically favorable on a flat Cu(111) surface” (lines 272-273). However, the energy barrier they report for ring formation, 0.25 eV/atom, is only slightly higher than the barrier for chain formation, 0.2 eV/atom. The energy released does not seem dramatically lower (0.62 eV/atom vs. 0.43 eV/atom with CGM-MPL). This suggests that ring formation should be less prevalent but not entirely absent.

✓ We apologize for any misunderstanding. According to the DFT calculations in Fig. 4b (*i.e.*, **Fig. R4**), the energy barrier for ring formation is about 1.6 eV (0.2 eV/atom) while for chain formation, the energy barrier is only 0.75 eV (0.1 eV/atom). From a kinetic standpoint, carbon chain formation is more favorable than carbon ring formation, as the energy barrier for carbon ring formation is approximately twice as high.

Fig. R4 Comparison of the formation pathway for a carbon chain and ring on a flat Cu(111) surface from 4 carbon dimers.

From a thermodynamic perspective, the carbon-chain reaction process releases an

energy of 0.5 eV/atom while the ring-forming reaction releases 0.25 eV/atom. The higher energy release in the carbon-chain reaction suggests greater stability and a lower energy state of the chain configuration. Therefore, the formation of carbon chains is thermodynamically more favorable compared to carbon rings.

In our simulations, the spontaneous formation of carbon rings on a flat Cu(111) surface is rarely observed due to the unfavorable thermodynamics and kinetics of graphitic ring formation. However, it is theoretically possible for carbon rings to form on a flat Cu(111) surface. The challenge in observing this phenomenon in practical simulations is primarily because the surface of Cu(111) is not always perfectly flat, and other factors such as surface-adsorbed atoms or defects can influence the growth process, providing the nucleation site for carbon rings.

- ✓ We have already changed all the energy units from eV to eV/atom to avoid misunderstanding. We appreciate your insights and have included this explanation in the revised manuscript to provide a more comprehensive understanding of our findings.

Line 287-294:

“From a kinetic standpoint, carbon chain formation is more favorable than carbon ring formation, as the energy barrier for carbon ring formation is approximately twice as high. From a thermodynamic perspective, the carbon-chain reaction process releases an energy of 0.5 eV/atom while the ring-forming reaction releases 0.25 eV/atom. The higher energy release in the carbon-chain reaction suggests greater stability and a lower energy state of the chain configuration. Therefore, the spontaneous formation of carbon rings on a flat Cu(111) surface is rarely observed due to the unfavorable thermodynamics and kinetics of graphitic ring formation.”

2c: The authors claim a large energy barrier of 2.47 eV in the absence of Cu atoms (line 287).

This claim is key to the authors' thesis that Cu adsorption facilitates graphene formation. The authors should therefore include this barrier in Fig. 5c and report it in eV/atom for consistency.

✓ Thank you for your suggestions. The energy barrier of 2.47 eV represents the required energy for adding a single carbon atom to the edge of graphene in the absence of Cu atoms, as illustrated in **Fig. R5** (the rightmost image). This value was obtained from the previous work conducted by Shu et al.³ and the potential energy obtained from the no Cu-involved reaction pathway is shown as the dashed line in **Fig. R5**. Therefore, the correct unit for this energy barrier is 2.47 eV/atom.

Fig. R5 Comparison of the reaction pathways for the addition of a single carbon atom to the graphene edge, (a) without and (b) with the presence of a Cu atom³.

✓ We have included the dashed line to compare the energy barriers, as shown in Fig. R6.

When edge-passivated Cu atoms are incorporated, the graphene growth process exhibits a lower energy barrier of 0.95 eV/atom, in contrast to the higher energy barrier of 2.47 eV/atom observed for the pristine edge.

Fig. R6 Minimum energy paths of carbon diffusion and graphene nucleation obtained using CGM-MLP and DFT-based CI-NEB calculations. (a) C monomer and dimer diffusion, (b) conversion from dimers to carbon chains or rings, and (c) graphene growth with passivated edges by Cu atoms. Carbon atoms (black), Cu atoms (orange), and adsorbed Cu atoms (wheat). Incorporating edge passivated Cu atoms reduces the energy barrier for

graphene growth (0.95 eV/atom) compared to pristine edges (2.47 eV/atom, the dashed line)³. CGM-MLP-based MD/tfMC captures the diverse growth pathways accurately and efficiently.

2d. Fig. 5 would be clearer if per-atom potential energies were plotted instead of total potential energies to match the descriptions in the text.

- ✓ Thank you for your suggestion. We have considered it and made the necessary revisions to Fig 4 (originally Fig.5). Specifically, we have replaced the potential energies with per-atom potential energies, as depicted in **Fig. R6** (also referred to as Fig. 4 in the main text). This adjustment enhances the accuracy and clarity of the figure, allowing readers to better understand the energy landscape associated with the studied processes.

Minor concerns:

1. The discussion of on-the-fly learning techniques in lines 64-68 is inaccurate. Recent work in this area, including Refs. 36 and 38 cited by the authors, makes use of predictive uncertainties when selecting training structures and shows they are correlated with true error.

- ✓ Thank you for your comments. Several previous studies have explored the integration of predictive uncertainties with selective training principles. These approaches include the posterior methods like Bayesian inference^{4, 5} and prior methods such as spilling factor⁶, Maxvol method⁷, and CUR decomposition⁸. Each of these analytical uncertainty estimators has its advantages and disadvantages:

The Bayesian inference method provides a posterior distribution based on observed first-principles data sets. This approach incorporates both the sampling density in the descriptor space and the shape of the potential energy landscape, allowing for a

comprehensive assessment of uncertainty.⁴ However, this method needs to make assumptions regarding the prior selection of training structures for first-principles calculations. In our simulations, we generate at least 1600 structures in each cycle. Therefore, selecting all of these structures would be impractical due to training efficiency concerns. Conversely, directly choosing only a subset of structures presents challenges in justifying the selection criteria.

For the prior approaches, where the uncertainty is determined only from the descriptor, there is no need for the prior selection of structures. In this study, we refer to the work of Gábor Csányi and colleagues⁸, the prominent contributors to the development of the Gaussian Approximation Potential (GAP) machine learning model.⁹ Their approach, based on the Smooth Overlap of Atomic Positions (SOAP) vectors or kernel matrix, employed a configuration-averaged kernel metric to identify the most diverse structures at each step.

However, directly applying the configuration-averaged metric to the on-the-fly training of deposition simulations may have limited effectiveness. This is because, during the deposition simulation, the coordination environment of most atoms in the substrate may undergo slight changes. Relying only on the configuration-averaged metric for selecting new structures could omit the structures that exhibit significant variations only in the local areas surrounding the deposition atom. To address this limitation, we propose not only using the configuration-averaged similarity metric (referred to as D^{ave} in the main text) but also including the structure with a single atom displaying the most distinct environment (referred to as D^{max}). By adopting this new-structure selection protocol, we achieve enhanced efficiency and effectiveness in the on-the-fly training of deposition processes (**Fig. R1**).

- ✓ We have revised the discussion in the main text to accurately position our work within the academic context.

Line 73-84:

“In the production of the on-the-fly training set, the selection of the most diverse structures is crucial for achieving high efficiency and accuracy¹⁰. So far, several selective principles have been proposed to construct a training set with predictive uncertainties, including both the posterior methods such as Bayesian inference^{4, 5} and prior methods such as spilling factor⁶, Maxvol method⁷, and CUR decomposition⁸. Among the prior methods, the CUR decomposition method stands out for its accuracy and robustness when dealing with carbon-based and metal materials⁸. This method employs a configuration-averaged kernel metric to identify the most different structures. However, relying only on the configuration-averaged metric for selecting new structures during deposition simulation could omit structures that exhibit significant variations only in the local areas surrounding the deposited atom. To enhance the efficiency and effectiveness in the on-the-fly training of deposition processes, a well-defined selection protocol is required.”

2. It is not clear from the text what carbon clusters were included in the initial training set: line 100 states that C1-C18 structures were included, which seems to match Fig. 1a, but line 326 in Methods claims that only C1-C8 structures were included. Which is correct?

- ✓ We are sorry for the typos in line 326. We would like to clarify that the initial training set includes all carbon clusters C₁ to C₁₈. The necessary revisions have been made in the Method section to rectify any errors.

3. It is unclear how the “predictive error” discussed in lines 120-128 and plotted in Fig. 1c is defined. From the SI, my best guess is the average per-atom energy error on a held-out set of structures that were not selected by the screening protocol. The authors should state the definition clearly in the text.

- ✓ Thank you for your questions. In **Table R1**, we have provided a detailed description of how we selected the training and testing datasets to evaluate the predictive error. These datasets were employed to assess the effectiveness of the screening parameters S^{max} and S^{ave} in optimizing different types of structures in the final training dataset. The XYZ configuration files are also available on our GitHub: <https://github.com/sjtudizhang/CGM-MLP>.

Table R1. Generation of the test and training sets used for testing the screen parameters.

Structure Group	Training Set	Testing set	File Name
Graphite	40 graphite structures with different lattice constants ranging from 2.0 to 3.2 Å, with a 0.03 Å increment	20 randomly generated graphite crystal structures with lattice constants ranging from 2.0 to 3.2 Å	Graphite_Training_Set.xyz Graphite_Testing_Set.xyz
Amorphous Carbon	2558 structures selected from the GAP-20 database ¹	493 structures available from the GAP-20 database ¹ , excluding any structures present in the training set.	Amorphous_Carbon_Training_Set.xyz Amorphous_Carbon_Testing_Set.xyz
Carbon-Cluster@Cu	588 structures selected from the AIMD simulation of the Cu(111) slab, including both the C1-C18 clusters on the Cu(111) slab	192 structures were uniformly selected from the AIMD simulation, excluding any structures that are part of the training set.	Carbon_Cluster_Training_Set.xyz Carbon_Cluster_Testing_Set.xyz
Deposited-Carbon@Cu	1090 structures uniformly selected from the MD/tfMC simulation during the training process of CGM-MLPS	468 structures uniformly selected from the MD/tfMC simulation, excluding any structures that are part of the training set.	Deposited_Carbon_Training_Set.xyz Deposited_Carbon_Testing_Set.xyz

Following the creation of the training and test sets, we proceeded to screen the training

sets using various screen parameters (S^{max}) ranging from 0.06 to 0.40, while the S^{ave} was fixed at 0.08. The screened training sets were then employed to train MLPs. Subsequently, these trained MLPs were used to calculate the energy errors for the corresponding test sets. The results of these calculations are presented in **Fig. R7**. It is worth noting that the similarity measure can imply different errors for different training sets. Generally, the more drastic the variations in the potential energy surface, while achieving the same level of predictive accuracy, the smaller the value of S^{max} is required. In different system applications, it is necessary to test various systems based on the precision requirements and ultimately select the appropriate value of S^{max} and S^{ave} . In this study, it was found that when the value of S^{max} is below 0.1, all predictive errors remain below 0.05 eV/atom. Therefore, we selected $S^{max}, S^{ave} = 0.08$ as the parameters to train the CGM-MLP in this work.

Fig. R7 The screen parameter S^{max} and the corresponding predictive accuracies for different structures, with a fixed low S^{ave} value of 0.08.

- ✓ In **Fig. R7**, we also conducted tests to examine the relationship between S^{ave} and predictive accuracy when using only D^{av} to filter the candidate structure dataset. As mentioned before, relying only on the configuration-averaged metric for selecting new structures could omit

structures that exhibit significant variations only in the local areas surrounding the deposition atom. To address this limitation, we propose not only using the configuration-averaged similarity metric (D^{ave}) but also including the structure with a single atom displaying the most distinct environment (D^{max}). In **Fig. R7**, by simultaneously using both D^{ave} and D^{max} as filtering criteria, as compared to solely utilizing D^{ave} , we achieve the same accuracy level with a more relaxed screen parameter.

- ✓ We have replaced the original **Fig. 1c** with a force correlation plot (**Fig. R1c**), as it may better help readers understand the impact of the screen parameter on MLP prediction errors. Additionally, we have included **Fig. R7 (Fig. S4)** in the updated Supplementary Information, along with **Table R1**, to ensure that readers have a comprehensive understanding of the methods we employed to optimize the training dataset.

4. While I find the authors' results interesting and compelling, they should avoid overstating their results, *e.g.* when they claim to “perfectly reproduce many key structures that previous static DFT calculations have studied” (line 291).

- ✓ Thank you for your comments. We have thoroughly reviewed the statements in our paper to ensure that the results are not overstated and have eliminated any repetitive statements, as mentioned earlier.

5. There are some grammatical errors in the text that should be corrected. A few examples:

“The substrate-catalyzed growth” (line 12) should be “Substrate-catalyzed growth”.

“Controllable synthesizing carbon nanomaterials” (line 27) should be “Controllable synthesis of carbon nanomaterials”.

“While other widely used methods...” (line 52) is not a complete sentence.

“also needs further studies” (line 72) should be “also needs further study”.

- ✓ Thank you for your careful reading. We have conducted a careful review of the statements in our paper to minimize grammatical errors.

We would like to express our sincere gratitude to the reviewers for your valuable comments and suggestions. Their insightful feedback has greatly contributed to the improvement of our manuscript. We are truly grateful for their valuable input, which has undoubtedly strengthened the quality and clarity of our manuscript.

To Reviewer #2:

In this contribution, Zhang et al. employ a combination of Gaussian Approximation Potential (GAP), hybrid molecular dynamics (MD), and time-stamped Monte Carlo method to conduct fully dynamic simulations of substrate-catalyzed growth of graphene on Cu(111). The simulations reveal crucial subprocesses, including diffusion paths, formation of carbon chains or rings, and carbon bond breaking. The authors deduce the low-barrier reaction paths of graphene nucleation and growth with the incorporation of edge-passivated Cu atoms. The study is conducted systematically with a robust and well-documented methodology, which integrates tailor-made MC simulations to extend the time-scale span of the simulations with efficient and accurate (active/on-the-fly) machine-learned interatomic potentials, while also comparing them to their reference ab initio DFT calculations and the literature reports. The manuscript is presented in a concise and clear style, with proficient use of English language and minimal grammar errors or typos, making it easy to follow. Proper citation and discussion of previous literature work, except for a couple of missing ones (see below), is also evident.

That being said, however I have some concerns regarding the novelty of the results presented in this study. Most of the key atomic-level physical insights into the chemical vapor deposition (CVD)-enabled graphene nucleation and growth mechanism on pure Cu substrates have already been reported in previous theoretical (mainly based on ab initio and reactive force-field simulations) and experimental studies. While the authors provide appropriate citations to previous work, such as refs 15, 18-20, 23, 25, 26, 45-48, and make connections to the main findings, it seems that a large portion (85-90%) of the current findings shown in the Results section is used to validate the new machine-learned potential (CGM-MLP) and the combined MC+ML-MD approach. In addition, I must

underline that the new active machine-learned potential set (force-field) developed within this work is very vital in terms of computational methodology and demonstrating a wider application of ML potentials in materials modelling. However, the combination of MC (in different forms, e.g. kMC, SOF-kMC, etc.) and MD simulations has been previously applied to the specific process of graphene growth on a Cu(111) substrate as well as other related substrates, as discussed in a recent review by Momeni et al. (npj Computational Materials 2020). In particular, the work by Qiu et al. (Acc. Chem. Res. 2018) combines kMC simulations with ab initio static calculations and MD simulations to study the kinetics and mechanism of graphene growth on Cu(100) and Cu(111) surfaces, and highlights the importance of edge passivation by Cu atoms for proper graphene nucleation and growth. However, the work by Qiu et al. has not been cited in the manuscript.

Another crucial aspect that seems to have been overlooked in this study is the effect of surface contamination of the Cu substrate on the graphene growth mechanism, particularly the role of oxygen and hydrogen contamination. The presence of surface oxygen groups on Cu substrates due to oxidation in CVD experimental setups, even under ultra-vacuum conditions, has been detailed in previous studies, such as Science 342, 720–723 (2013). Similarly, hydrocarbon molecules, such as methane, commonly used as the carbon source in CVD processes, introduce hydrogen contamination that should be considered for a more comprehensive analysis of nucleation and growth dynamics. The role of hydrogen contamination in the growth process has been demonstrated in the kMC+DFT study by Qiu et al. (Acc. Chem. Res. 2018).

Considering these major points, I believe the manuscript is not eligible for publication in Nature Communications in its current form. It might be worthwhile to reconsider it if the Authors can adequately (1) clarify the novelty of this current work by directly comparing the new findings with

the known facts from previous reports and (2) explicitly address the surface contamination aspect in the simulations.

Based on the major points highlighted, I believe that the manuscript may not meet the eligibility criteria for publication in Nature Communications in its current form. To reconsider its suitability for publication, I recommend that the authors address the following concerns:

Comment 1: Novelty: The authors should further clarify the novelty of their work by directly comparing their findings with known facts from previous reports on graphene growth, which have been extensively addressed in recent literature through various atomistic simulation techniques such as KS-DFT, tight-binding, machine-learned and reactive force-fields as well as experiments. Relevant reviews by Bohwmik et al. (iScience, 2022), Dong et al. (J. Phys. Chem. Lett., 2021), and Momeni et al. (npj Computational Materials, 2020) should be considered to accurately position the current work in the context of existing research.

- ✓ Thank you very much for the comments. This study aims to develop an efficient and transferable training framework for an on-the-fly machine learning potential (MLP), specifically for the deposition simulation of carbon growth on a metallic surface. While providing comprehensive insights into a specific metallic surface is not the primary focus, the key contributions of this work can be summarized as follows:

Contribution 1: Combining a machine learning potential (MLP) with the enhanced sampling method—MD/time-stamped force-bias MC methods (MD/tfMC). As listed in **Table R2**, some representative works employed three primary simulation techniques for modeling carbon growth on metallic substrates, namely *ab initio* molecular dynamics (AIMD), DFT-based KMC, and a hybrid MD and force-biased MC using empirical force

fields. However, these techniques encounter different challenges when applied to the atomistic simulation of carbon growth on various metallic surfaces, as depicted in **Fig. R8**.

Table R2. Recent reviews about simulation techniques for carbon deposition

Material	Substrate	Methodology	Interatomic potential	Ref.
methane	Ni(111)	AIMD	DFT	11
methane	Cu(111)	AIMD	DFT	12
graphene	Cu(111)	KMC	DFT	13
graphene	Cu(100)	KMC	DFT	14
graphene	Ir(111)	KMC	DFT	15
graphene	Ni(111)	MD/UFMC	ReaxFF	16
graphite nanocrystals	Amorphous carbon	MD/tfMC	ci-ReaxFF	2
carbon nanotube	Ni cluster	MD/UFMC	ReaxFF	17
graphene	Cu(111) and (100)	Classic MD	COMB3	18
graphene	Metallic surfaces	MD/tfMC	MPLs	this work

As a straightforward and dynamic way, DFT-based MD (AIMD) simulations were applied to investigate how graphene grows on transition metal surfaces from CH₄.^{11, 12} However, due to the short time scale of AIMD, their simulations mainly focused on the decomposition process of CH₄. Therefore, limited insights into the growth process of graphene were revealed, during which the chemical reactions are often with high energy barriers.

Fig. R8 The position of this work within the context of existing research, is determined by evaluating two critical factors: timescale and accuracy. This work takes a pioneering step in integrating MLPs and tfMC, aiming to advance the development of deposition simulation techniques.

Kinetic Monte Carlo (KMC) is widely used for predicting the growth morphology and kinetic mechanisms of 2D materials. However, applying KMC to model the fully dynamic deposition of carbon on metallic surfaces presents significant challenges. One major challenge is to establish a prior event table that lists all possible kinetic events. In real deposition processes, the formation of unpredictable structures like rings, chains, cross-linked clusters, or defected and amorphous carbons further complicates the construction of the event table. Additionally, the movement of substrate atoms during a fully dynamic simulation introduces numerous diffusions or reaction events, making it extremely challenging to predefine the event table. To our best knowledge, there is currently no on-the-fly algorithm available to construct an event table in real-time for amorphous structures. As a result, despite the high accuracy and extended time scale achieved by DFT-based KMC,

it is not suitable for simulating carbon growth on metallic surfaces due to the aforementioned limitations.

Another common method to extend the time scale of AIMD is through MD simulations based on empirical force fields, such as reactive force field (ReaxFF)^{19, 20}, Tersoff potential²¹, reactive empirical bond order (REBO)²², adaptive intermolecular reactive empirical bond order (AIREBO)²³, and charge optimized many-body potential (COMB)^{18, 24}. However, the accuracy of these force fields heavily relies on the systems they were originally fitted for. As a result, the lack of universal functional forms for reactive potentials restricts their transferability to different stages of deposition. For instance, MD simulations of graphene on Cu surfaces have employed the COMB3 potential¹⁸, but these simulations started with pre-constructed graphene sheets on the Cu surface rather than simulating the deposition process from carbon-based precursors. Therefore, it remains uncertain whether this potential is suitable for deposition simulations. In the Supplementary Information of this work, we demonstrate that when the COMB3 potential is applied to the deposition simulation, no graphene is generated on Cu surfaces. Furthermore, developing classical potential energy surfaces for complex reactive systems is challenging due to the extensive requirement of first-principles calculations. These limitations further hinder the computational design of new catalyzed substrates.

In summary, MLPs have the potential of combining the advantages of the accuracy of first-principles methods and the efficiency of the classical force field. Among various Monte Carlo (MC), tfMC stands out as the most effective approach for extending the time scale in simulating off-lattice structures. It offers the advantage of a more realistic trajectory,

capturing the dynamics and transitions of the system in a highly accurate manner.² As a result, this work takes a pioneering step in integrating MLPs and tfMC, aiming to advance the development of deposition simulation techniques. By combining these methodologies, it is anticipated that the accuracy and efficiency of the simulations can be effectively enhanced, paving the avenue for further advancements in this field.

Contribution 2: A specialized query strategy for constructing an on-the-fly training set in deposition simulations, emphasizing the significance of the local environment surrounding the deposited atoms. Each of the existing MLPs has a nontrivial functional form that captures the physical symmetries of interatomic interaction. More importantly, active MLPs should have a query strategy to determine whether a new configuration should be included in the training set. Among various MLP training methods, such as the neural network potentials (NNP)²⁵, Gaussian approximation potential (GAP)⁹, spectral neighbor analysis potential (SNAP), and moment tensor potential (MTP)²⁶, GAP has shown successful applications in the deposition simulation²⁷ of amorphous carbon and studying the surface chemistry²⁸ of carbon nanostructures. Notably, it has been reported that GAP-20 achieves an accuracy of 95% when compared to DFT-calculated energy²⁹, surpassing a majority of empirical potentials, such as Tersoff, mo-Tersoff, AIREBO, mo-AIREBO, LCBOP, EDIP, and ReaxFF₂₀₁₃. Moreover, GAP also exhibits slightly better performance in simulating amorphous or disordered structures compared to NNP³⁰.

Fig. R9. Energy/force correlation plots in production runs by using different quality control parameters, namely N_f , S^{max} , and S^{ave} . (a) Energy and (b) force correlation plots. (c) Production runs by using different S^{max} , with S^{ave} fixed at a low value of 0.08.

As a key similarity descriptor, SOAP has been widely used in GAP fitting and a configuration-averaged kernel metric was previously proposed to do an active machine learning⁸. However, directly applying the configuration-averaged metric to the on-the-fly training of deposition simulations may not be fully effective. This is because, during the deposition simulation, the coordination environment of most atoms in the substrate may have slight changes. Relying only on the configuration-averaged metric for selecting new structures could omit structures that exhibit significant variations only in the local areas surrounding the deposition atom. To address this limitation, we propose not only using the

configuration-averaged similarity metric (referred to as D^{ave} in the main text) but also including the structure with a single atom displaying the most distinct environment (referred to as D^{max}). As shown in **Fig. R9**, by adopting this new query strategy, we achieve enhanced efficiency and accuracy in the on-the-fly training of deposition processes.

Contribution 3. Transferable design strategies for metallic substrate-catalyzed graphene and beyond. This study successfully conducted fully dynamic simulations of carbon deposition on metallic surfaces, incorporating a dynamic catalytic surface and capturing important rare events simultaneously. The dynamic simulations provide a wealth of possible reaction pathways and an overall picture of the growth of various nanostructures. Based on the direct observations, we can infer the underlying growth mechanism and gain new insights for the design and development of carbon-based materials.

To demonstrate this, we extended our study by training additional carbon-metal MLPs, namely Cr-C and Ti-C MLPs, as well as considering the C-Cu system with surface oxygen contamination. Using these CGM-MLPs, we conducted simulations of carbon film deposition on metallic surfaces, as depicted in **Fig. R10**. Our simulation results reveal that compared to carbon growth on Cu(111), fewer carbon rings are observed on the Cr(110) and Ti(001) surfaces, and almost no carbon rings are observed on the latter. To validate our simulations, we employed a magnetron sputtering system to deposit approximately 30 nm of carbon film on Cu, Cr, and Ti surfaces (see Methods in the main text for more experimental details). High-resolution transmission electron microscopy (HRTEM) images and selected area electron diffraction (SAED) patterns in **Fig. R10** show that the carbon film deposited on Cu(111) exhibits the highest crystalline degree, followed by the Cr(110) surface, while

the catalytic effect on the Ti(001) surface is the weakest. It should be noted that the diffraction rings observed in the HRTEM and SAED images may arise from both the metal substrate and the carbon films. However, based on the surface morphologies, it is believed that the catalytic effect on the formation of nanocrystalline carbon follows the order of Cu, Cr, and Ti.

Fig. R10 Representative metallic surfaces for the growth of carbon nanostructures. (a) pure Cu(111), (b) Cr(110) surface, and (c) Ti(001) surface. Below each surface, HRTEM and SAED images of carbon nanostructures prepared by magnetron sputtering deposition (see Methods) are provided. (d) The number of sp²-C as a function of deposited carbon atoms on different metal substrates and (e) O-contaminated Cu(111). By comparing the simulation and experimental results, the CGM-MLP-based MD/tfMC model accurately predicts the catalytic effect of metallic surfaces on the initial nucleation of graphite-like nanocrystalline films.

Generally, under the same deposition temperature and energy, the crystalline degree of carbon films mainly depends on the initial nucleation rate and the metal-carbon interface. Our simulation results demonstrate that the initial nucleation rate (**Fig. R10d**) of these three metals follows the order of Cu, Cr, and Ti, which is consistent with experimental observations and previous DFT calculations³¹. Therefore, this work provides a transferable and efficient strategy for designing metallic or alloy substrates to achieve desired carbon nanostructures or films.

In summary, this work represents a pioneering integration of MLPs and tfMC to advance the development of deposition simulation techniques, providing a practical and efficient approach for designing metallic or alloy substrates to achieve desired carbon nanostructures and explore further reaction possibilities. A specialized query strategy for constructing an on-the-fly training set in deposition simulations was proposed, highlighting the importance of considering the local environment surrounding the deposited atoms. In addition, the simulations and experimental observations involving the deposition of carbon atoms on various metallic substrates demonstrated transferable design strategies for metallic substrate-catalyzed graphene and other related materials.

Comment 2: Surface contamination: The authors should explicitly address the aspect of surface contamination in their simulations. This is an important factor that can significantly affect the quality of the deposited graphene and should be appropriately considered and discussed in the manuscript.

- ✓ Thank you for your valuable comments. We acknowledge that the presence of oxygen or hydrogen contamination can significantly influence the nucleation and epitaxial growth of graphene on Cu(111). To demonstrate the transferability of our CGM-MLP training

framework, we have extended our method to construct a ternary Cu-C-O MLP. Building upon the Cu-C MLP, we initiated the training process using an O-contaminated Cu(111) surface (**Fig. R11a**) as the starting point. The quality control parameters for the training process were consistent with those used for training the Cu-C MLPs. After 25 iterations of training, additional oxygen-containing 1723 structures were added to the training set, which was then employed to fit the Cu-C-O MLP.

Using the Cu-C-O MLP-based MD/tfMC method, we performed deposition simulations of carbon on O-contaminated Cu(111). At the initial stage (**Fig. R11b**), the deposited carbon atoms exhibited a preference for binding with surface oxygen, forming CO molecules. These CO molecules were observed to diffuse rapidly on the Cu(111) surface or escape into the vacuum region. Due to the binding with oxygen, fewer carbon atoms penetrated the sublayer of Cu(111). As the deposition process continued, short carbon chains were observed on the Cu(111) surface, but with the presence of oxygen contamination, the formation of C-Cu-O bridges was also generally evident (**Fig. R11c**). As discussed in the main text, the presence of C-Cu bridges is crucial for the formation of initial carbon rings. However, the C-Cu-O bridges may hinder the combination of carbon chains to form potential carbon rings. Additionally, oxygen atoms were observed to bind with the short carbon chains, acting as terminal groups and further inhibiting the formation of small carbon rings. After depositing 100 carbon atoms, the absence of oxygen contamination led to the formation of small graphene islands on Cu(111) (**Fig. R10a**). However, in the presence of oxygen, the final structure consisted of cross-linked carbon chains (**Fig. R11e**). Therefore, the presence of C-Cu-O bridges and terminated oxygen atoms significantly reduced the

initial nucleation rate of graphene on Cu(111), which is in agreement with previous experimental observations³².

Fig. R11 The significant effect of oxygen surface contamination on the nucleation process. The presence of C-Cu-O bridges and oxygen terminal groups on the Cu(111) surface significantly reduces the nucleation rate of carbon, resulting in the formation of cross-linked carbon chains instead of small graphene islands.

- ✓ In the revised manuscript, we have included a comparison of carbon growth on Cu(111) with and without oxygen contamination, as depicted in **Fig. R10** and Fig. 5 in the main text. The results demonstrate a significant reduction in the number of sp^2 -C atoms in the presence of oxygen contamination. Furthermore, we have included growth process images in the Supplementary Information (**Fig. S9**) to illustrate the impact of C-Cu-O bridges and oxygen-terminated carbon chains. These simulations showcase the high transferability and flexibility of the CGM-MLP MD/tfMC method in predicting the growth behavior of carbon on metallic surfaces containing three or more elements. Although this study specifically focuses on oxygen contamination, we believe that the contamination of other elements, such

as hydrogen, can be readily simulated using the training framework established in this work.

Other minor points to consider:

1. The authors stated in the paper that ‘little is known about dynamic and atomic-level factors that control the quality of graphene, such as nucleation and growth kinetics.’ (page 2, line 37-38). Unfortunately, I cannot agree with this statement in view of the ever-growing body of literature addressing the atomistic simulations, see the recent review papers: Bohwmik et al. (iScience 2022), Dong et al. (J. Phys. Chem. Lett. 2021), and Momeni et al. (npj Computational Materials 2020).

✓ Thank you for the comments. As indicated by recent review papers, the majority of studies have focused on the growth of graphene on commonly investigated surfaces such as Cu(111), Cu(100), and Ni(111). The authors agree that the growth mechanism of graphene on these surfaces has been extensively studied. However, apart from the well-explored metallic catalysts like Cu and Ni, there have been reports on the efficient production of wafer-scale single-crystalline graphene on less common substrates such as high-index Cu facets³³, Cu-Ni alloys³⁴, Cu₂O surface³⁵, hydrogen-terminated Ge³⁶, and hexagonal boron nitride³⁷. Recent research on these unconventional substrates has indicated that the high-index facets, alloys, and metal oxides may significantly contribute to the growth of high-quality graphene. Nevertheless, the lack of a universal and transferable force field and simulation technique presents challenges in conducting theory-guided design of suitable catalyzed substrates.

To address this challenge, this work offers an active machine-learning framework to

investigate the growth mechanisms of carbon on less well-studied surfaces. The high accuracy and transferability of our training and simulation method have been demonstrated by applying it to different carbon-metal systems and comparing the results with previous DFT calculations and experimental observations.

- ✓ To avoid any potential misunderstanding, we have revised the initial sentence as follows:

Line 36-40:

“While the growth mechanisms on common surfaces have been extensively studied, there is limited knowledge regarding the dynamic and atomic-level factors that govern the quality of graphene on high-index or composite surfaces, including nucleation and growth kinetics. This research gap significantly hinders the development of the theory-guided design approaches for novel catalytic metal substrates in the growth of carbon nanostructures.”

2. Transferability of machine-learned potentials: The authors suggest that their machine-learned potentials can be applied to explore other substrates and design substrate architectures for growing carbon nanostructures. However, it should be noted that active/on-the-fly machine-learned potentials are typically specific to the system for which they are trained. Therefore, further dedicated studies would be required to investigate the transferability of the current potentials to other substrates, similar to what has been done in this study using reference sets of carbon machine-learned potentials (*e.g.* GAP20 from Rowe et al, Journal of Chemical Physics, 2020) and considering the additional interactions with the new substrates under investigation.

- ✓ Thank you for your valuable comments. We acknowledge that active/on-the-fly machine-learned potentials are typically specific to the system for which they are trained. However, the framework and methodology we have developed, including the training protocol, quality

control parameters, and the integration of MLP with MD/tfMC, are readily transferable to other substrates.

In the updated manuscript, we have extended our study by training additional carbon-metal MLPs, specifically Cr-C and Ti-C MLPs, and have considered the C-Cu system with surface oxygen contamination. The simulation results, as shown in **Fig. R10**, reveal that carbon growth on Cr(110) and Ti(001) surfaces exhibits fewer carbon rings compared to the growth on Cu(111), with almost no carbon rings observed on Ti(001). The evolution of sp^2 -C content throughout the deposition process, as shown in **Fig. R10d**, demonstrates that the initial nucleation rate follows the order of Cu, Cr, and Ti, which is consistent with experimental observations (**Fig. R10a-c**) and previous DFT calculations³¹.

Additionally, using the Cu-C-O MLP-based MD/tfMC method, we performed deposition simulations of carbon on O-contaminated Cu(111). The effect of oxygen contamination on the nucleation of graphene is discussed in detail in **Fig. R11**. These simulations demonstrate the high transferability and flexibility of the CGM-MLP MD/tfMC method in predicting the growth behavior of carbon on metallic surfaces containing three or more elements.

We appreciate your valuable feedback, and these updates further strengthen the scope and applicability of our study.

3. Accuracy of energy barriers and reaction energies: The authors claim an 'excellent' agreement between the energy barriers and reaction energies obtained from density functional theory (DFT) and machine-learned potentials (CGM-MPL) for the reaction pathways shown in Fig. 5b and 5c, and Fig. S2. However, a difference of 0.5-1.0 eV between DFT and machine-learned potentials is

far off from the chemical accuracy. The authors should discuss these large differences and their potential implications on the interpretation of the kinetic and thermodynamic processes studied, and compare them with values reported for other types of machine-learned potentials based on the GAP model, as demonstrated in various papers by Deringer, Csanyi, and Bartok-Partay.

- ✓ Thank you for the comments. We appreciate the Reviewer's perspective on the accuracy requirements in precise reaction calculations. In our work, we acknowledge the importance of accuracy and have made efforts to improve it. In the revised version, we retrained the Cu-C MLP using higher quality control parameters ($N_f = 20$, $S^{max} = 0.08$). As shown in **Fig. R12**, the differences between the potential energies along the reaction coordinate calculated by CGM-MLP (solid lines) and DFT (dashed lines) are generally below 0.2 eV/atom. By adding more structures to the training set, the accuracy of the predictive energy barrier can be further improved to less than 0.1 eV/atom. This level of accuracy represents a significant improvement compared to classical empirical potentials. It should be noted that the flexibility of our framework allows researchers to adjust the level of accuracy according to their specific research requirements and available computational resources.
- ✓ However, we must acknowledge that the accuracy achieved by MLPs may still be insufficient for conducting detailed reaction pathway inference. In the context of inferring the reaction pathway of a complex system, starting directly with a highly accurate method such as AIMD may be computationally expensive and resource-intensive. A compromised but more practical approach is to combine MD simulations with tfMC based on an active MLP. This approach allows us to explore a wide range of possible reaction pathways efficiently, even though some of them may have energy differences that deviate from strict

accuracy. Subsequently, a more accurate calculation, such as DFT-NEB calculations, can be performed to validate and refine the identified pathways. This step helps to minimize the potential impact of MLP errors. If we initially lack the knowledge about the reaction pathways of carbon growth on a new metallic surface, conducting MLP-based MD/tfMC simulations can be an adventurous yet promising approach before more refined calculations.

Fig. R12 Minimum energy paths of carbon diffusion and graphene nucleation obtained using CGM-MLP and DFT-based CI-NEB calculations. (a) C monomer and dimer diffusion, (b) conversion from dimers to carbon chains or rings, and (c) graphene growth with passivated edges by Cu atoms. Carbon atoms (black), Cu atoms (orange), and adsorbed

Cu atoms (wheat). Inset: Incorporating edge passivated Cu atoms reduces the energy barrier for graphene growth (0.95 eV/atom) compared to pristine edges (2.47 eV/atom)³. CGM-MLP-based MD/tfMC captures the diverse growth pathways accurately and efficiently.

- ✓ In addition, we have compared the energy differences obtained in this work with the values reported for MLPs in previous studies, specifically by Deringer, Csányi, Michaelides, and others.^{1,30} We find that an energy error of approximately 0.05 eV/atom and a force error of approximately 0.5 eV/Å may represent the highest level of accuracy achieved by MLPs for carbon-based systems, including the neural network potentials (NNP)²⁵ and Gaussian approximation potential (GAP).⁹

Fig. R13 presents the energy and force errors of two recent studies on the MLPs for carbon-based systems, specifically GAP¹ and NNP.³⁰ For amorphous carbon bulks, although the majority of the average energy errors are below 0.05 eV/atom (indicated by the dashed red line), there are still data points that exceed this line. The force errors, as shown in **Fig. R13b**, are approximately 0.5 eV/Å when the training parameters l_{max} and n_{max} are set to 4 and 12, respectively.

Another work in **Fig. R13c** shows the poor performance of classical potentials, such as Tersoff and ReaxFF, when compared to machine-learned potentials like GAP and NNP. However, even with these MLPs, energy errors still range from 0.05 to 0.1 eV/atom. Considering the total number of atoms involved in our NEB calculations, the energy errors observed in our work are comparable to the latest advancements in the field.

Fig. R13 Reported machine-learned potentials for the carbon-based system. (a) Energy and (b) force errors between DFT and GAP-20¹. (c) Energy error predicted by NNP compared to DFT and other models.³⁰

- ✓ The limited accuracy of MLPs for carbon systems can also be attributed to several factors:
 - 1) **Cutoff Radius:** Similar to classic empirical potentials, MLPs also employ a cutoff radius for energy and force calculations. This cutoff may introduce errors by neglecting long-range dispersion interactions. One effective approach to mitigate this issue is the use of a semi-analytical two-body term, which has been shown to correct these errors (Deringer, Csányi, and Michaelides, 2020).¹
 - 2) **Complex Carbon Interactions:** Unlike metallic systems, the interactions between carbon atoms are inherently challenging to describe due to the complexity of competing hybridizations and long-range effects associated with π electrons. Developing specialized descriptors tailored for carbon-based structures would be applicable but falls outside the scope of this work.

3) **Prior Approach for Training Set Selection:** In this work, a prior approach is used to select the most representative structures and include them in the training set. However, this approach only relies on the descriptor and does not involve any DFT calculations to determine uncertainty. Consequently, it may not fully account for the corrugation of the potential energy surface, resulting in relatively larger errors in energy barriers. As demonstrated earlier (**Fig. R12**), this issue can be addressed by adjusting the screening parameters proposed in this study.

Despite these limitations, this study's MLP-based MD/tfMC methods remain valuable tools for exploring atomistic growth mechanisms. While MLPs may have certain accuracy constraints, they still offer a practical and efficient approach to studying complex systems such as carbon growth on metallic surfaces.

✓ To address the Reviewer's concerns, we have made additional statements in our paper to provide clarity on the potential impact of the energy error. Specifically, the potential implications of the energy error are discussed in the context of Fig.4 in the main text. Furthermore, we have included testing results for screen parameters (N^f and S^{max}) in Fig.4c. These additions aim to provide a more comprehensive understanding of the limitations and uncertainties associated with our approach.

4. A coordination/hybridization (sp/sp²/sp³) analysis on the carbon atoms should be performed to accurately quantify the different carbon allotropes present in the model systems, as done in similar previous studies. This analysis will support statements such as 'The less graphene-like structure may be mainly attributed to breaking carbon rings into 1-dimensional chains.' (l. 237)

✓ Thank you for your suggestions. We have performed a hybridization analysis on the carbon

atoms deposited on Cu(111) surfaces. **Fig. R14a** shows the evolution of carbon hybridization types during the deposition simulation. At the incident energy of 2.5 eV, the deposited carbon atoms primarily form carbon chains, resulting in a higher proportion of sp-hybridized carbon compared to sp²- or sp³-carbon. With increasing incident energy, the content of sp²-carbon initially increases and then decreases, indicating that high-energy bombardments can induce the breaking of carbon rings into 1-dimensional carbon chains.

- ✓ To provide direct evidence of the process, **Fig. R14b** presents the breaking of carbon rings into chains under high-energy bombardments. The process starts with a carbon ring on a Cu(111) surface subjected to a 10 eV carbon atom bombardment (seen as the timestep 0). The subsequent images demonstrate the gradual dissociation of carbon rings and the formation of carbon chains due to the impact of high-energy carbon atoms. This observation aligns with the findings in Fig. 4 of the main text, highlighting that high-energy bombardment impedes the formation of carbon rings and, consequently, the nucleation of graphene. Additionally, we have included an animated demonstration of carbon ring breaking in the Supplementary Materials, titled "Carbon_Ring_Breaking.mov" showcasing the powerful capability of the MLP-based MD/tfMC method in capturing dynamic atomistic processes.

Fig. R14 Carbon structure analysis and observation of carbon ring breakage by high-energy bombardments. (a) Evolution of hexagonal carbon rings and hybridization analysis with increasing deposited carbon atoms. The maximum number of hexagonal carbon rings and optimal sp^2 -content are achieved at a moderate incident energy of 5.0 eV. Lower or higher incident energies hinder graphene formation due to insufficient adsorbed Cu atoms or carbon ring breakage, respectively. (b) MLP-based MD/tfMC dynamic simulation capturing the transformation from carbon rings to chains. Coordination numbers with a 1.8 Å cutoff radius for the nearest neighbors were used to determine the hybridization orbitals

- Regarding the use of COMB3 results, Authors claim that ‘If one uses the existing empirical potentials to simulate the growth process of carbon growth on metals, the simulation results

could appear inconsistent with the experiments either at the initial or final stage of nucleation (Comparative simulation results based on empirical Cu-C potentials are available in the supplementary information).’ In contrast, Klaver et al (Carbon, 2015) showed that COMB3 can reproduce the experimental finding to a satisfactory degree. The authors should further comment on this discrepancy and provide direct comparisons with experimental results (with proper citations) that contradict their COMB3 results, along with relevant discussions in the supporting information.

- ✓ Thank you for the suggestion. The COMB3 potential has been utilized in previous studies to study the annealing-induced wrinkles of graphene on Cu(100) and Cu(111) surfaces. However, similar to other empirical carbon potentials, the COMB3 potential has its specific range of applicability. For instance, it is well-suited for simulating organic-copper interactions²⁴ and studying the fracture³⁸ or deformation behavior¹⁸ of Cu/graphene composites, particularly involving larger carbon-based molecules and Cu slabs. However, when it comes to simulating the deposition and growth process of graphene on Cu surfaces, it is essential to have a potential that accurately describes the interaction and reaction regimes of small carbon-based molecules on metallic surfaces, including carbon monomers, dimers, and clusters comprising a few dozen atoms²⁷. Unfortunately, the COMB3 potential exhibits limitations in accurately reproducing the process from small carbon clusters to initial graphene islands.
- ✓ **Fig. R15** presents a comparison of final structures obtained from MD/tfMC simulations using different force fields: CGM-MLP, COMB3, and ReaxFF. The simulation using CGM-MLP produces the most realistic results, with the formation of a graphene island on Cu(111).

In contrast, when carbon atoms are deposited individually on a Cu(111) surface using COMB3, no graphene islands are observed, which contradicts to the experimental observations.

Fig. R15 MD/tfMC deposition simulations with different force fields. (a) CGM-MLP (b) COMB3³⁹ (c) ReaxFF⁴⁰.

6. The authors should discuss how they control/prevent the formation of thin carbon films beyond graphene, particularly at higher impact energies. This discussion would provide insights into potential limitations and challenges associated with their simulations.

- ✓ Thank you for the suggestions. To avoid the formation of thin carbon films, it is essential to have precise control over the deposition energy of carbon atoms or other ion bombardments.² The simulations conducted in this study highlight the importance of maintaining the deposition energy within an optimal range to promote the nucleation of graphene. Excessive bombardment energy, whether too high or too low, may hinder the nucleation process, leading to the formation of carbon chains instead of carbon rings.

Previous experimental and theoretical studies have indicated that nanocrystalline carbon films are typically obtained within a deposition energy range of 15 to 60 eV². Within this range, carbon chains are more likely to form and subsequently undergo cross-linking, transforming into disordered carbon structures. If the deposition energy is sufficiently high or if other ions, such as Ar (commonly used in physical vapor deposition processes), impact the deposited carbon structures, vertically aligned nanocrystalline carbon films are formed². However, the deposition energies employed in this study do not exceed 10 eV, which falls within the appropriate energy window for growing graphene.

- ✓ The decision to end the iterations in our training framework is based on specific research objectives and the desired number of carbon atoms to be deposited on the metal surface. In this study, we predetermined the number of carbon atoms to be deposited as 100 (25 iterations in total). However, if there is a need to simulate further growth processes or carbon-based films, the number of carbon atoms can be increased accordingly. It is important to note that the microstructure of carbon-based films is strongly influenced by the initial nucleation processes and the metal-carbon interface. As demonstrated in **Fig. R10**, we can predict the structures of carbon films by simulating the initial nucleation processes.

More specific points:

1. line 17: Authors state that they present ‘the first fully dynamic simulations of graphene growth on Cu(111)’. Unfortunately, this statement may not be accurate as similar simulations have been reported by Klaver et al. (Carbon, 2015) using COMB3 reactive interatomic potentials and probably others.

- ✓ Thank you for the comments. The word "fully" here has two main implications. Firstly, it

refers to the fact that in the deposition simulation, both the carbon and metallic substrate atoms are allowed to move. Secondly, it highlights that the simulation process encompasses the entire process from carbon monomers/dimers to the formation of a graphene island. Taking into account the literature you provided, we still believe that this work represents the first instance of meeting both of these criteria in simulations.

As mentioned previously, there are three primary simulation techniques commonly used for modeling carbon growth on metallic substrates: AIMD, DFT-based KMC, and MD simulations employing empirical force fields. In Klaver et al.'s work¹⁸, their focus was primarily on investigating the formation of wrinkles in graphene on Cu(111) using classical MD simulations with the COMB3 empirical force field. They initiated the simulation with a complete graphene structure instead of individual carbon atoms on the Cu surface and performed annealing processes. For this specific application, the MD simulation based on the COMB3 potential may be sufficient. However, it is important to note that this potential exhibits limitations in accurately describing the reactions involved in the nucleation and growth of graphene on Cu surfaces. **Fig. R16.** illustrates the results of the COMB3-based simulations, revealing that during the initial stages, the deposited carbon atoms are capable of diffusing into the sublayer of the Cu(111) surface, as observed in Fig. 3 of the main manuscript. However, it is notable that the carbon atoms tend to form metal carbide rather than graphitic structures on the Cu(111) surface, and even at the end of the simulations, no graphene structures are observed. This outcome contradicts to both experimental observations and DFT calculations. Consequently, it suggests that the COMB3 potential may not be suitable for accurately simulating the growth of graphene on metal surfaces.

Fig. R16 COMB3-based³⁹ hybrid MD/tfMC simulations of carbon growth on Cu(111) surface. (a) Top view (b) Left view.

Alternative methods such as KMC and AIMD have their limitations when it comes to fully simulating the deposition process. KMC requires the substrate to be fixed, which restricts its ability to capture the dynamic behavior of the deposition process. On the other hand, AIMD simulations are limited by their time scale, making it challenging to simulate the entire deposition process accurately. Therefore, we are confident that this study represents the first instance of fully dynamic simulations of graphene growth on Cu(111) that successfully reproduce a sequence of critical subprocesses during the growth process.

2. Line 99: The reference seems to be incorrect, and it is likely that the authors are referring to Rowe et al. (ref 32) instead of ref 35. It is important to double-check the in-text citations and references for accuracy.

✓ We would like to thank the Reviewer for pointing out the discrepancy in the reference. The correct reference should be Rowe et al. (ref 32) instead of ref 35. We have made the necessary changes to ensure the accuracy of the citation. We also thoroughly reviewed the entire manuscript and cross-checked all in-text citations and references for accuracy.

3. Line 140: The authors should provide more information on how they determine that the training of CGM-MLP is completed. Clarifying the criteria or convergence metrics used for training completion would be helpful.

✓ Thank you for the suggestion. Firstly, we define one iteration as a complete cycle consisting of MLP fitting, MD/tfMC simulations, structure selection, DFT calculations, and error calculations. Following each iteration, to assess the accuracy of the energy calculations, we use an energy criterion where the energy error should be less than 0.05 eV/atom. Similarly, we employ a force criterion where the force error should be less than 0.5 eV/Å. Based on these criteria, we assess whether the newly generated structures should be added to the training set and undergo another fitting procedure, or if we should proceed with the deposition of carbon atoms without the trainset fitting process, as shown in Fig. R17.

Fig. R17 Energy and force error of the initial 20th iteration during a training process.

We define one iteration as a comprehensive cycle encompassing MLP fitting, MD/tfMC simulations, structure selection, DFT calculations, and error calculations. The iteration continues until the energy error is below 0.05 eV and the force error is below 0.5 eV/Å. If

these criteria are met, the training proceeds to the next cycle. If the energy or force error exceeds the specified thresholds, the selected structures are added to the training set, and the current cycle is repeated.

The decision to end the iterations depends on the specific research objectives and the desired number of carbon atoms to be deposited on the metal surface. The decision to end the iterations depends on the specific research objectives and the desired number of carbon atoms to be deposited on the metal surface. In our training framework, the number of carbon atoms to be deposited is predetermined, allowing for adjustments in the total number of iterations, the accuracy criteria for energy and forces, and the selection of screening parameters based on the specific research objectives. This flexibility enables customization and optimization of the simulation parameters according to the requirements of the study.

4. Line 174: Figure 5 appears before Figure 3 in the text, and this order should be corrected to match the order of appearance in the text.

✓ Thank you for pointing out the issue regarding the order of appearance of figures in our manuscript. We have made the necessary correction, and Fig. 5 never appears before the citation of Fig. 3. We appreciate your valuable feedback and apologize for any confusion caused.

5. Line 228: The statement 'achieving excellent agreements with previous DFT studies' requires references to support the claim. The authors should provide appropriate references to validate their findings.

✓ Thank you for your feedback about the lack of necessary references in the figure caption.

We have included appropriate references in the related sentences, which now read as

follows:

Line 235-238:

“The fully dynamical simulations have correctly reproduced many subprocesses, such as the spontaneous diffusion of carbon monomers and dimers⁴¹, the stabilization effect of carbon-metal bridges for carbon rings^{3, 42}, the energetically favorable property of carbon chains⁴³, and the carbon ring breaking process⁴³, achieving excellent agreements with previous DFT studies⁴⁴.”

6. Line 231: Change 'step-to-step' to 'step-by-step' for better clarity and accuracy.
 - ✓ Thank you for your feedback on our manuscript. We have made the necessary revision. The phrase now reads as 'step-by-step,' which accurately conveys the intended meaning.
7. Line 278: Change 'as' to 'with' to improve the wording and accuracy of the sentence.
 - ✓ Thank you for your attention to detail. We have made the necessary revision based on your suggestion. The sentence now reads as follows: *"it is observed that the binding energy of initial graphene clusters increases **with** the addition of passivated Cu atoms."*
8. Line 303: 'Gaussian approximation potential' should be written as 'Gaussian Approximation Potential' for correct capitalization and consistency.
 - ✓ Thank you for bringing this to our attention. We have made the necessary correction in the manuscript. The term 'Gaussian Approximation Potential' is now capitalized and consistent throughout the text.
9. Fig. 2a: The type of energy reported in the plot should be clarified, whether it is the total energy per atom or formation energy, for better understanding.
 - ✓ Thank you for your feedback. We have revised Figure 2a (the updated version is presented

in Fig. S3) and provided clarification regarding the type of energy reported in the plot. The energy represented in the plot is the total energy per atom, which can be calculated using the following equation.

$$E_{ave} = \frac{E_{total}}{N}$$

where E_{total} is the total energy of a structure and N is the number of atoms in a structure.

Fig. R18 Energy/force correlation plots in production runs by using different quality control parameters, namely N_f , S^{max} , and S^{ave} . (a) Energy and (b) force correlation plots.

10. Fig. 2b: It is not clear how the authors define and determine the force error shown in Fig. 2b. The authors should provide more details, such as whether the errors are averaged over all atoms (RMSE) or averaged over the trajectory, and how it is directly linked to the number of deposited atoms.

✓ Thank you for your suggestion. In Fig. 2b, the force errors are defined as the average error of the force components along three directions (x, y, and z) for all atoms between two structures. The force error is calculated using the following equation:

$$F_{error} = \frac{|F_x^{DFT} - F_x^{MLP}| + |F_y^{DFT} - F_y^{MLP}| + |F_z^{DFT} - F_z^{MLP}|}{3N}$$

where F_x , F_y , and F_z represent the x, y, and z components of the force obtained from both

DFT calculations and other force fields, including the MLP and classical empirical potentials, and N is the total number of the structures.

To link the force error with the number of deposited atoms, we employed the CGM-MLP and MD/tfMC methods to simulate the growth process of graphene on metallic surfaces. During the simulations, we obtained trajectories under various incident energies, and from each simulation with incident energies of 2.5, 5.0, 7.5, and 10 eV, we selected 100 structures that spanned a range of 1 to 100 carbon atoms. For each of these structures, we calculated the atom forces and subsequently determined the force errors using the equations mentioned above. By analyzing these force errors, we were able to plot them as a function of the number of carbon atoms deposited on metallic surfaces.

- ✓ We have provided a more detailed explanation of the methodology used to link force errors with the number of deposited atoms in the Supplementary Information file.

11. Fig. 3: The labels 'I', 'II', 'III', etc. used in the caption of Fig. 3 need to be clarified and explained for better comprehension.

- ✓ Thank you for your suggestion. In the revised version, for the convenience of positioning the images in the figure, we have kept the labels 'I', 'II', and 'III'. Additionally, we have incorporated the number of carbon atoms to label each simulation snapshot, as shown in **Fig. R19**. This modification enhances clarity and improves the understanding of the simulation results.

Fig. R19 CGM-MLP driven simulations of graphene growth on Cu(111) with different carbon incident energies. *i.e.*, a 2.5 eV, b 5.0 eV, c 7.5 eV, and d 10 eV. Carbon atoms are colored black, and copper atoms are color-coded according to their height coordination. Different graphitization degrees can be observed when the carbon incident energy varies during the MD/tfMC simulations. The fully dynamical simulations have correctly reproduced many subprocesses, such as the spontaneous diffusion of carbon monomers and dimers⁴¹, the stabilization effect of carbon-metal bridges for carbon rings^{3, 42}, the energetically favorable property of carbon chains⁴³, and the carbon ring breaking process⁴³, achieving excellent agreements with previous DFT studies⁴⁴.

12. Fig. 4: The inclusion of 'Rings' in the key legend of Fig. 4 may not be necessary. The authors should provide more details on how the rings were analyzed (i.e., method used), and consider including the expected number of rings for achieving ideal graphene for each cluster size as a guide-to-eye.

- ✓ Thank you for your suggestion. In the revised version, we have removed the term "Rings" from the key legend as per your suggestion. The definition of an N-atom ring refers to a closed loop formed by N consecutive atoms in a molecular structure. To analyze the number of N-atom rings in a LAMMPS simulation trajectory, we have provided the source code called "Carbon_Ring_Analysis.cpp" at our GitHub: <https://github.com/sjtudizhang/CGM-MLP>. This code counts the number of such rings present in the structure based on the specified value of N and a bond cutoff of 1.8 Å.
- ✓ To provide a guide-to-eye reference, we have added the expected number of rings for achieving the ideal graphene for each cluster size as shown in **Fig. R20**. For any positive integer N values greater than or equal to 6, the maximum number of hexagonal rings can be calculated using the formula $(N-3)//3$, where the "//" operator denotes floor division.

Fig. R20 Evolution of hexagonal carbon rings and hybridization analysis with increasing deposited carbon atoms. we have added the expected number of rings for achieving ideal graphene for each cluster size to provide a guide-to-eye reference.

13. Fig. 5: Adding a color code for different elements in Fig. 5 would be helpful for a better interpretation of the results.

- ✓ Thank you for your suggestion. We have revised Fig. 5 by adding a color code to distinguish between different elements. Carbon atoms are represented in black, Cu atoms in orange, and adsorbed Cu atoms in wheat. The updated version is as follows:

"Fig. 1 Minimum energy paths of carbon diffusion and graphene nucleation obtained using CGM-MLP and DFT-based CI-NEB calculations. (a) C monomer and dimer diffusion, (b) conversion from dimers to carbon chains or rings, and (c) graphene growth with passivated edges by Cu atoms. Carbon atoms (black), Cu atoms (orange), and adsorbed Cu atoms (wheat). Inset: Incorporating edge passivated Cu atoms reduces the energy barrier for graphene growth (0.65 eV/atom) compared to pristine edges (2.47 eV/atom). CGM-MLP-based MD/tfMC captures the diverse growth pathways accurately and efficiently. "

14. Computational Details of the CI-NEB calculations, such as the number of images, convergence criteria, optimization algorithm, etc., are missing in the text as well as in the Supporting Information. The authors should provide these details for transparency and reproducibility.

✓ Thank you for pointing out the missing computational details in the manuscript. The CI-NEB calculations were performed using the following specifications: We used a total of 4 or 8 replica geometries along the path with a spring constant of 0.001 to restrain the replicas. Convergence criteria were set as follows: the maximum displacement (MAX_DR) was limited to 0.02 Å, the maximum force (MAX_FORCE) to 0.05 eV/Å, the root mean square displacement (RMS_DR) to 0.01 Å, and the root mean square force (RMS_FORCE) to 0.001 eV/Å. The optimization algorithm employed was Direct Inversion in the Iterative Subspace (DIIS), with a maximum of 1000 optimization steps and 3 DIIS vectors. The BAND_TYPE used for the calculations was Climbing-image NEB (CI-NEB). We apologize for the oversight in not including these details in the text and Supporting Information. We will update the manuscript to provide transparency and reproducibility of the calculations.

15. The study by Qiu et al. (Acc. Chem. Res., 2018, 51, 3, 728-735), which presents kinetic Monte Carlo simulations combined with first-principles molecular dynamics for studying the kinetics and mechanistic studies of undoped/N-doped graphene growth on Cu(100) and Cu(111) surfaces, and other relevant studies should be cited to provide a comprehensive review of the literature.

✓ Thank you for your suggestion. We have thoroughly reviewed the existing literature on atomistic simulations of carbon growth on metallic surfaces. Related statements regarding the atomistic simulation of carbon growth on metallic surfaces, including the significant contribution of Qiu et al. (Acc. Chem. Res., 2018, 51, 3, 728-735), are as follows:

Line 48-62:

"Theoretical simulations, such as kinetic Monte Carlo (KMC) and ab initio molecular dynamics (AIMD), have significantly enhanced our understanding of the graphene growth on certain metal surfaces. However, it is still challenging to go beyond these atomic details to obtain the fully dynamic and overall picture of carbon growth on an arbitrary metal or alloy substrate. For instance, Qiu et al.⁴⁵ used KMC simulations coupled with DFT calculations to study the diffusion-limited growth of carbon dimers on Cu(111) and Cu(100), highlighting the crucial role of hydrogen in the growth process. Nevertheless, KMC simulations may have certain limitations in capturing the complete dynamics of carbon growth on a dynamically evolving substrate due to the construction of event tables in off-lattice simulations. On the other hand, AIMD suffers from limited time and length scales despite advancements in computational power. Conventional methods such as MD/MC simulations with empirical interatomic potentials face difficulties in accurately calculating interactions between graphitic layers², experimentally observed sp^2/sp^3 fractions²⁷, and the interactions between carbon species and metal surfaces¹⁸ due to the limitations of their fixed functional form."

Thank you very much for your insightful comment. We greatly appreciate your valuable feedback and we have carefully considered your suggestion and have made the necessary revisions to improve the quality of our work. We hope that our revisions have satisfactorily addressed your concern. Once again, thank you for your contribution to our work.

To Reviewer #3:

In this study, the authors investigate the catalytic growth of carbon nanostructures on Cu(111) using atomistic simulations based on a machine learning potential (MLP). They employ an active learning strategy to train the MLP to density functional theory calculations. Using a Monte-Carlo (MC) sampling scheme they simulate the growth process upon carbon atom deposition from the gas phase. In their simulations, the authors observe many processes of the early stages of growth like carbon monomer diffusion and dimer formation, their nucleation, and the subsequent growth to chains and hexagonal clusters. Additionally, the authors find the Cu surface to be dynamically changing during the growth process. Based on their observations they propose a mechanism highlighting a stabilizing role of Cu adatoms, which can act as nucleation seeds for chains or hexagonal clusters depending on their abundance.

The present studies' concept and the overall simulation strategy are appealing and the active learning is in accord with state-of-the-art MLP protocols. The demonstration of the complex growth of carbon nanostructures on a dynamically changing catalyst surface is scientifically interesting and a great demonstration of MLP capabilities. Despite this favorable general assessment, the study reveals profound shortcomings in the training and quality assessment of the MLPs. Many standard procedures to ensure the reliability of MLPs are omitted which shows in the resulting accuracy of the few benchmark cases presented. This is contrasted by conflictive language praising the produced MLP as "highly accurate" and celebrating the quantitative agreement, while errors are as large as > 1 eV for different processes. Qualitatively, simulation results are likely physical, but quantitatively below the standard commonly reachable for MLPs. This demonstration of a rather sloppy handling of MLP training which leads to inferior accuracy compared to other purpose-driven MLPs, does not

include standard quality control, and is accompanied by misleading language may in fact hurt the careful standards developed by the MLP community in the last decade. In addition to this major issue, the publication appears in part incomplete since its missing many details necessary to understand/appreciate the conducted work and is accompanied by a partially imprecise and ambiguous language.

Dear Reviewer,

Thank you for your detailed review and valuable feedback. We appreciate your positive comments on the concept and simulation strategy employed in our study. We have carefully considered your concerns regarding the training and quality assessment of the MLPs and have made the necessary revisions to address these issues.

Regarding the training and quality assessment of the MLPs, we have incorporated a significant number of calculations and provided a more detailed description of the Gaussian Approximation Potential (GAP) parameters used, the selection process of the training and testing sets, as well as the plots depicting the force/energy correlation. Additionally, we have conducted additional benchmark cases with different quality control protocols to enhance the reliability and accuracy of the CGM-MLPs.

Fig. R21 The position of this work within the context of existing research, is determined by evaluating two critical factors: timescale and accuracy. This work takes a pioneering step in integrating MLPs and tfMC, aiming to advance the development of deposition simulation techniques.

As shown in **Fig. R21**, our integration of MLPs and MD/tfMC represents a pioneering approach that allows for a fully dynamic simulation providing a comprehensive view of the growth process with significantly improved accuracy compared to classical empirical force fields. This framework provides an efficient means to explore a wide range of possible reaction pathways. Subsequently, more accurate calculations, such as DFT-NEB, can be performed to validate and refine the identified pathways. In addition, we fully understand the importance of maintaining high accuracies in simulating chemically reactive processes. In the revised version, we have retrained the Cu-C MLP using stricter quality control parameters. The flexibility of our framework allows researchers to adjust the level of accuracy according to their specific research requirements and available computational resources.

We have carefully addressed each of your detailed comments and suggestions:

Detailed comments

Comment 1: No basic information about the training results is given. I.e. beyond the starting structures – which also don't contain details about the carbon clusters on Cu(111) – it is unclear what model structure was used in the MD/tfMC training simulations, how many configurations were added each iteration, or in total, and how many iterations were conducted. Force/energy correlation plots & learning curves are missing and no final energy, force, and stress error is provided. The error metrics are not clearly defined, only a formula is provided in the SI showing a MAE metric for the forces.

- ✓ We appreciate the Reviewer's feedback. In response to this concern, we have included a more comprehensive description of the training methodology and results in the revised manuscript.

The initial training dataset includes GAP-20 carbon structures¹ and all the C₁-C₁₈ structures on the Cu(111) surface. By including these structures, the initial training dataset provides a good starting point for generating a machine learning potential capable of accurately describing Cu-C interactions. The starting structures and all the final training structures are now available at https://github.com/sjtudizhang/CGM-MLP/training_structures, which are formatted as an extended XYZ file. The final structures used for training the MLP for MD/tfMC simulations are also available on the above website. The number of final structures for Cu-C, Cr-C, Ti-C, and Si-C-O are 7885, 8103, 8041, and 9608, respectively.

Fig. R22 presents the energy and force error during a training process, as well as the number of structures in the training dataset. Each iteration includes MLP fitting, MD/tfMC

simulations, structure selection, DFT calculations, and error estimation. During the error estimation step, we use an energy criterion of less than 0.05 eV/atom and a force error of less than 0.5 eV/Å to determine whether newly generated structures should be added to the training set or if the deposition of carbon atoms can proceed without further training. The number of structures added to the training set in each iteration depends on the sampling frequency (N_f) and screen parameters (S^{max} and S^{ave}). For instance, with $N_f = 20$, S^{max} , $S^{ave} = 0.08$, approximately 40~60 structures will be added to the training set.

Fig. R22 Energy and force error of 20 iterations during a training process. An energy criterion of less than 0.05 eV/atom and a force error of less than 0.5 eV/Å is used to determine whether newly generated structures should be added to the training set or if the deposition of carbon atoms can proceed without further training. With $N_f = 20$, S^{max} , $S^{ave} = 0.08$, approximately 40~60 structures will be added to the training set.

- ✓ We have included the final energy, and force correlation plots in the revised Supplementary Information. During the training cycles, a subset of structures was selected from the MD/tfMC simulations using different values of N_f , S^{max} , and S^{ave} . Meanwhile,

approximately 500 structures were randomly selected from the held-out structures to serve as the testing sets. By increasing N_f and lowering S^{max} , as shown in **Fig. R23a-b**, the CGM-MLP exhibits improved energy/force accuracies and a significant improvement compared to classical empirical potentials, such as COMB3³⁹ and ReaxFF⁴⁰. Specifically, the energy and force errors of the CGM-MLP trained with $N_f = 20$ and S^{ave} , $S^{max} = 0.08$ converge to approximately 0.013 eV and 0.43 eV/Å, respectively (**Fig. R23c**).

Fig. R23 Energy/force correlation plots in production runs by using different quality control parameters, namely N_f , S^{max} , and S^{ave} . (a) Energy and (b) force correlation plots. (c) Production runs by using different S^{max} , with S^{ave} fixed at a low value of 0.08.

- ✓ In this study, we chose not to include Virial stresses in the training data. This decision was based on the fact that Virial stresses are primarily used for obtaining elastic constants in

periodic solids, which is not the focus of our research. Since the deposition simulation involves a nonperiodic boundary along the Z axis, the elastic constants are not of particular interest in this study. Therefore, we omitted the inclusion of Virial stresses in our training data.

Comment 2: The convergence criteria are unreasonable loose and not adequate for MLPs. An energy error of 0.05 eV/atom and a force error of 0.5 eV/Å are about an order of magnitude higher than what is usually reported. The large final errors are somewhat recognizable in Fig. 2 which shows a non-discussed spread for low-energy structures and a systematic shift in the total energy of > 0.5 eV which needs to be discussed. The y-axis for the force error in Fig. 2 requires magnification to judge the accuracy of the MLP.

- ✓ We appreciate the Reviewer's comment regarding the convergence criteria for our MLPs. We agree that the chosen energy and force errors of 0.05 eV/atom and 0.5 eV/Å may appear relatively loose compared to some reported values. However, it is worth noting that the framework presented in this work offers a flexible approach where the level of accuracy can be adjusted based on specific research requirements and available computational resources.

To address this concern, we conducted a series of additional tests and optimizations. We generated a subset of structures with a sampling rate of N_f from the MD/tfMC simulations to serve as the initial training dataset for the MLPs. A screening process was then employed with the parameter S^{ave} and S^{max} to further refine the dataset. During training, 500 data points were randomly selected from the held-out structures as a test set. We focused on enhancing the sampling rate of MD/tfMC simulations (N_f) and tested different values of the screening parameter (S^{max}). By increasing N_f and lowering S^{max} , as shown in **Fig. R24a-b**,

the updated CGM-MLP exhibits improved accuracy (red points) compared to the originally trained MLP (yellow points) with $N_f = 5$ and $S = 0.08$. **Fig. R24c** also revealed that the energy and force errors converged to approximately 0.013 eV and 0.43 eV/Å, respectively, as S^{max} varied while keeping N_f fixed at 20.

Regarding the original Fig. 2, we have considered the Reviewer's suggestion. We removed the original version and replaced it with Fig. 1c and additional figures in **Fig. S3** that provide a clearer representation of the force error and the accuracy of the CGM-MLPs.

Fig. R24. Energy/force correlation plots in production runs by using different quality control parameters, namely N_f , S^{max} , and S^{ave} . (a) Energy and (b) force correlation plots. (c) Production runs by using different S^{max} , with S^{ave} fixed at a low value of 0.08.

Fig. R25 Energy and force correlation plots reported by Deringer, Csányi, Michaelides, and others^{1, 30} showing the performance of MLPs in previous studies for complex carbon-containing systems. (a) Energy correlation plot. (b) Force correlation plot. (f) Well-optimized training parameters GAP MLPs used to describe amorphous or defected carbon systems, which still exhibit a force error of approximately 0.4 eV/Å.

- ✓ Despite the improved energy accuracy from 0.05 to 0.013 eV, we acknowledge that the force error has not reached a high level. It is important to consider the inherent challenges in accurately describing the carbon atom interactions, which involve complex hybridizations and long-range effects associated with π electrons. MLPs for carbon-based systems can be more limited in accuracy compared to metallic systems and simpler molecular systems⁴⁶. To validate our results, we have compared the energy differences obtained in this work with the

values reported for MLPs in previous studies by Deringer, Csányi, Michaelides, and others¹,³⁰ **Fig. R25d-f** shows that an energy error of approximately 0.05 eV/atom and a force error of approximately 0.5 eV/Å may represent the highest level of accuracy achieved by MLPs for carbon-based systems, including neural network potentials (NNP)²⁵ and Gaussian approximation potentials (GAP)⁹. Tighter convergence criteria could potentially yield higher accuracy but at the expense of significant computational cost, which was not our intended approach with MLPs.

- ✓ In our study, we aimed to strike a balance between accuracy and computational efficiency. The CGM-MLP-based MD/tfMC approach we employed offers a practical and efficient means to explore a wide range of reaction pathways, surpassing other methods such as AIMD, DFT-based MC, and classical MD using empirical potentials as shown in **Fig. R21**. Based on the dynamic and comprehensive view of growth processes, we can subsequently perform precise calculations, such as DFT-NEB calculations, to validate and refine the identified pathways. Considering the significant improvement of CGM-MLP accuracy compared to classical empirical potentials and the absence of metallic slab-carbon MLPs reported previously, we believe that the achieved energy and force errors are appropriate for our specific research goals.
- ✓ In the main text, we also add more explanations why we used convergence criteria of 0.05 eV and 0.5 eV/Å.

Lines 135-152:

"To evaluate the effectiveness of the screening parameters, we selected a subset of structures from the MD/tfMC simulations using different values of N_f , S^{max} , and S^{ave} .

Meanwhile, approximately 500 structures were randomly selected from the held-out structures to serve as the testing sets. By increasing N_f and lowering S^{max} , as shown in Fig. 1c, the CGM-MLP exhibits improved force accuracies and a significant improvement compared to classical empirical potentials, such as COMB3⁴⁷ and ReaxFF⁴⁸. In the Supplementary Information, we also provide an energy correlation plot (Fig. S3b) and additional tests of the parameter S^{max} for various structure types, including amorphous carbon, graphite crystals, and a Cu(111) slab with carbon clusters and deposited carbon atoms. The results demonstrate that a predictive energy error below 0.05 eV/atom and a force error below 0.5 eV/Å can be achieved for all relevant structures in a Cu-C system when S^{max} is set to less than 0.08. Specifically, the energy and force errors of the CGM-MLP trained with $N_f = 20$ and S^{ave} , $S^{max} = 0.08$ converge to approximately 0.013 eV and 0.43 eV/Å, respectively. Considering the inherent challenges in accurately describing complex hybridizations and long-range effects associated with π electrons in carbon interactions, the energy and force error are considered to be close to the highest-level accuracy achieved by MLPs for amorphous or defective carbon-containing systems^{33, 49}. As a practical and efficient means to explore a wide range of reaction pathways during carbon growth, it is believed that the achieved energy and force errors are appropriate for the research goals."

Comment 3: It appears, that the hyper-parameters and descriptors from the GAP-20 potential were just taken over without adjusting for the inclusion of Cu which may require careful re-validation.

- ✓ Thank you for pointing out the need for re-validation. In response to this concern, we have conducted additional analyses and experiments to address it. In **Fig. R26**, we present the

force errors as a function of hyper-parameters (sparse points and SOAP cutoff) and descriptor parameters (n_{max} and l_{max}) specifically for the Cu-containing system. We generated training sets consisting of Cu(111) slabs deposited with varying numbers of carbon atoms (a total of 1085 structures) using the CGM-MLP training framework. Additionally, we selected testing sets from the structures held out during the training process (a total of 500 structures). The results demonstrate that the hyper-parameters and descriptor parameters we employed, namely *Sparse Points*=9000, *SOAP cutoff*=3.7 Å, l_{max} =4, and n_{max} =12, are capable of achieving the same level of predictive force accuracy as observed with the GAP-20 potential.

Fig. R26 Hyper-parameter and descriptor tests of the Deposited-C@Cu(111) set. the hyper-parameters and descriptor parameters we employed, namely *Sparse Points*=9000, *SOAP cutoff*=3.7 Å, l_{max} =4, and n_{max} =12, are capable of achieving the same level of predictive

force accuracy as observed with the GAP-20 potential.

✓ We have incorporated these calculations and discussions into our revised Supplementary Information to address this concern adequately.

Comment 4: The screening parameter S has several issues: The testing procedure explained in the SI is unclear, *e.g.* how distances are computed (between what sets) and how the test and training set are split exactly. The choice of parameter S is also only weakly motivated. It appears it is expected to be transferable since structures of C and Cu allotropes are compared, but the application of S is for the Cu-C system. It needs to be states (and treated) as an arbitrary and non-transferrable parameter. The similarity measure can imply different errors for different training sets. A performance measure for this criterion would be adequate (statistics of what type of structure were added).

- ✓ Thank you for addressing the concern regarding the screen parameter S . The environmental similarity of two atoms is evaluated using the Euler distance between their respective SOAP vectors. During the evaluation of a candidate structure, each atom is assigned the closest distance, which is the closest Euler distance between the atom and an existing structure. Atoms with larger closest distances are given higher priority for inclusion in the training set. Consequently, the D^{ave} value represents the average closest distance among all atoms in the candidate structure, while the D^{max} value represents the maximum closest distance. For enhanced clarity, we have provided the Python codes for calculating D^{ave} and D^{max} via: https://github.com/sjtudizhang/CGM-MLP/Calculate_similarity.py.
- ✓ To demonstrate the generation of the test and training sets, we have provided a detailed description in **Table R3**. To validate the screening parameter in the Cu-C system, we have generated the structure set of Carbon-Cluster@Cu and Deposited-Carbon@Cu. Additionally,

we have made all the datasets accessible for evaluating the screening parameter S on the website: https://github.com/sjtudizhang/CGM-MLP/testing_screening_params.

Table R3 Generation of the test and training sets used for testing the screen parameter S

Structure Group	Training Set	Testing set	File Name
Graphite	40 graphite structures with different lattice constants ranging from 2.0 to 3.2 Å, with a 0.03 Å increment	20 randomly generated graphite crystal structures with lattice constants ranging from 2.0 to 3.2 Å	Graphite_Training_Set.xyz Graphite_Testing_Set.xyz
Amorphous Carbon	2558 structures selected from the GAP-20 database ¹	493 structures available from the GAP-20 database ¹ , excluding any structures present in the training set.	Amorphous_Carbon_Training_Set.xyz Amorphous_Carbon_Testing_Set.xyz
Carbon-Cluster@Cu	588 structures selected from the AIMD simulation of the Cu(111) slab, including both the C1-C18 clusters on the Cu(111) slab	192 structures were uniformly selected from the AIMD simulation, excluding any structures that are part of the training set.	Carbon_Cluster_Training_Set.xyz Carbon_Cluster_Testing_Set.xyz
Deposited-Carbon@Cu	1090 structures uniformly selected from the MD/tfMC simulation during the training process of CGM-MLPS	468 structures uniformly selected from the MD/tfMC simulation, excluding any structures that are part of the training set.	Deposited_Carbon_Training_Set.xyz Deposited_Carbon_Testing_Set.xyz

- ✓ Following the creation of the training and test sets, we proceeded to screen the training sets using various screen parameters (S^{max}) ranging from 0.06 to 0.40, while the S^{ave} was fixed at 0.08. The screened training sets were then employed to train MLPs. Subsequently, these trained MLPs were used to calculate the energy errors for the corresponding test sets. The results of these calculations are presented in **Fig. R27**. It is worth noting that the similarity measure can imply different errors for different training sets. Generally, the more drastic the variations in the potential energy surface, while achieving the same level of predictive

accuracy, the smaller the value of S^{max} is required. In different system applications, it is necessary to test various systems based on the precision requirements and ultimately select the appropriate value of S^{max} and S^{ave} . In this study, it was found that when the value of S^{max} is below 0.1, all predictive errors remain below 0.05 eV/atom. Therefore, we selected S^{max} , $S^{ave} = 0.08$ as the parameters to train the CGM-MLP in this work.

Fig. R27 The screen parameter S^{max} and corresponding predictive accuracies for different structures, with a fixed low S^{ave} value of 0.08.

- ✓ In Fig. R27, we also conducted tests to examine the relationship between S and predictive accuracy when using only D^{ave} , as well as when using both D^{ave} and D^{max} , to filter the candidate structure dataset. Previously, a configuration-averaged kernel metric was proposed to do an active machine learning⁸. However, directly applying the configuration-averaged metric to the on-the-fly training of deposition simulations may not be fully effective. This is because, during the deposition simulation, the coordination environment of most atoms in the substrate may have slight changes. Relying only on the configuration-averaged metric for selecting new structures could omit structures that exhibit significant variations only in the local areas surrounding the deposition atom. To address this limitation,

we propose not only using the configuration-averaged similarity metric (D^{ave}) but also including the structure with a single atom displaying the most distinct environment (D^{max}). In **Fig. R27**, by simultaneously using both D^{ave} and D^{max} as filtering criteria, as compared to solely utilizing D^{ave} , we achieve the same accuracy level with more relaxed screen parameters.

Comment 5: The quality of the MLP is not insured re-checked in production runs (e.g. via further similarity screening).

- ✓ Thank you for your suggestions. As previously mentioned, we conducted seven production runs of MLPs, incorporating different similarity screening parameters (S^{ave} and S^{max}). The force/energy correlation plots depicted in **Fig. R28** (a) and (b) demonstrate the enhanced accuracy and quality obtained through an increase in N_f and a decrease in S .

Fig. R28. Energy/force correlation plots in production runs by using different quality control parameters, namely N_f , S^{max} and S^{ave} . (a) Energy and (b) force correlation plots. (c) Production runs by using different S^{max} , with S^{ave} fixed at a low value of 0.08.

- ✓ Additionally, to ensure the reproducibility of MLPs in practical applications, we repeatedly conducted MD/tfMC simulations under various carbon atom incident energies (Fig. R29 and Fig. R30). This allowed us to replicate and validate the performance of MLPs.

Fig. R29 CGM-MLP driven simulations of graphene growth on Cu(111) with different carbon incident energies. *i.e.*, **a** 2.5 eV, **b** 5.0 eV, **c** 7.5 eV, and **d** 10 eV.

Fig. R30 Repeated CGM-MLP driven simulations of graphene growth on Cu(111) with different carbon incident energies. *i.e.*, **a** 2.5 eV, **b** 5.0 eV, **c** 7.5 eV, and **d** 10 eV.

Comment 6: The authors do not clarify what a “time stamp” MC scheme is and how it is different from a standard MC simulation; the publication should be self-containing with some brief explanation and not only refer to older publications.

- ✓ Thank you for indicating the need for clarification. We have included additional statements in the Supplementary Information to explain the "time stamp" Monte Carlo (tfMC) scheme in more detail. Originally known as the uniform-acceptance force-bias Monte Carlo (UFMC) method,⁴⁷ tfMC differs from molecular dynamics (MD) algorithms by employing a force-bias probabilistic description of atomic motion. This approach has demonstrated success in

replicating long-term events that occur during phase transitions, surface diffusion, and growth.^{2, 16, 47, 48}. In 2012, Mees *et al.*⁴⁸ introduced a statistically relevant time step per UFMC iteration, resulting in a significant speed-up compared to MD simulations. As a result, the UFMC method became known as the time-stamped force-bias Monte Carlo (tfMC).

A brief introduction to the tfMC method: During tfMC simulations, atoms are moved iteratively, similar to MD. However, tfMC differs from MD in that it only considers positions and forces, excluding atomic velocities.⁴⁷ In a single tfMC iteration step, each atom undergoes stochastic displacements in the three Cartesian directions ($\delta_v, v=x, y, z$), with a user-selectable parameter $\Delta/2$ defining the range. The displacements (δ_v) are normalized by $\Delta/2$ to ensure they fall within the range of -1 to 1, as shown in the Equation (1):

$$\xi_v = \frac{\delta_v}{\Delta/2} \quad (1)$$

To generate a value for ζ_v , a uniform random number η_v is used, which is sampled from the interval $[0, 1]$, as shown in the Equation (2):

$$\xi_v = \frac{1}{\gamma_v} \ln[\eta_v(e^{|\zeta_v|} - e^{-|\zeta_v|}) + e^{-|\zeta_v|}] \quad (2)$$

where:

$$\gamma_v = \frac{F_v \Delta / 2}{2k_B T} \quad (3)$$

with F_v the v component of the force \mathbf{F} acting on the atom under consideration, k_B the Boltzmann constant, and T the temperature chosen for the tfMC simulation. The tfMC algorithm described above has been successfully implemented in the LAMMPS program⁴⁹.

In the tfMC method, two critical parameters, temperature (T) and maximum allowed

displacement (Δ), need to be carefully chosen. For T, it is important to select a realistic and physically meaningful value. The parameter Δ is determined through a series of simulation tests at different deposition temperatures, following the criterion proposed by Timonova *et al*⁴⁷. This criterion ensures that a perfect crystal remains perfect after tfMC simulation and a short MD equilibration. The parameter values $\Delta = 0.18 \text{ \AA}$ and $T = 573 \text{ K}$ were carefully determined and utilized in our previous work and were also employed in this study.

✓ **Advantages of tfMC compared to other MC methods:**

- ①. **Dynamics and Reliability:** Compared to the Metropolis Monte Carlo (MC) method, tfMC is not strictly a sampling method. In Metropolis MC, the structural phase space is sampled based on well-known equilibrium ensemble distributions, such as the NVT Gibbs petit-canonical ensemble, without considering the role of time as a driving factor. In contrast, tfMC approximates the dynamics of a system regardless of its distance from equilibrium. In tfMC, each system change is driven by a Boltzmann distribution, which differs from classical ensembles but can be viewed as a sequence of instantaneous and local statistical processes. This characteristic enables tfMC to provide more reliable and robust dynamic routes and final states compared to Metropolis MC. As a result, tfMC has a distinct advantage in predicting the reaction pathways of complex reactions. The unique approach of tfMC, focusing on dynamics rather than equilibrium sampling, allows for a more comprehensive understanding of system behavior and enhances its ability to capture intricate reaction mechanisms.
- ②. **Acceptance Rate and Efficiency:** tfMC is not a true sampling method, and its "acceptance rate" is 100%. This characteristic led to its previous name, uniform-

acceptance force-bias Monte Carlo. The simulation efficiency of tfMC is also higher than that of Metropolis MC.

③. **Simplicity and Flexibility:** Compared to another commonly used MC method called kinetic Monte Carlo (KMC), tfMC is a more straightforward approach. KMC requires a prior event table listing all possible kinetic events, which can be challenging to construct in real deposition processes, especially when unpredictable structures or substrate atom movements are involved. In contrast, tfMC does not require such pre-knowledge and allows for more flexible simulations.

④. **Real-time Construction:** While DFT-based KMC offers high accuracy and extended time scales, it is not suitable for simulating carbon growth on metallic surfaces due to the difficulty of constructing an event table in real time. tfMC overcomes this limitation by enabling on-the-fly simulations without the need for predefined event tables.

In summary, tfMC offers advantages in terms of dynamic modeling, reliability, acceptance rate, efficiency, simplicity, and flexibility compared to traditional MC methods like Metropolis MC and KMC.

✓ We have expanded on the details of the MD/tfMC method in the revised Supplementary Information to provide a comprehensive understanding of its implementation. Additionally, we have compared tfMC with other MC methods to highlight the significance of using tfMC in dynamic simulations of deposition processes.

Comment 7: It is unclear how energy/velocity rescaling walls affect the kinetic energy of impeding C atoms.

✓ We appreciate the Reviewer's comment. The main purpose of the energy/velocity rescaling

walls is to prevent the generation of artificial simulation results caused by the repetitive entry of impinging atoms' excessive kinetic energy through periodic boundary conditions⁵⁰. In reality, the surface of a material is considered infinite at the microscopic level. However, in simulations, we use periodic boundary conditions to mimic an infinitely large surface on a finite-sized box. In actual deposition processes, the kinetic energy induced by the impinging atoms dissipates into the surrounding environment. The energy/velocity rescaling walls are implemented to simulate this dissipative process of kinetic energy. They serve to regulate and control the dissipation of kinetic energy, ensuring more realistic and physically meaningful simulation results.

Fig. R31 clearly illustrates the temperature profile during the MD/tfMC deposition simulation. As observed, the velocity/energy walls in the deposition system play a crucial role in maintaining stability. Without them, the excessive energy from impinging atoms would repeatedly enter the system, leading to instability and unrealistic high thermal spike.

Fig. R31 Thermal spike comparison of the deposition simulation with and without the energy/velocity rescaling walls.

- ✓ We have emphasized the importance of the energy/velocity walls in the revised Supplementary Information to provide a better understanding of their roles in

maintaining the system stability.

Comment 8: No explanation is provided why high kinetic energy (10 eV) simulations lead to different carbon cluster growth. It is unclear from the data because in the snapshots it looks like the same amount of Cu adatoms is produced.

- ✓ Thank you for your question. We have performed more analyses to show the two distinct effects of high-energy bombardment has. Firstly, it imparts energy to the copper atoms on the Cu(111) surface, increasing their thermal motion and making them more likely to be adsorbed onto the surface, serving as the active nucleation site. Secondly, direct high-energy bombardment on existing carbon rings results in their pronounced susceptibility to breakage, particularly in the case of isolated carbon rings.

Fig. R32a shows the evolution of hexagon carbon rings and carbon hybridization types during the deposition simulation. At an incident energy of 2.5 eV, the deposited carbon atoms primarily form carbon chains, resulting in a higher proportion of sp-hybridized carbon compared to sp²- or sp³-carbon. With increasing incident energy, the content of sp²-carbon initially increases and then decreases, indicating that high-energy bombardments can induce the breaking of carbon rings into 1-dimensional carbon chains.

Fig. R32b presents the direct evidence of carbon ring fragmentation into carbon chains when subjected to high-energy bombardments. At timestep 0, a carbon ring on a Cu111 surface was subjected to a 10 eV carbon atom bombardment. **Fig. R32c-f** shows that under the impact of high-energy carbon atoms, the existing carbon rings undergo gradual dissociation, leading to the formation of carbon chains. This phenomenon explains why, as depicted in the main text Fig 3, high-energy bombardment does not facilitate the nucleation

of graphene but rather hinders the formation of carbon rings. Additionally, we have included an animated demonstration in the Supplementary Materials (i.e., Carbon_Ring_Breaking.mov), showcasing the powerful capability of the MLP-based MD/tfMC method in capturing the dynamic atomistic processes.

Fig. R32 Carbon structure analysis and observation of carbon ring breakage by high-energy bombardments. (a) Evolution of hexagonal carbon rings and hybridization analysis with increasing deposited carbon atoms. The maximum number of hexagonal carbon rings and optimal sp²-content are achieved at a moderate incident energy of 5 eV. Lower or higher incident energies hinder graphene formation due to insufficient adsorbed Cu atoms or carbon ring breakage, respectively. (b) MLP-based MD/tfMC dynamic simulation capturing

the transformation from carbon rings to chains. Coordination numbers with a 1.8 Å cutoff radius for the nearest neighbors were used to determine the hybridization orbitals

To further illustrate the dual effects of high-energy bombardments, we have made additional statements to Fig. 4 in the main text.

Line 239-257:

“Carbon Ring Breaking. To gain atomistic insight into the observed suppression of graphene island formation at an incident energy of 10 eV in Fig. 3d, we performed further analyses including the evolution of hexagonal carbon rings and hybridization analysis on the carbon atoms deposited on Cu(111) surfaces, as shown in Fig.4a. At an incident energy of 2.5 eV, the deposited carbon atoms primarily form carbon chains, resulting in a higher proportion of sp -hybridized carbon compared to sp^2 - or sp^3 -carbon. With increasing incident energy, the content of sp^2 -carbon initially increases and then decreases, indicating that high-energy bombardments can induce the breaking of carbon rings into carbon chains. Previous DFT studies¹⁸ also indicated that without the adsorbed copper atoms to stabilize the hexagonal carbon rings, they would change to linear chains easily. To provide direct evidence of the process, Fig. 4b presents compelling evidence of carbon ring fragmentation into carbon chains when subjected to high-energy bombardments. The sequence of images begins with a carbon ring on a Cu(111) surface being subjected to a 10 eV carbon atom bombardment (timestep 0). Subsequent images illustrate the gradual dissociation of carbon rings and the formation of carbon chains resulting from the impact of high-energy carbon atoms. This observation aligns with the findings in Fig. 4 of the main text, suggesting that high-energy bombardment impedes the formation of carbon rings and, consequently, the

nucleation of graphene. Additionally, we have included an animated demonstration in the supplementary materials (i.e., Carbon_Ring_Breaking.mov), showcasing the powerful capability of the MLP-based MD/tfMC method in capturing dynamic atomistic processes.”

Comment 9: The soap distance formula and symbols require explanation in Fig 1.

- ✓ Thank you for your comments. In the updated manuscript, we have included an explanation of the inset formula and the corresponding symbols. The revised version is presented below.

Fig. R33 Schematic illustrations of CGM-MLP generated by active learning on-the-fly during MD/tfMC simulations. (a) The initial training dataset includes representative carbon structures from GAP-20¹ and C₁-C₁₈ carbon clusters on Cu(111) surfaces. (b) The CGM-MLP trained from this dataset is then used in a deposition simulation employing a hybrid MD/tfMC method². (c) A SOAP-based algorithm used to select the most representative structures from the MD/tfMC simulations. The inset formula calculates the similarity matrix D^{max} and D^{ave} between the existing training dataset and a new structure, where $\vec{A}(a_i)$ and $\vec{A}(a_j)$ represent the SOAP vectors of atom i and j , respectively. The $\| \cdot \|$ denotes the Euclidean distance between the two SOAP vectors. (see Supplementary Information for more details). (d) DFT benchmarks energy and force, and if the error is below a threshold, MD/tfMC continues. Otherwise, the training dataset is updated with newly selected

structures.

Comment 10: In SI screening parameter is inconsistently termed TS.

- ✓ Thank you for pointing out the inconsistency in the terminology used for the screening parameter in the Supplementary Information. We apologize for the confusion. In the revised version, we have made the necessary changes to ensure consistency, and the screening parameter is now consistently referred to as "**S**" throughout the main text and Supplementary Information.

Comment 11: Fig 3 has bad quality, marker I-V is not explained (timesteps?)

- ✓ Thank you for your comment. In the revised manuscript, the markers I-V are replaced the number of carbon atoms that are already deposited, indicating the progress of the deposition process. The updated Fig.3 is presented in

Comment 12: Language is generally imprecise and ambiguous, e.g. often “well-known truths” or similar problematic expressions are used for experimental observations.

- ✓ Thank you for your feedback. In the revised manuscript, we have made efforts to provide clearer and more specific descriptions, avoiding the use of expressions that may imply "well known truths" without proper experimental evidence or references. We appreciate your comment and have considered it to enhance the clarity and accuracy of our manuscript.

Your comments have helped us to enhance the quality of our work, and we greatly appreciate your effort in helping us improve. We hope that our revisions have satisfactorily addressed your concerns, and we would like to thank you again for your valuable contribution to our work.

Please do not hesitate to contact us if you have any further comments or suggestions. We look forward to hearing from you.

Warm regards,

Corresponding Author: Di Zhang

* E-mail: zhangdi2015@sjtu.edu.cn

Corresponding Author: Hao Li

* E-mail: li.hao.b8@tohoku.ac.jp

Phone: +81-09013980212

Fax: 021-34206304

Sincerely,

Di Zhang (Email: zhangdi2015@sjtu.edu.cn)

Peiyun Yi (Email: yipeiyun@sjtu.edu.cn)

Linfa Peng (Email: penglinfa@sjtu.edu.cn)

Xinmin Lai (Email: xmlai@sjtu.edu.cn)

Hao Li (Email: li.hao.b8@tohoku.ac.jp)

References

1. Rowe P, Deringer VL, Gasparotto P, Csányi G, Michaelides A. An accurate and transferable machine learning potential for carbon. *The Journal of Chemical Physics* **153**, 034702 (2020).
2. Zhang D, Peng L, Li X, Yi P, Lai X. Controlling the Nucleation and Growth Orientation of Nanocrystalline Carbon Films during Plasma-Assisted Deposition: A Reactive Molecular Dynamics/Monte Carlo Study. *Journal of the American Chemical Society* **142**, 2617-2627

(2020).

3. Shu H, Chen X, Tao X, Ding F. Edge Structural Stability and Kinetics of Graphene Chemical Vapor Deposition Growth. *ACS Nano* **6**, 3243-3250 (2012).
4. Jinnouchi R, Miwa K, Karsai F, Kresse G, Asahi R. On-the-Fly Active Learning of Interatomic Potentials for Large-Scale Atomistic Simulations. *The Journal of Physical Chemistry Letters* **11**, 6946-6955 (2020).
5. Jinnouchi R, Lahnsteiner J, Karsai F, Kresse G, Bokdam M. Phase Transitions of Hybrid Perovskites Simulated by Machine-Learning Force Fields Trained on the Fly with Bayesian Inference. *Physical Review Letters* **122**, 225701 (2019).
6. Miwa K, Ohno H. Molecular dynamics study on β -phase vanadium monohydride with machine learning potential. *Physical Review B* **94**, 184109 (2016).
7. Podryabinkin EV, Shapeev AV. Active learning of linearly parametrized interatomic potentials. *Comput Mater Sci* **140**, 171-180 (2017).
8. Bernstein N, Csányi G, Deringer VL. De novo exploration and self-guided learning of potential-energy surfaces. *npj Computational Materials* **5**, 99 (2019).
9. Bartók AP, Payne MC, Kondor R, Csányi G. Gaussian Approximation Potentials: The Accuracy of Quantum Mechanics, without the Electrons. *Physical Review Letters* **104**, 136403 (2010).
10. Botu V, Batra R, Chapman J, Ramprasad R. Machine Learning Force Fields: Construction, Validation, and Outlook. *The Journal of Physical Chemistry C* **121**, 511-522 (2017).
11. Shibuta Y, Arifin R, Shimamura K, Oguri T, Shimojo F, Yamaguchi S. Ab initio molecular dynamics simulation of dissociation of methane on nickel(111) surface: Unravelling initial

- stage of graphene growth via a CVD technique. *Chemical Physics Letters* **565**, 92-97 (2013).
12. Shibuta Y, Arifin R, Shimamura K, Oguri T, Shimojo F, Yamaguchi S. Low reactivity of methane on copper surface during graphene synthesis via CVD process: Ab initio molecular dynamics simulation. *Chemical Physics Letters* **610-611**, 33-38 (2014).
 13. Taioli S. Computational study of graphene growth on copper by first-principles and kinetic Monte Carlo calculations. *Journal of Molecular Modeling* **20**, 2260 (2014).
 14. Fan L, *et al.* Topology evolution of graphene in chemical vapor deposition, a combined theoretical/experimental approach toward shape control of graphene domains. *Nanotechnology* **23**, 115605 (2012).
 15. Jiang H, Hou Z. Large-scale epitaxial growth kinetics of graphene: A kinetic Monte Carlo study. *The Journal of Chemical Physics* **143**, (2015).
 16. Neyts EC, van Duin ACT, Bogaerts A. Formation of single layer graphene on nickel under far-from-equilibrium high flux conditions. *Nanoscale* **5**, 7250-7255 (2013).
 17. Neyts EC, Shibuta Y, van Duin ACT, Bogaerts A. Catalyzed Growth of Carbon Nanotube with Definable Chirality by Hybrid Molecular Dynamics–Force Biased Monte Carlo Simulations. *ACS Nano* **4**, 6665-6672 (2010).
 18. Klaver TPC, Zhu S-E, Sluiter MHF, Janssen GCAM. Molecular dynamics simulation of graphene on Cu (100) and (111) surfaces. *Carbon* **82**, 538-547 (2015).
 19. van Duin ACT, Dasgupta S, Lorant F, Goddard WA. ReaxFF: A Reactive Force Field for Hydrocarbons. *The Journal of Physical Chemistry A* **105**, 9396-9409 (2001).
 20. Yeon J, Adams HL, Junkermeier CE, van Duin ACT, Tysoe WT, Martini A. Development of a ReaxFF Force Field for Cu/S/C/H and Reactive MD Simulations of Methyl Thiolate

- Decomposition on Cu (100). *The Journal of Physical Chemistry B* **122**, 888-896 (2018).
21. Tersoff J. New empirical model for the structural properties of silicon. *Physical Review Letters* **56**, 632-635 (1986).
 22. Donald WB, Olga AS, Judith AH, Steven JS, Boris N, Susan BS. A second-generation reactive empirical bond order (REBO) potential energy expression for hydrocarbons. *Journal of Physics: Condensed Matter* **14**, 783 (2002).
 23. Stuart SJ, Tutein AB, Harrison JA. A reactive potential for hydrocarbons with intermolecular interactions. *The Journal of Chemical Physics* **112**, 6472-6486 (2000).
 24. Liang T, Devine B, Phillpot SR, Sinnott SB. Variable Charge Reactive Potential for Hydrocarbons to Simulate Organic-Copper Interactions. *The Journal of Physical Chemistry A* **116**, 7976-7991 (2012).
 25. Behler J, Parrinello M. Generalized Neural-Network Representation of High-Dimensional Potential-Energy Surfaces. *Physical Review Letters* **98**, 146401 (2007).
 26. Shapeev AV. Moment Tensor Potentials: A Class of Systematically Improvable Interatomic Potentials. *Multiscale Modeling & Simulation* **14**, 1153-1173 (2016).
 27. Caro MA, Deringer VL, Koskinen J, Laurila T, Csányi G. Growth Mechanism and Origin of High sp^3 Content in Tetrahedral Amorphous Carbon. *Physical Review Letters* **120**, No. 166101 (2018).
 28. Deringer VL, *et al.* Computational Surface Chemistry of Tetrahedral Amorphous Carbon by Combining Machine Learning and Density Functional Theory. *Chemistry of Materials* **30**, 7438-7445 (2018).
 29. Qian C, McLean B, Hedman D, Ding F. A comprehensive assessment of empirical potentials

for carbon materials. *APL Materials* **9**, (2021).

30. Shaidu Y, Küçükbenli E, Lot R, Pellegrini F, Kaxiras E, de Gironcoli S. A systematic approach to generating accurate neural network potentials: the case of carbon. *npj Computational Materials* **7**, 52 (2021).
31. Li X, Li L, Zhang D, Wang A. Ab Initio Study of Interfacial Structure Transformation of Amorphous Carbon Catalyzed by Ti, Cr, and W Transition Layers. *ACS Applied Materials & Interfaces* **9**, 41115-41119 (2017).
32. Hao Y, *et al.* The Role of Surface Oxygen in the Growth of Large Single-Crystal Graphene on Copper. *Science* **342**, 720-723 (2013).
33. Wu M, *et al.* Seeded growth of large single-crystal copper foils with high-index facets. *Nature* **581**, 406-410 (2020).
34. Wu T, *et al.* Fast growth of inch-sized single-crystalline graphene from a controlled single nucleus on Cu–Ni alloys. *Nature Materials* **15**, 43-47 (2016).
35. Zhou H, *et al.* Chemical vapour deposition growth of large single crystals of monolayer and bilayer graphene. *Nat Commun* **4**, 2096 (2013).
36. Lee J-H, *et al.* Wafer-Scale Growth of Single-Crystal Monolayer Graphene on Reusable Hydrogen-Terminated Germanium. *Science* **344**, 286-289 (2014).
37. Dean CR, *et al.* Boron nitride substrates for high-quality graphene electronics. *Nature Nanotechnology* **5**, 722-726 (2010).
38. Safina LR, Rozhnova EA, Murzaev RT, Baimova JA. Effect of Interatomic Potential on Simulation of Fracture Behavior of Cu/Graphene Composite: A Molecular Dynamics Study. *Applied Sciences* **13**, 916 (2023).

39. Liang T, *et al.* Classical atomistic simulations of surfaces and heterogeneous interfaces with the charge-optimized many body (COMB) potentials. *Materials Science and Engineering: R: Reports* **74**, 255-279 (2013).
40. Monti S, Li C, Carravetta V. Reactive Dynamics Simulation of Monolayer and Multilayer Adsorption of Glycine on Cu(110). *The Journal of Physical Chemistry C* **117**, 5221-5228 (2013).
41. Li P, Li Z. Theoretical Insights into the Thermodynamics and Kinetics of Graphene Growth on Copper Surfaces. *The Journal of Physical Chemistry C* **124**, 16233-16247 (2020).
42. Gao J, Yip J, Zhao J, Yakobson BI, Ding F. Graphene Nucleation on Transition Metal Surface: Structure Transformation and Role of the Metal Step Edge. *Journal of the American Chemical Society* **133**, 5009-5015 (2011).
43. Wesep RGV, Chen H, Zhu W, Zhang Z. Communication: Stable carbon nanoarches in the initial stages of epitaxial growth of graphene on Cu(111). *The Journal of Chemical Physics* **134**, 171105 (2011).
44. Zhang X, Xu Z, Hui L, Xin J, Ding F. How the Orientation of Graphene Is Determined during Chemical Vapor Deposition Growth. *The Journal of Physical Chemistry Letters* **3**, 2822-2827 (2012).
45. Qiu Z, Li P, Li Z, Yang J. Atomistic Simulations of Graphene Growth: From Kinetics to Mechanism. *Accounts of Chemical Research* **51**, 728-735 (2018).
46. Schran C, Thiemann FL, Rowe P, Müller EA, Marsalek O, Michaelides A. Machine learning potentials for complex aqueous systems made simple. *Proceedings of the National Academy of Sciences* **118**, e2110077118 (2021).

47. Timonova M, Groenewegen J, Thijsse BJ. Modeling diffusion and phase transitions by a uniform-acceptance force-bias Monte Carlo method. *Physical Review B* **81**, 144107 (2010).
48. Mees MJ, Pourtois G, Neyts EC, Thijsse BJ, Stesmans A. Uniform-acceptance force-bias Monte Carlo method with time scale to study solid-state diffusion. *Physical Review B* **85**, 134301 (2012).
49. Thompson AP, *et al.* LAMMPS - a flexible simulation tool for particle-based materials modeling at the atomic, meso, and continuum scales. *Computer Physics Communications* **271**, 108171 (2022).
50. Marks NA, Cover MF, Kocer C. Simulating temperature effects in the growth of tetrahedral amorphous carbon: The importance of infrequent events. *Applied Physics Letters* **89**, 131924 (2006).

Reviewers' Comments:

Reviewer #1:

Remarks to the Author:

The authors have responded adequately to my questions and concerns, and I found the revised manuscript to be much more clearly written. A few minor points on the rebuttal:

* The authors should describe what is being depicted in the parity plot in the revised Fig. 1c. There is no description of this plot in the figure caption.

* In Fig. S5, it would be helpful to scale the vertical axes so that the energy threshold of 0.05 eV/atom and the force threshold of 0.5 eV/Å are aligned, and to draw a horizontal line at the threshold to make it clear when the threshold has been exceeded.

* Only D_{ave} is defined in Fig. 1c, but the revised text states that both D_{ave} and D_{max} are defined in the figure (line 127). Both quantities should be defined somewhere, perhaps in the Methods section.

* In the rebuttal and revised text, the authors draw a distinction between "prior" and "posterior" methods for quantifying uncertainty that I am unfamiliar with. These terms are of course well-defined within the domain of Bayesian probability and statistics, but I haven't seen them used to classify different approaches to uncertainty quantification. If this terminology has been used in other works, the authors should cite them, otherwise they should define the terms more clearly.

Reviewer #2:

Remarks to the Author:

Authors have improved the manuscript significantly by effectively addressing the main and minor points I have previously raised. In a proactive approach, they have added further simulations that covers the graphene deposition process on copper oxide as well as other metallic substrates to show the effect of the type and contamination of the growth substrate. They have also introduced newly trained GAP models they have used to address these problems. Authors also clarified the novelty and contributions of this work as compared to the relevant literature. Taken together, I would recommend the publication of this manuscript in Nature Comm without further revisions.

Reviewer #3:

Remarks to the Author:

In this revision the authors have added previously elusive methodological details as well as new tests, analysis, and motivation which they presented in a very wordy and lengthy (92 pages) reply-letter. The new data includes extended analysis (e.g. of MLP error analysis and learning curves, ring breaking quantification) and additional test-cases (i.e. Cr, Ti, O-contaminated Cu) that delivered important information to judge the quality of the work and to demonstrate the versatility of active learning approaches via MLP. Following mine and another reviewers' criticism of accuracy, the authors also retrained their potential to achieve moderately improved accuracy based on their active learning approach.

Despite the authors' efforts, I still deem their MLP too inaccurate as evident in the mean average errors (MAEs), regularization parameters and most likely (see below) in the computed reaction pathways, in which they, importantly, fall behind state-of-the-art approaches (including works based on GAPs). The goal of the presented active learning approach is a purpose-driven MLP for the prediction of the carbon structure growth on metal catalysts for which reaction energetics need to be described with high accuracy to discriminate a viable mechanism. The demonstrated accuracy (which is still not flawlessly reported) does not support the fact that this goal has been reached within this work. Further, this inaccuracy is not reflected in the written language since self-critical statements and honest evaluation is absent, albeit previous statements commending a

high accuracy have been weakened. Unfortunately, the lack of accuracy is especially problematic since the authors (now) motivate this work mainly with their MLP iterative learning approach due to the fact that reviewer 2 commented that the scientific insight is not new. Out of all these reasons, I still cannot recommend this manuscript for publication.

I would like to amend, that I still appreciate the authors' efforts in applying MLPs (which is not trivial!) but recognize (through the now provided data) a missing expertise in applying them to reproduce DFT data within almost chemical accuracy. It needs to be stressed that the latter, however, is indeed achievable due to the careful standards developed by the MLP community in the last decade. More work needs to be invested to reach (and communicate) this goal correctly.

Please see below detailed comments about the above mentioned and further issues:

Comments about accuracy and training procedure:

- The stated MAEs of 0.013 eV/atom and 0.43 eV/Å only give a rough understanding of the accuracy. They should still be reported as they reflect on the used regularization (and other hyper-) parameters. The most important demonstration is the validation of the NEB calculations against DFT data as this comparison directly shows the accuracy in reaction energies and barriers. Using per-atom energies are extremely misleading and hide the actual errors, I would believe the total reaction energies and barriers need to be compared to judge accuracy. From the energy differences of ~0.05 eV/atom in Figure 4b&c one, can deduce errors of about 0.4 eV (C8 structure in figure 4b,c) and an error of 0.5 eV in Figure S6 can be made out (here total energies are reported). These deviations are extremely large and thus problematic if accurate reaction energies are targeted. Note that state-of-the art approaches reach < 0.025 eV and < 0.05 eV/Å accuracy along total NEB reaction pathways in similar applications (see <https://arxiv.org/abs/2301.09931>)
- Possible reasons for the low accuracy are likely the hyperparameters: (a) Using global energy and force regularization parameters of 0.05 eV/atom & 0.5 eV/Å may be a major cause of the inaccuracy (and is bad practice). The authors may note, that it is possible to assign configuration and even atom-based regularization parameters in GAP. (b) The used cutoff of 3.7 Å is really small and even in Deringer's work, which the authors mainly refer to, a longer cutoff of 4.5 Å was used to (at least) correctly describe graphitic structures. (c) the large number of sparse-points may affect accuracy only minorly but definitely impedes on the GAP performance! In figure S2, it can be seen that the authors should probably choose 5000 instead of 9000 sparse points, since they do not gain any accuracy but sacrifice speed by factor 3. This practice counters the argument of "a compromise of tighter convergence and computational cost". All listed points on hyperparameters (a-c) are an example of an inadequate application of GAP and cannot correspond to a model-active-learning-approach. Especially, main tweaks that lead to a peak performance should be conveyed by the authors in such a publication if they want to make their approach the paper's main message.
- Considering the authors comparison to the MAE's in energy and forces to Deringers general GAP-20 work to justify high errors, I would like to note that this comparison is inadequate since the GAP-20 MLP describes all allotropes of carbon and with this a much larger phase space. Note also, that Deringer et al. carefully validate the correct description of local minima in great detail to show that they actually achieved a practical accuracy that is much higher than the MAEs.
- In the paper, the context of MLP training and production runs is not clearly defined or rather distinguished. Especially, table R1 is confusing, since it suggests that all "results" are based on the same final MLP and hence the MLP is not really an "on-the-fly" procedure where the MLP training and production occurs at the same time. From the latter I previously understood that the last iteration of each carbon number training cycle also corresponds to those specific results and that the MLP is different for every type of simulation.

Minor comments:

- The arrow in Fig 1 pointing back to the initial training set is confusing, I think it needs to be removed?
- Fig 2 still has I-V numbering, that may be misinterpreted, maybe an instead use an arrow "progressing simulation" above/below. The label's "CX" with X=number of deposited carbon atoms needs to be explained.
- The formula for the "mean average force error" (MAE) is still erroneous (no sum over N). Also, the formula for the energy error should be given. In the text the error should be labelled MAE.

- Description of time-stamped MC method is incomplete, exponent " r_v " is not explained
- "During tfMC simulations, atoms are moved iteratively, similar to MD" is a problematic sentence as the movement is fundamentally different in both methods and "iteratively" is very ambiguous.
- For the explanation of the "walls", I think you need to explain the geometry of the walls, e.g. "normal to the surface plane", or "along xz, and yz"? To make their function clear.

Dear Reviewers,

Thank you for taking the time to review our paper. Your feedback and suggestions are highly appreciated, and we are grateful for your effort in helping us improve our work. We have carefully considered your comments and suggestions, and we have made the changes to address the issues you have raised to the best of our abilities. We would like to respond to each of your comments in detail.

To Reviewer #1:

The authors have responded adequately to my questions and concerns, and I found the revised manuscript to be much more clearly written.

A few minor points on the rebuttal:

1. The authors should describe what is being depicted in the parity plot in the revised Fig. 1c. There is no description of this plot in the figure caption.

✓ Thank you for pointing out missing the description of the parity plot in the revised Fig. 1c. We have revised the figure caption to include a comprehensive description of the parity plot. Here's a revised caption for Fig. 1:

Fig. 1 Schematic illustrations of CGM-MLP generated by active learning on-the-fly during MD/tfMC simulations. (a) The initial training dataset includes representative carbon structures from GAP-20¹ and C₁-C₁₈ carbon clusters on Cu(111) surfaces. (b) The CGM-MLP trained from this dataset is then used in a deposition simulation employing a hybrid MD/tfMC method.² (c) A SOAP-based algorithm used to select the most representative structures from the MD/tfMC simulations. The inset figure presents the force correlation plots by using different quality control parameters, namely N_f , S^{max} , and S^{ave} . The inset formula calculates the similarity matrix

\mathbf{D}^{ave} between the existing training dataset and a new structure, where $\vec{A}(a_i)$ and $\vec{A}(a_j)$ represent the SOAP vectors of atom i and j , respectively. The $\| \cdot \|$ denotes the Euclidean distance between the two SOAP vectors. (see **Method** for the similarity matrix \mathbf{D}^{max}). (d) DFT benchmarks energy and force, and if the MAE is below a threshold, MD/tfMC continues. Otherwise, the training dataset is updated with newly selected structures.

2. In Fig. S5, it would be helpful to scale the vertical axes so that the energy threshold of 0.05 eV/atom and the force threshold of 0.5 eV/Å are aligned, and to draw a horizontal line at the threshold to make it clear when the threshold has been exceeded.

- ✓ Thank you for your suggestion regarding Fig. S5. We have adjusted the scales for clarity and added a horizontal line to indicate the specified thresholds. Please refer to the updated version of Fig. S5 (Fig. R1) for these enhancements.

Fig. R1. Energy and force errors over 20 iterations during the training process. A criterion of less than 0.05 eV/atom for energy and less than 0.5 eV/Å for force (highlighted by the horizontal dashed line) determines whether newly generated structures are integrated into the

training set or if the deposition of carbon atoms can continue without additional training. With

$N_f = 20$, S^{max} , $S^{ave} = 0.08$, approximately 40~60 structures will be added to the training set.

3. Only D_{ave} is defined in Fig. 1c, but the revised text states that both D_{ave} and D_{max} are defined in the figure (line 127). Both quantities should be defined somewhere, perhaps in the Methods section.

✓ Thank you for your suggestion. We have elaborated on both quantities in the Methods section to ensure a clear and thorough description. Please refer to the revised content in the Methods section for more details.

“Throughout the training procedure, we introduce two SOAP-based screening parameters, D^{max} and D^{ave} , specifically designed to optimize the training set. Their respective equations are presented below:

$$D^{ave} = ave_{i < N_1} \left(\min_{j < N_0} \left(\|\vec{A}(a_i), \vec{A}(a_j)\| \right) \right) \text{ and } D^{max} = \max_{i < N_1} \left(\min_{j < N_0} \left(\|\vec{A}(a_i), \vec{A}(a_j)\| \right) \right).”$$

✓ Additionally, we've revised the statement at line 127 to read: *"In Fig. 1c, the equation delineates the similarity measure D^{ave} . Notably, this study introduces an additional measure, D^{max} , which, though resembling D^{ave} in form (detailed in the Methods section), plays a pivotal role in refining the on-the-fly training set. These measures specifically capture the maximum and average SOAP-based distances between atoms in newly observed and previously chosen structures, respectively."*

4. In the rebuttal and revised text, the authors draw a distinction between "prior" and "posterior" methods for quantifying uncertainty that I am unfamiliar with. These terms are of course well-defined within the domain of Bayesian probability and statistics, but I haven't seen them used to

classify different approaches to uncertainty quantification. If this terminology has been used in other works, the authors should cite them, otherwise they should define the terms more clearly.

- ✓ Thank you for highlighting the potential ambiguity in our terminology. Our intent in using "prior" and "posterior" to differentiate methods of uncertainty quantification was indeed inspired by Bayesian theorem. In the Bayesian framework, the prior distribution for target data, such as potential energy, is typically assumed to be a multidimensional Gaussian distribution. Consequently, the posterior distribution also follows a Gaussian pattern. The uncertainty in the prediction can therefore be inferred from the variance of this posterior distribution.³ Given this background, we classified such methodologies as "posterior methods."

Conversely, there are alternative methods that aim to maximize linear independence among feature vectors.⁴ These strategies determine uncertainty only from the descriptor of atomic environments, without utilizing observed potential surfaces. Such techniques deliberately avoid the Gaussian assumption inherent in Bayesian approaches. In our initial discussions, we have labeled these as "prior methods" to contrast them with the posterior distribution of Bayesian inference.

- ✓ However, given your insightful feedback and the potential overlap with established Bayesian terminologies of prior and posterior probabilities, we've chosen to avoid the terms "prior" and "posterior" when describing these uncertainty prediction methods in our revised version:

“So far, several selective principles have been proposed to construct a training set with predictive uncertainties, including the Bayesian inference,^{4, 5} spilling factor,⁶ Maxvol method,⁷ and CUR decomposition.⁸ Notably, the CUR decomposition method stands out because it does not rely on a prior assumption of a Gaussian distribution for potential energy, as is often the case in Bayesian

inference. Additionally, with its foundation in a configuration-averaged metric, the CUR decomposition method has consistently shown impressive accuracy and robustness, especially when applied to carbon-based and metal materials.⁸”

We deeply appreciate your comments and insights that help us present our work with more clarity and precision.

To Reviewer #2:

Authors have improved the manuscript significantly by effectively addressing the main and minor points I have previously raised. In a proactive approach, they have added further simulations that covers the graphene deposition process on copper oxide as well as other metallic substrates to show the effect of the type and contamination of the growth substrate. They have also introduced newly trained GAP models they have used to address these problems. Authors also clarified the novelty and contributions of this work as compared to the relevant literature. Taken together, I would recommend the publication of this manuscript in Nature Comm without further revisions.

- ✓ Thank you for your positive feedback on our revised manuscript. We're pleased that our added simulations, newly trained GAP models, and clarifications on our work's novelty were well-received. Your recommendation for publication in Nature Communications is greatly appreciated.

To Reviewer #3:

In this revision the authors have added previously elusive methodological details as well as new tests, analysis, and motivation which they presented in a very wordy and lengthy (92 pages) reply-letter. The new data includes extended analysis (*e.g.* of MLP error analysis and learning curves, ring breaking quantification) and additional test-cases (*i.e.* Cr, Ti, O-contaminated Cu) that delivered important information to judge the quality of the work and to demonstrate the versatility of active learning approaches via MLP. Following mine and another reviewers' criticism of accuracy, the authors also retrained their potential to achieve moderately improved accuracy based on their active learning approach.

Despite the authors' efforts, I still deem their MLP too inaccurate as evident in the mean average errors (MAEs), regularization parameters and most likely (see below) in the computed reaction pathways, in which they, importantly, fall behind state-of-the-art approaches (including works based on GAPs). The goal of the presented active learning approach is a purpose-driven MLP for the prediction of the carbon structure growth on metal catalysts for which reaction energetics need to be described with high accuracy to discriminate a viable mechanism. The demonstrated accuracy (which is still not flawlessly reported) does not support the fact that this goal has been reached within this work. Further, this inaccuracy is not reflected in the written language since self-critical statements and honest evaluation is absent, albeit previous statements commending a high accuracy have been weakened. Unfortunately, the lack of accuracy is especially problematic since the authors (now) motivate this work mainly with their MLP iterative learning approach due to the fact that reviewer 2 commented that the scientific insight is not new. Out of all these reasons, I still cannot recommend this manuscript for publication.

I would like to amend, that I still appreciate the authors' efforts in applying MLPs (which is not trivial!) but recognize (through the now provided data) a missing expertise in applying them to reproduce DFT data within almost chemical accuracy. It needs to be stressed that the latter, however, is indeed achievable due to the careful standards developed by the MLP community in the last decade. More work needs to be invested to reach (and communicate) this goal correctly.

Please see below detailed comments about the above mentioned and further issues:

Comments about accuracy and training procedure:

- The stated MAEs of 0.013 eV/atom and 0.43 eV/Å only give a rough understanding of the accuracy. They should still be reported as they reflect on the used regularization (and other hyper-) parameters. The most important demonstration is the validation of the NEB calculations against DFT data as this comparison directly shows the accuracy in reaction energies and barriers. Using per-atom energies are extremely misleading and hide the actual errors, I would believe the total reaction energies and barriers need to be compared to judge accuracy. From the energy differences of ~0.05 eV/atom in Figure 4b&c one, can deduce errors of about 0.4 eV (C8 structure in figure 4b,c) and an error of 0.5 eV in Figure S6 can be made out (here total energies are reported). These deviations are extremely large and thus problematic if accurate reaction energies are targeted. Note that state-of-the-art approaches reach < 0.025 eV and < 0.05 eV/Å accuracy along total NEB reaction pathways in similar applications (see <https://arxiv.org/abs/2301.09931>)

✓ Thank you for your insights and feedback. We understand your concerns regarding the accuracy of our machine learning potential (MLP). However, we would like to emphasize that assessing the accuracy of a MLP without considering its specific application may overlook important details. We believe there isn't a one-size-fits-all benchmark for state-of-the-art accuracy due to

the unique nature of each system. Using accuracy standards from one system as a measure for another might not always be appropriate or directly comparable, especially for graphite-like and low-density amorphous structures.

A fundamental assumption underlying all interatomic potentials, including MLPs and other empirical potentials, is that of locality: the energy associated with a specific atom or bond relies on its nearby surroundings, but excludes atoms beyond a specified cutoff radius (r_{fix}). A study published in Physical Review B (2017, 95(9), 094203) thoroughly examined the locality of various crystalline carbon allotropes, including graphite and diamond, as well as amorphous carbon (a-C) structures.

As shown in **Fig. R2a**, all atoms outside r_{fix} were displaced randomly with a standard deviation of 0.1 Å. The fundamental assumption of interatomic potentials implies that atomic movements beyond r_{fix} do not have any impact on the calculated force of the central atom. However, the inherent nonlocality of quantum-mechanical models such as DFT can capture these atomic movements, leading to discrepancies between DFT and MLPs.

Fig. R2b highlights the distinct behaviors of diamond and graphite. Diamond exhibits pronounced locality, with minimal force deviations resulting from displacements beyond specific spheres, which ultimately converge to nearly zero at $r_{fix} = 5.5$ Å. In stark contrast, graphite showcases substantial nonlocality, featuring significantly larger force deviations compared to diamond (approximately 0.9 eV/Å) that exhibit slower decay.

Amorphous carbon aligns with the crystalline patterns: the sp^2 -rich form (2.0 g/cm³) displays diminished locality. Even ta-C (3.0 g/cm³) retains a notable level of nonlocality, as depicted in **Fig. R2e**. Despite that the cutoff radius increases to 7.0 Å, the force errors persist

around 0.6 eV/\AA and 0.3 eV/\AA . While a further increase in the cutoff radius might be conceivable, the trade-off of significantly heightened computational expenses, both in training and applying the GAP, appears to lack justification.

Fig. R2. Locality tests for carbon-based structures. (a) Schematic overview of the procedure used here for locality tests. (b), (c) Results for diamond and graphite, respectively, obtained by displacing all atoms outside r_{fix} randomly and inspecting the standard deviation of the force on the central atom. Force locality in (d) low and (e) high-density forms, evaluated in the presence of a small distortion that preserves the major topological features of the amorphous network.

Therefore, the precision achieved by the preprint article with accuracies of $< 0.025 \text{ eV}$ and $< 0.05 \text{ eV/\AA}$ (as outlined in <https://arxiv.org/abs/2301.09931>) is not directly comparable to the precision of our system. Their high accuracy can be attributed to two main factors. Firstly, their systems do not contain graphite-like structures or amorphous carbon structures. Secondly, preprint studies primarily focus on the catalytic reduction of carbon dioxide, and the reaction pathway is more clearly defined compared to the growth of graphene on metal surfaces as

studied in this paper. **Regarding potential functions that account for graphite-like or amorphous carbon structures, we believe the CGM-MLP has already achieved state-of-the-art accuracies.** Therefore, we have added more statements to evaluate the energy and force error for complex carbon structures. *"Given the intrinsic challenges associated with complex hybridizations in carbon and the long-range interactions beyond the MLPs' cutoff, the energy and force errors we observed are believed to approach the peak accuracy achieved by MLPs for systems containing amorphous or defective carbon^{34,50}".*

- ✓ **The reason for using per-atom energies:** In **Fig. R3** (also referred to as Figure 4c in the main manuscript), we compared the minimum energy pathways for Cu(111) graphene growth, both with and without Cu-atom passivated edges. In **Fig. R3**, the dashed line depicts the energy path when a single carbon atom is incorporated. Yet, in the growth process involving Cu, three carbon atoms are added to the graphene edge at once. For a fair comparison, we used per-atom energy, emphasizing the lowered energy barrier in contrast to the growth process without Cu. Presenting the total reaction energies could lead to an imbalanced comparison when varying numbers of carbon atoms are involved in the reaction. Additionally, in response to Reviewer 1's previous suggestions, there was an emphasis on maintaining consistent y-axis labels for enhanced clarity in Figure 4. Consequently, all sub-figures within Figure 4 are presented using per-atom energy.

Fig. R3. Graphene growth with passivated edges by Cu atoms.

✓ We are also very grateful to the reviewer for pointing out that in the previous version of the supplementary material, we did not use the latest trained MLPs to calculate the energies in **Fig. S6**. In the updated version of the **Supplementary Information**, we employed the newly trained Cu-C MLPs (specifically, the CGM-MLP with $N_f = 20$ as shown in **Fig. R4**) to compute the relevant energies, leading to a revised **Fig. S6** (also labeled as **Fig. R4**). Evidently, the Cu-C CGM-MLP with a larger N_f value exhibits markedly improved accuracy. Given this, we are confident that the well trained CGM-MLP can accurately predict the crucial energy pathways for graphene's growth on the Cu(111) surface.

Fig. R4. The role of Cu adatoms in stabilizing the early stages of graphene nucleation. (a)

Potential energies tracked during the off-surface movement of a singular Cu atom. **(b)** The

binding energy of the C8 ring on the Cu(111) surface in the presence of varying Cu adatoms.

- Possible reasons for the low accuracy are likely the hyperparameters: (a) Using global energy and force regularization parameters of 0.05 eV/atom & 0.5 eV/Å may be a major cause of the inaccuracy (and is bad practice). The authors may note, that it is possible to assign configuration and even atom-based regularization parameters in GAP. (b) The used cutoff of 3.7 Å is really small and even in Deringer's work, which the authors mainly refer to, a longer cutoff of 4.5 Å was used to (at least) correctly describe graphitic structures. (c) the large number of sparse-points may affect accuracy only minorly but definitely impedes on the GAP performance! In figure S2, it can be seen that the authors should probably choose 5000 instead of 9000 sparse points, since they do not gain any accuracy but sacrifice speed by factor 3. This practice counters the argument of "a compromise of tighter convergence and computational cost". All listed points on hyperparameters (a-c) are an example of an inadequate application of GAP and cannot correspond to a model-active-learning-approach. Especially, main tweaks that lead to a peak performance should be conveyed by the authors in such a publication if they want to make their approach the paper's main message.

- ✓ We sincerely appreciate the insights regarding hyperparameter selections in MLP training. Based on the previous study⁹, while tuning hyperparameters can influence outcomes, the gains in force prediction accuracy might be minimal when dealing with graphite-like or amorphous carbon structures. We also understand that the flexibility of GAP in allowing for configuration-specific and atom-centered regularization parameters. However, for systems with non-local characteristics, like graphite-like structures and amorphous carbon, employing an excessively restrictive threshold of force error could greatly hinder the training convergence and efficiency.
- ✓ In the study by Deringer *et al.*¹, a cutoff of 4.5 Å and a sparse value of 9000 were used for the GAP-20. As they previously indicated, a sparse value of 9000 represents a tightly-converged

choice for pure carbon potential. However, our training database incorporated all the carbon structures from GAP-20 and additional on-the-fly Cu-C configurations. As illustrated in Figure S2, for the Cu-C configurations, force error drops as the sparse number rises, plateauing when the sparse value exceeds 5000. Therefore, to cater to both carbon-centric structures and Cu-C configurations, we settled on a sparse number of 9000.

- ✓ Regarding the cutoff radius, our simulations mainly focus on graphene growth on varied metal surfaces. Given that stacked graphite structures are rarely observed in our simulations, we opted to reduce the cutoff from 4.5 Å to 3.7 Å. This adjustment aims to enhance the precision of short-range interactions. In Figure S2, it reveals that the force accuracy for the Cu-C system deteriorates when the cutoff surpasses 3.7 Å. Even with this reduction, the 3.7 Å cutoff remains larger than the typical 3.3 Å spacing between graphitic layers. In addition, **Fig. R5** presents that CGM-MLPs trained with two group of parameters displayed comparable accuracies, with energy and force MAE of approximately 0.013 eV and 0.43 eV/Å, respectively.

Fig. R5. Comparative analysis of accuracy for the CGM-MLP trained with varying parameters. (a) Force correlation (b) Energy correlation.

- ✓ While fine-tuning hyperparameters may be beneficial, the key to elevating force prediction

accuracy is understanding the nonlocal interactions of carbon structures. Merging these long-distance interactions into the existing MLPs framework is certainly challenging. A detailed exploration of this topic is actually beyond the scope of our present study, yet we acknowledge its significance.

- Considering the authors comparison to the MAE's in energy and forces to Deringers general GAP-20 work to justify high errors, I would like to note that this comparison is inadequate since the GAP-20 MLP describes all allotropes of carbon and with this a much larger phase space. Note also, that Deringer et al. carefully validate the correct description of local minima in great detail to show that they actually achieved a practical accuracy that is much higher than the MAEs.

- ✓ We truly appreciate your insightful feedback. The GAP-20 MLP indeed stands out by offering a comprehensive representation, encompassing all allotropes of carbon. Inspired by this, our CGM-MLP has been carefully constructed on the solid foundation provided by GAP-20 and extends its training set. Consequently, our model includes all the structural phase space inherent to GAP-20.

We chose to compare with GAP-20 because both our systems encompass graphite-like or amorphous carbon structures. In such contexts, the force error is profoundly influenced by the locality of the structure due to the interaction cutoff. Pertaining to the practical accuracy, our simulations—which shed light on the growth process of graphene on the Cu(111) surface—seem to resonate with a range of experimental observations and prior DFT studies. We believe that such alignment speaks to the practical accuracy of our CGM-MLP.

- In the paper, the context of MLP training and production runs is not clearly defined or rather distinguished. Especially, table R1 is confusing, since it suggests that all “results” are based on the

same final MLP and hence the MLP is not really an “on-the-fly” procedure where the MLP training and production occurs at the same time. From the latter I previously understood that the last iteration of each carbon number training cycle also corresponds to those specific results and that the MLP is different for every type of simulation.

- ✓ We sincerely apologize for any confusion caused by our description of the production runs. To clarify, the production runs can be bifurcated into two main stages:

Stage 1: Screen Parameter Validation: The first stage involves using different screen parameters (S^{max} , S^{ave}) to generate distinct MLPs. The goal here is to evaluate the impact of these screen parameters. Our approach in this stage is based on an understanding: Structures chosen during the on-the-fly training process by more lenient screen parameters (larger values) will undoubtedly be selected by stricter ones (smaller values). For instance, in **Table S1**, the "Deposited-Carbon@Cu" group was chosen using the strictest screen parameters (*i.e.*, S^{max} , $S^{ave}=0.06$). Based on this set of structures, we employed more lenient screen parameters (e.g., $S^{max}=0.40$, $S^{ave}=0.08$) to derive different training sets. These sets, in turn, were utilized to train various MLPs and the results are shown in **Fig. S3** and **Fig. S4**. A significant benefit of this method is the elimination of potential influence due to the randomness of MD/tfMC simulations in production runs. For the other production runs aiming to evaluate the influence of N_f and testing the applicability of the CGM-MLP to other metallic surfaces, we performed the on-the-fly training process. This was deemed necessary since these training sets are distinct and not a subset of any pre-existing training set.

Stage 2: MD/tfMC Validation: In the subsequent stage, we employed the well trained CGM-MLPs to conduct MD/tfMC simulations, altering the random seed numbers and incident

energies. This step ensures that the reliability of our CGM-MLPs is verified, as shown in the Supporting Information **Section 8**.

- ✓ In our revised Supporting Information, we have made additional clarifications concerning the production runs, especially those pertaining to the validation of screen parameters. We hope these enhancements further elucidate our methodology and address any previous ambiguities. *“Our approach here is based on an understanding: Structures chosen during the on-the-fly training process by more lenient screen parameters (larger values) will undoubtedly be selected by stricter ones (smaller values). For instance, in **Table S1**, the "Deposited-Carbon@Cu" group was chosen using the strictest screen parameters (i.e., S^{max} , $S^{ave}=0.06$). Based on this set of structures, we employed more lenient screen parameters (e.g., $S^{max}=0.08\sim 0.40$, $S^{ave}=0.08$) to derive different training sets. These sets, in turn, were utilized to train various MLPs and the results of energy error are shown in **Fig. S3** and **Fig. S4**. A significant benefit of this method is the elimination of potential influence due to the randomness of MD/tfMC simulations.”*

Minor comments:

- The arrow in Fig 1 pointing back to the initial training set is confusing, I think it needs to be removed?

✓ Thank you for pointing that out. We understand how it might be confusing and have made necessary adjustments to enhance clarity (see Fig. R6).

Fig. R6. Schematic illustrations of CGM-MLP generated by active learning on-the-fly during MD/tfMC simulations.

- Fig 2 still has I-V numbering, that may be misinterpreted, maybe an instead use an arrow “progressing simulation” above/below. The label’s “CX” with X=number of deposited carbon atoms needs to be explained.

- ✓ Thank you for your constructive feedback. We have replaced the I-V with an arrow indicating “progressing deposition” above the graph, as you suggested. This should provide a clearer representation. For the labels "C_X", where X represents the number of deposited carbon atoms. We have added a detailed explanation in the figure caption to ensure clarity for readers.

Fig. R7. CGM-MLP driven simulations of graphene growth on Cu(111) with different carbon incident energies. *i.e.*, (a) 2.5 eV, (b) 5.0 eV, (c) 7.5 eV, and (d) 10 eV. Carbon atoms are colored black, and copper atoms are color-coded according to their height coordination. For the labels "C_X", "X" denotes the number of deposited carbon atoms.

- The formula for the “mean average force error” (MAE) is still erroneous (no sum over N). Also,

the formula for the energy error should be given. In the text the error should be labelled MAE.

- ✓ Thank you for bringing this to our attention. We have corrected the formula for the mean absolute error (MAE) to include the summation over N (see equation (3)). Additionally, we'll provide the formula for the energy error in the revised Support Information ((see equation (4)). We have ensured that the error is consistently labeled as 'MAE' throughout the text and figures.

Here is the revised version:

“The force errors are defined as the MAE of the force components along three directions (x, y, and z) for all structures in the training set, as shown in the following equation (3):

$$F_{MAE} = \sum_{j=1}^S \frac{\sum_{i=1}^{N_j} |F_{x,i}^{DFT} - F_{x,i}^{MLP}| + |F_{y,i}^{DFT} - F_{y,i}^{MLP}| + |F_{z,i}^{DFT} - F_{z,i}^{MLP}|}{3N_j} / S \quad (3)$$

where $F_{x,i}$, $F_{y,i}$, and $F_{z,i}$ represent the x, y, and z components of the force. These components are obtained from both DFT calculations and other force fields, which encompass the MLP and classical empirical potentials. N represents the total number of atoms within a given structure, while S denotes the total number of structures present in the training set.

The energy errors are quantified as the MAE between the energies derived from both DFT calculations and the MLP model, as shown in the equation (4).

$$E_{MAE} = \sum_{i=1}^S \frac{|E_i^{DFT} - E_i^{MLP}|}{S} \quad (4)$$

- Description of time-stamped MC method is incomplete, exponent “r_v” is not explained.
- ✓ Thank you for pointing out the oversight regarding the time-stamped MC method. The variable r_v serves simply as an intermediary in the equation and doesn't hold intrinsic physical meaning. To avoid causing unnecessary misunderstandings, we have deleted the variable r_v and modified the formula accordingly.

$$\xi_v = \frac{2k_B T}{F_v \Delta / 2} \ln \left[\eta_v \left(e^{\frac{|F_v \Delta / 2|}{2k_B T}} - e^{-\frac{|F_v \Delta / 2|}{2k_B T}} \right) + e^{-\frac{|F_v \Delta / 2|}{2k_B T}} \right] \quad (5)$$

where F_v is the v component of the force \mathbf{F} acting on the atom under consideration, k_B the Boltzmann constant, η_v is a uniform random number, and T the temperature chosen for the tfMC simulation.

- “During tfMC simulations, atoms are moved iteratively, similar to MD” is a problematic sentence as the movement is fundamentally different in both methods and “+” is very ambiguous.
- ✓ Thank you for pointing out the ambiguity in our statement. We have revised the sentence to provide a clearer and more accurate comparison between the two methods, ensuring the distinct mechanisms of each are appropriately represented.

“During tfMC simulations, in each step, all atoms in the selected group are displaced using the stochastic tfMC algorithm¹, which is designed to sample the canonical (NVT) ensemble at the temperature. Although tfMC is a Monte Carlo algorithm and thus strictly speaking does not perform time integration, it is similar in the sense that it uses the forces on all atoms in order to update their positions. In a single tfMC iteration step, each atom undergoes stochastic displacements in the three Cartesian directions ($\delta_v, v=x, y, z$), with a user-selectable parameter $\Delta/2$ defining the range.”

- For the explanation of the “walls”, I think you need to explain the geometry of the walls, e.g. “normal to the surface plane”, or “along xz, and yz”? To make their function clear.
- ✓ Thank you for your valuable feedback. To address this, we have provided two schematic diagrams in the Supporting Information, as depicted in Fig R8a and R8b (representing the side and top view, respectively). These diagrams depict the wall as a circular ring with a thickness of 2Å. The axis of symmetry of this wall is perpendicular to the deposition surface.

Fig. R8. The impact of the energy/velocity wall. (a) Side view, (b) Top view, and (c) A comparative analysis of thermal spikes in deposition simulations, highlighting differences with and without the energy/velocity rescaling walls.

Your comments have helped us to enhance the quality of our work, and we greatly appreciate your effort in helping us improve. We hope that our revisions have satisfactorily addressed your concerns, and we would like to thank you again for your valuable contribution to our work.

Reference

1. Rowe P, Deringer VL, Gasparotto P, Csányi G, Michaelides A. An accurate and transferable machine learning potential for carbon. *The Journal of Chemical Physics* **153**, 034702 (2020).
2. Zhang D, Peng L, Li X, Yi P, Lai X. Controlling the Nucleation and Growth Orientation of Nanocrystalline Carbon Films during Plasma-Assisted Deposition: A Reactive Molecular

Dynamics/Monte Carlo Study. *Journal of the American Chemical Society* **142**, 2617-2627 (2020).

3. Bishop CM, Nasrabadi NM. *Pattern recognition and machine learning*. Springer (2006).
4. Jinnouchi R, Miwa K, Karsai F, Kresse G, Asahi R. On-the-Fly Active Learning of Interatomic Potentials for Large-Scale Atomistic Simulations. *The Journal of Physical Chemistry Letters* **11**, 6946-6955 (2020).
5. Jinnouchi R, Lahnsteiner J, Karsai F, Kresse G, Bokdam M. Phase Transitions of Hybrid Perovskites Simulated by Machine-Learning Force Fields Trained on the Fly with Bayesian Inference. *Physical Review Letters* **122**, 225701 (2019).
6. Miwa K, Ohno H. Molecular dynamics study on β -phase vanadium monohydride with machine learning potential. *Physical Review B* **94**, 184109 (2016).
7. Podryabinkin EV, Shapeev AV. Active learning of linearly parametrized interatomic potentials. *Comput Mater Sci* **140**, 171-180 (2017).
8. Bernstein N, Csányi G, Deringer VL. De novo exploration and self-guided learning of potential-energy surfaces. *npj Computational Materials* **5**, 99 (2019).
9. Deringer VL, Csányi G. Machine learning based interatomic potential for amorphous carbon. *Physical Review B* **95**, 094203 (2017).

Reviewers' Comments:

Reviewer #1:

Remarks to the Author:

The authors have responded adequately to my comments and I'm happy to recommend the article for publication.

I have read Reviewer 3's comments in detail as well as the authors' response. Here are some specific comments on the issues raised:

- **Model accuracy:** The authors correctly note that the best accuracy achievable with a strictly local potential depends strongly on the material in question. In my opinion, they have provided sufficient justification that their method performs about as well as can be expected. Moreover, the comparison with DFT in Figs. R3 and R4 is compelling evidence that their updated models can accurately model reaction pathways, which should address Reviewer 3's primary concern about model accuracy.
- **Hyperparameter selection:** In my opinion, Reviewer 3 is too critical of the authors' choice of hyperparameters. It is of course true that the authors could have done more to optimize the hyperparameters, but I don't think it is essential to the study. The values the authors chose struck me as reasonable, and Fig. R5 in their rebuttal is convincing evidence that increasing the cutoff radius does not meaningfully affect model accuracy, although it would be helpful if the RMSEs with both sets of hyperparameters were reported in the figure.
- **Minor comments:** The authors responded to Reviewer 3's minor comments convincingly.

Reviewer #3:

Remarks to the Author:

In this revision the authors have clarified further details about their computational procedure, fixed mistakes they previously overlooked and finally put forward convincing arguments for explaining the inherently low accuracy of their GAP. These arguments refer to the hypothesis that long-range interactions which play a role in some carbon chemistries and cannot be described by short-ranged MLPs. The deeper understanding of the capabilities of their GAP puts the authors' work into a more favorable context that offers the scientific community a deeper insight into the capabilities of MLPs. Although this insight is not new – the authors merely refer to the paper by Deringer et al. establishing the database they utilize themselves – a transparent assessment of their GAP has an educational value, especially in light of the new application the authors are demonstrating. Unfortunately, the authors only state this hypothesis in the rebuttal but did not share it in their manuscript and do not include any quantitative data to support it. I believe the authors would direly need to do so and provide a few minor tests of their own in order to demonstrate the long-range effects in their application (see my comments below for details). Without such a transparent rationalization of their GAP shortcomings, the manuscript appears misleading towards the actual quality of their results. Since the latter needs to be avoided in order not to harm for the MLP community standards as stated in my previous reviews, I can only recommend this manuscript for publication after this part is added.

Detailed comments:

- The argument that a high MAE derives from badly described long-range interactions in DFT in the case of graphite or amorphous carbon which are also contained in the GAP17 database used for training is convincing. Whether this is also the reason for the lower accuracy in the predicted barriers is unclear, however. Further, Cu metal interactions are short-ranged and I think the authors need to demonstrate that the carbon structures they grow in their simulations are subject to this long-range interaction. In that sense I would like to see the MAE of the authors GAP on the different groups of structures they used for training and production runs, i.e. separated by element for GAP20, and actively learned data, as well as for some example structures of their final simulations. This data should convincingly support the authors' argument. All this data and

discussion should be presented in the manuscript and SI.

- It is essential to emphasize that per-atom energies may obscure accuracy. For clarity and precision, I would still prefer that the authors present the total reaction barriers, along with their deviations, either parenthetically as labels in the plot or within the main text. This will contribute to a more transparent and comprehensive representation of the findings.
- The per-atom regularization approach, particularly when applied to systems with very different chemistries, such as graphite-like structures and amorphous carbon, is in contrast to the authors' notion, favorable and allows for a better flexibility to achieve higher accuracy. I agree that finding an optimal balance is challenging, often requiring algorithmic assignment and I don't expect the authors to retrain and optimize at this stage. However, I would think that this practice should be acknowledged for educational purpose in the manuscript, i.e. that it can potentially improve results.
- Considering the other hyperparameters, sparse points and cutoff: The rationale for the high number of sparse points remains unclear to me. I would understand that authors view this as a design choice based on the GAP-17 work. But motivation for such decisions would enhance clarity since the GAP performance is impeded by a factor of three with no visible gain in accuracy. I am confused about the authors notion for the choice of cutoff = 3.7 Å due the deterioration of accuracy when this deterioration commences already at > 3 Å according to Figure S2. Also, it would be useful if the authors could state MAEs for the different hyperparameter sets shown in R5. The correlation plots alone do not show much.
- The explanation of when on-the-fly and when iterative training is used is still confusing to me. I hope the authors can simplify this explanation.

Dear Reviewers,

We express our sincerest appreciation for the time and expertise you have dedicated to reviewing our manuscript. We have added more tests and implemented necessary changes to address the issues and concerns raised by Reviewers.

To Reviewer #1:

The authors have responded adequately to my comments and I'm happy to recommend the article for publication. I have read Reviewer 3's comments in detail as well as the authors' response. Here are some specific comments on the issues raised:

- **Model accuracy:** The authors correctly note that the best accuracy achievable with a strictly local potential depends strongly on the material in question. In my opinion, they have provided sufficient justification that their method performs about as well as can be expected. Moreover, the comparison with DFT in Figs. R3 and R4 is compelling evidence that their updated models can accurately model reaction pathways, which should address Reviewer 3's primary concern about model accuracy.
 - **Hyperparameter selection:** In my opinion, Reviewer 3 is too critical of the authors' choice of hyperparameters. It is of course true that the authors could have done more to optimize the hyperparameters, but I don't think it is essential to the study. The values the authors chose struck me as reasonable, and Fig. R5 in their rebuttal is convincing evidence that increasing the cutoff radius does not meaningfully affect model accuracy, although it would be helpful if the RMSEs with both sets of hyperparameters were reported in the figure.
 - **Minor comments:** The authors responded to Reviewer 3's minor comments convincingly.
- ✓ Thank you very much for the encouraging feedback on our revised manuscript. We are delighted that our amendments have met with approval. Your recommendation for publication in Nature Communications is deeply appreciated.

To Reviewer #3:

In this revision the authors have clarified further details about their computational procedure, fixed mistakes they previously overlooked and finally put forward convincing arguments for explaining the inherently low accuracy of their GAP. These arguments refer to the hypothesis that long-range interactions which play a role in some carbon chemistries and cannot be described by short-ranged MLPs. The deeper understanding of the capabilities of their GAP puts the authors' work into a more favorable context that offers the scientific community a deeper insight into the capabilities of MLPs. Although this insight is not new – the authors merely refer to the paper by Deringer et al. establishing the database they utilize themselves – a transparent assessment of their GAP has an educational value, especially in light of the new application the authors are demonstrating. Unfortunately, the authors only state this hypothesis in the rebuttal but did not share it in their manuscript and do not include any quantitative data to support it. I believe the authors would direly need to do so and provide a few minor tests of their own in order to demonstrate the long-range effects in their application (see my comments below for details). Without such a transparent rationalization of their GAP shortcomings, the manuscript appears misleading towards the actual quality of their results. Since the latter needs to be avoided in order not to harm for the MLP community standards as stated in my previous reviews, I can only recommend this manuscript for publication after this part is added.

Dear Reviewer,

Thank you very much for your meticulous review and insightful comments. Your expertise in machine learning force fields has significantly enriched our understanding and consequently enhanced the quality of this work. The specificity and depth of your feedback have enabled us to articulate our arguments with enhanced clarity. We deeply appreciate your time and effort to advance our work, and we hope we have addressed all the details you mentioned by making the necessary additions and modifications. Below are our detailed responses.

Detailed comments:

- The argument that a high MAE derives from badly described long-range interactions in DFT in the case of graphite or amorphous carbon which are also contained in the GAP17 database used for training is convincing. Whether this is also the reason for the lower accuracy in the predicted barriers is unclear, however. Further, Cu metal interactions are short-ranged and I think the authors need to demonstrate that the carbon structures they grow in their simulations are subject to this long-range interaction. In that sense I would like to see the MAE of the authors GAP on the different groups of structures they used for training and production runs, *i.e.* separated by element for GAP20, and actively learned data, as well as for some example structures of their final simulations. This data should convincingly support the authors' argument. All this data and discussion should be presented in the manuscript and SI.

- ✓ To elucidate the influence of long-range interactions on the carbon structures formed on metallic surfaces in our simulations, we constructed seven distinct data sets, details of which are presented in **Fig. R1**. The first set (Set#1) comprises 2000 pure-carbon structures, which were randomly sampled from GAP20. For the data derived from active learning and the final simulation structures, we categorized them based on the number of carbon atoms into two groups: those associated with a Cu slab containing fewer than 10 carbon atoms (C_{1-10} on Cu) and those with more than 10 carbon atoms (C_{10-100} on Cu, **Fig. R1a**). To evaluate the impact of long-range interactions across various metal-carbon configurations, we also created two groups: Ti and Cr slabs, each with fewer than 10 carbon atoms, both of which were derived from actively learned data.

- ✓ **Fig. R1b** displays force correlations for three groups: GAP20 structures, C_{1-10} on Cu, and

$C_{10\sim 100}$ on Cu. GAP20 structures exhibit the strongest long-range interactions. Initially, carbon monomers primarily interact with the Cu(111) surface, reducing both long-range interactions and the force mean absolute error (MAE). As deposition progresses, carbon atoms accumulate on the Cu surface and form clusters. This resurgence in long-range interactions can influence the force accuracy of machine learning potentials (MLPs). This trend persists in the test of final Cu-C simulation structures. In contrast, the Cr-C and Ti-C systems, with fewer carbon atoms, achieve a force accuracy of 0.08 eV/\AA , indicating minimized long-range interaction effects (**Fig. R1c**).

Fig. R1. Test of long-range interactions for metal-carbon systems. (a) Three examples of the testing structural groups. (b) Force correlations of GAP20 structures, $C_{1\sim 10}$ on Cu, and $C_{10\sim 100}$ on Cu. (c) Force and energy MAE of seven different data sets, primarily encompassing varying numbers of carbon atoms and different metal elements. Specific labels can be found in the inset provided.

- ✓ **In summary**, our testing confirms that the carbon structures we grow are influenced by long-range interactions, further impacting the force MAE and the accuracy of the energy barrier. In short-range systems, such as Cr-C and Ti-C, force accuracy markedly improves. We hope this data would address the reviewer's concerns, and we've included additional

tests and data in our Supplementary Information (SI).

- It is essential to emphasize that per-atom energies may obscure accuracy. For clarity and precision, I would still prefer that the authors present the total reaction barriers, along with their deviations, either parenthetically as labels in the plot or within the main text. This will contribute to a more transparent and comprehensive representation of the findings.

- ✓ Thank you for your valuable feedback. In accordance with your suggestion, we've presented the total reaction barriers in the SI, depicted in **Fig. R2**, which corresponds to **Fig. 4** in the main manuscript. The absolute error for the reactions in **Fig. R2** ranges from 0.1 eV to 0.49 eV. These deviations in total energy may arise from three primary factors: the precision of CGM-MLP, long-range interactions, and the number of atoms involved in the reactions.

While total energy barriers can indicate the overall predictive accuracy for a reaction, it's worth considering that reactions involving more atoms could potentially exhibit higher total energy errors. Nonetheless, when viewed on a per-atom basis, these errors may be relatively small, thereby ensuring that the predicted reaction pathway remains highly reliable in practice. To ensure a transparent and comprehensive error assessment, we've included both per-atom energy and total energy barriers. Additionally, we've incorporated these total energy barriers in the main text.

Fig. R2. Comparison of total energy barriers as calculated by DFT and CGM-MLP.

✓ We sincerely value your meticulous review and are committed to enhancing the clarity and precision of our work.

- The per-atom regularization approach, particularly when applied to systems with very different chemistries, such as graphite-like structures and amorphous carbon, is in contrast to the authors' notion, favorable and allows for a better flexibility to achieve higher accuracy. I agree that finding an optimal balance is challenging, often requiring algorithmic assignment and I don't expect the authors to retrain and optimize at this stage. However, I would think that this practice should be acknowledged for educational purpose in the manuscript, *i.e.* that it can potentially improve results.

✓ Thank you for your constructive feedback. Indeed, the per-atom regularization approach has room for further refinement to optimize the accuracy of our training framework. We can consider employing varying force or energy thresholds for different deposition stages, or implementing system-dependent total-energy training convergence thresholds. Moreover, fully addressing the long-range interaction challenges is pivotal for improving the accuracy of simulations involving metal-catalyzed graphene. This aspect truly necessitates advancements in foundational algorithms, which we plan to delve into in our future research.

In our revised manuscript, we have also added further commentary in Discussion section to highlight that the aforementioned improvements are worth exploring within our framework, emphasizing the educational value of this research. *i.e.*, “Given the distinct effects of long-range interactions across different metal-carbon systems, strategies such as adopting varying force or energy thresholds for distinct deposition stages, or setting system-dependent total-energy training convergence thresholds, can be considered to further enhance accuracy. Yet, addressing the challenges posed by long-range interactions remains crucial for refining the precision and adaptability of simulations, especially those centered on metal-catalyzed low-density carbon structures.”

- Considering the other hyperparameters, sparse points and cutoff: The rationale for the high number of sparse points remains unclear to me. I would understand that authors view this as a design choice based on the GAP-17 work. But motivation for such decisions would enhance clarity since the GAP performance is impeded by a factor of three with no visible gain in accuracy. I am confused about the authors notion for the choice of cutoff = 3.7 Å due the deterioration of accuracy when this deterioration commences already at > 3.0 Å according to Figure S2. Also, it would useful if the authors could state MAEs for the different hyperparameter sets shown in R5. The correlation plots alone do not show much.

- ✓ Thank you for your comment regarding the hyperparameters. Our choice of a high number of sparse points, as you rightly pointed out, was indeed influenced by the methodologies outlined in the GAP-17 work, aiming to ensure a robust selection. However, we'd like to gently point out that the choice of 9000 sparse points might not have as pronounced an effect on the GAP performance as initially perceived. As shown in **Fig. R3**, the force MAE continues to decrease notably until reaching 5000 sparse points. Therefore, settling on 5000

sparse points might be a risky decision, given that different metal-systems may have slightly varied optimal sparse point values. Opting for 7000 sparse points, the training time is approximately 1200 seconds, which is 500 seconds less than when choosing 9000 sparse points. Thus, the increase is not as dramatic as three times the training time. Additionally, as kindly noted by Reviewer #1, we acknowledge that there might have been room for further hyperparameter optimization. However, in the context of our study, its influence on accuracy appeared to be relatively minor.

- ✓ Nevertheless, we truly value and are grateful for your thoughtful and thorough insights regarding parameter selection.

Fig. R3. Hyper-parameter sparse point tests of the Deposited-C@Cu(111) set.

- ✓ Regarding the cutoff radius, our decision to use 3.7 Å is based on two primary considerations. Firstly, the spacing between two graphitic layers is roughly 3.3 Å, which necessitates a cutoff radius larger than this value. Secondly, as demonstrated in **Fig. R4**, interactions between two isolated carbon atoms become approximately zero when their distance surpasses 3.7 Å. A smoother potential function can prevent the occurrence of unphysical atomic forces.

Fig. R4. Potential-energy scans for an isolated carbon dimer.

- ✓ We have now tabulated the MAEs for each hyperparameter set in previous **Fig. R5**, *i.e.*, also **Fig R5** in this response. The data demonstrates that CGM-MLPs, when trained with two different groups of parameters, yield similar accuracy levels. Specifically, the energy and force MAEs are approximately 0.013 eV and 0.43 eV/Å, respectively. While optimizing hyperparameters can offer improvements, the crucial factor in enhancing force prediction accuracy lies in comprehending the nonlocal interactions inherent in carbon structures.

Fig. R5. Comparative analysis of accuracy for the CGM-MLP trained with varying parameters. (a) Force correlation (b) Energy correlation.

- The explanation of when on-the-fly and when iterative training is used is still confusing to me. I

hope the authors can simplify this explanation.

- ✓ Thank you for your question, and we apologize for any ambiguity in our description. To clarify, the training iteration consists of four sequential steps:

1. MLP training
2. MD/tfMC simulation
3. On-the-fly training data selection
4. Accuracy evaluation.

In the step 4, we assess the current energy and force MAE against specified thresholds. If the current MAE is below these thresholds, the iterative training progresses to the next round. During this round, the number of deposited carbon atoms increases, but no MLP training is conducted. If the MAE exceeds the thresholds, all four steps are repeated.

Whenever the existing MLP fails to meet the accuracy criteria, the selected structures are incorporated into the training set. They are then fitted once more to produce a new MLP. As such, our training framework exemplifies a typical active or on-the-fly training procedure.

We sincerely appreciate the time and effort you have invested in reviewing our manuscript. Your constructive feedback and insightful comments have been invaluable in enhancing the quality and rigor of our work.

Warm regards,

Di Zhang (Email: zhangdi2015@sjtu.edu.cn)

Peiyun Yi (Email: yipeiyun@sjtu.edu.cn)

Linfa Peng (Email: penglinfa@sjtu.edu.cn)

Xinmin Lai (Email: xmlai@sjtu.edu.cn)

Hao Li (Email: li.hao.b8@tohoku.ac.jp)

Reviewers' Comments:

Reviewer #3:

Remarks to the Author:

In this revision the authors have addressed last points that remained unclear and added comments into the manuscript that were previously missing, but needed to be contained for transparency. I thank the authors for these efforts. After clarifying and improving their manuscript, it includes enough information that readers can fully understand the methodology and assess the quality of the work. These points are key to allow for reproducibility which I strongly believe would otherwise not have been possible.